# TRAINED ON TOKENS, CALIBRATED ON CONCEPTS:
## THE EMERGENCE OF SEMANTIC CALIBRATION IN LLMS

**Preetum Nakkiran**
Apple

**Arwen Bradley**
Apple

**Adam Golinski**
Apple

**Eugene Ndiaye**
Apple

**Michael Kirchhof**
Apple

**Sinead Williamson**
Apple

## ABSTRACT

Large Language Models (LLMs) often lack meaningful confidence estimates for their outputs. While base LLMs are known to exhibit next-token calibration, it remains unclear whether they can assess confidence in the actual meaning of their responses beyond the token level. We find that, when using a certain sampling-based notion of semantic calibration, base LLMs are remarkably well-calibrated: they can meaningfully assess confidence in various open-ended question-answering tasks, despite being trained on only next-token prediction. To formalize this phenomenon, we introduce "$B$-calibration," a notion of calibration parameterized by the choice of equivalence classes. Our main theoretical contribution establishes a mechanism for why semantic calibration emerges in base LLMs, leveraging a recent connection between calibration and local loss optimality. This theoretical mechanism leads to a testable prediction: base LLMs will be semantically calibrated when they can easily predict their own distribution over semantic answer classes before generating a response. We state three implications of this prediction, which we validate through experiments: (1) Base LLMs are semantically calibrated across question-answering tasks, (2) instruction-tuning procedures systematically break this calibration, and (3) chain-of-thought reasoning breaks calibration (intuitively because models cannot predict their final answers before completing their generation). To our knowledge, our work provides the first principled explanation of when and why semantic calibration emerges in LLMs.

## 1 INTRODUCTION

As Large Language Models (LLMs) become increasingly capable, it is important to understand the nature and extent of their uncertainty. Addressing this is an active research question: can we extract a meaningful notion of confidence in an LLM's answers? This question is scientifically interesting even aside from applications: it is a way of asking, informally, do LLMs "know what they don't know"? (Kadavath et al., 2022)

In the classification literature, one well-understood criterion for uncertainty quantification is *calibration*: do the predicted probabilities reflect empirical frequencies? For example, if an image classifier is 80% confident on a set of inputs, then it should be correct on 80% of those predictions. To apply this definition to LLMs, one approach is to treat the LLM as a classifier that predicts the next-token, given all previous tokens. There is strong empirical and theoretical evidence that base LLMs, which are only pre-trained with the maximum likelihood loss, are typically *next-token-calibrated* (OpenAI, 2023; Zhang et al., 2024; Desai & Durrett, 2020). Next-token calibration is a meaningful notion of calibration in certain settings like True/False or multiple choice questions, where a single token encapsulates the entire response (Kadavath et al., 2022; Plaut et al., 2025). However, when the model produces long-form answers to open-ended questions, we desire a notion of uncertainty with respect to the *semantic meaning* of the response, which next-token calibration does not directly capture.

Prior works have proposed a variety of notions of semantic confidence for long-form text, including verbalized measures and sampling-based measures (e.g. *semantic entropy* of Farquhar et al. (2024)). See Vashurin et al. (2025) for a comprehensive overview. However, from the empirical data it is unclear whether LLMs are naturally calibrated with respect to any of these notions of confidence,

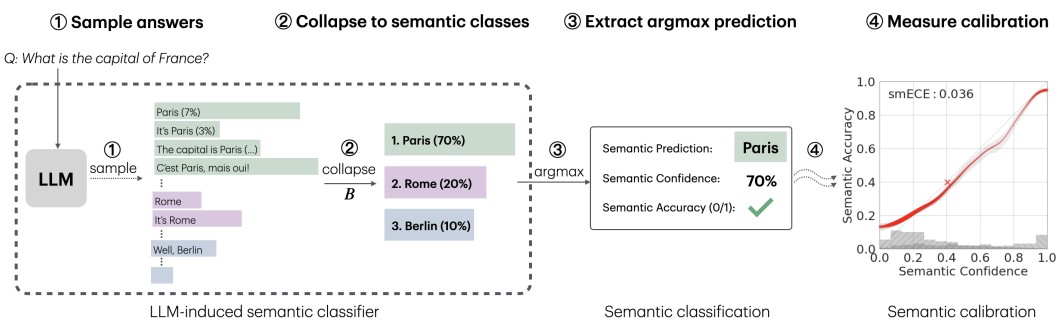

Figure 1: **Semantic calibration** refers to calibration of an *LLM-induced semantic classifier* (dashed box): the classifier induced by post-processing LLM outputs with a given semantic collapsing function, which we refer to as $B$ throughout. To measure semantic confidence calibration: for a given question, sample multiple temperature $T=1$ generations, and extract semantic answers by applying the collapsing function $B$ (e.g. a strong LLM prompted to extract one-word answers). This yields an empirical distribution over semantic classes (above: Paris, Rome, Berlin), which we treat as the classifier output. This classifier output defines a semantic prediction (=argmax probability) and a semantic confidence (=max probability). *Semantic confidence calibration* means, over all questions, these predictions are confidence-calibrated in the standard classification sense.

without being specifically trained for calibration (Kadavath et al., 2022; Yin et al., 2023; Band et al., 2024; Kapoor et al., 2024; Yoon et al., 2025; Mei et al., 2025). Empirically, calibration may depend on many factors: the test distribution (math, trivia, etc.), the post-training procedure (RLHF, DPO, RLVR, none, etc.), the inference-time procedure (few-shot examples, chain-of-thought (CoT), best-of-K, etc.), the model size, the model architecture, the sampling temperature, etc. All of these factors have been posited to affect calibration, for reasons that are not yet well understood (Kadavath et al., 2022; OpenAI, 2023; Leng et al., 2025; Xiao et al., 2025; Zhang et al., 2024; Wang et al., 2025).

A priori, there is no reason to expect *emergence* of any of these forms of semantic calibration as a product of standard pre-training with the maximum likelihood loss. In this work, we show both theoretically and empirically that a particular type of sampling-based semantic calibration actually does emerge for a large class of LLMs. Our definition is closely related to semantic entropy (Farquhar et al., 2024), as well as the sampling-based definitions of confidence in Wang et al. (2023), Wei et al. (2024), and Lamb et al. (2025). At a high level, our approach involves treating the LLM as a standard multi-class classifier (by collapsing outputs with the same semantic meaning), and then applying recent theoretical results from the literature on classifier calibration (Gopalan et al., 2024; Błasiok et al., 2023; 2024). Fig. 1 illustrates the overall setup, described in detail in the next section. To our knowledge, our work is the first to propose a theoretically plausible mechanism for semantic calibration in LLMs, and we validate the predictions of this theory empirically.

**Summary of Contributions.** We empirically show that LLMs *are* semantically-calibrated surprisingly often, for certain settings and types of questions. We offer a candidate theoretical mechanism to explain how this calibration emerges from standard LLM training (that does not explicitly encourage it), and discuss under which settings and for which questions we expect it. The basic prediction of our theory is that semantic calibration is likely to hold when (1) the model is a base LLM, and (2) the model is able to *directly* predict the probability that its answer will land in a given semantic class, even before it has started to generate it. Intuitively, in order to be semantically calibrated, the model must "know" how likely it is to generate a "Paris"-type answer, before it has determined exactly how it will phrase its answer. This theoretical insight leads to a number of practical predictions about which models and tasks should be semantically calibrated, which we then test experimentally.

**Organization.** We start by formally defining the notions of calibration we consider in Sec. 2. In Sec. 3, we introduce our proposed theoretical mechanism for emergent calibration, and state our formal results. In Sec. 4, we apply the theory to make three concrete predictions about when LLMs are semantically calibrated, and in Sec. 5, we experimentally test these predictions.

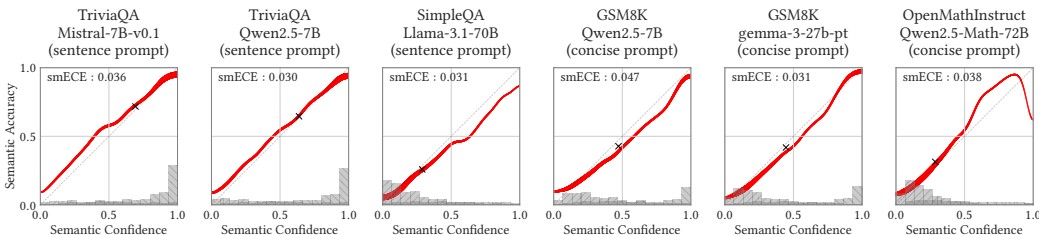

Figure 2: Reliability diagrams demonstrating *semantic confidence-calibration* of base (pretrained-only) LLMs across various combinations of datasets, models, and prompts. Calibration error measured with SmoothECE (smECE), average confidence and accuracy marked with a black cross, and density of semantic confidences shown in gray histogram; details in Appendix D.1.

## 2 SEMANTIC CALIBRATION AND $B$-CALIBRATION

We now informally describe our framework; formal definitions follow in Sec. 2.1. The core of our approach is a collapsing function $B$ which post-processes the LLM's raw text outputs, mapping each generation to one of a finite set of classes. Of particular interest are *semantic collapsing functions*[1], which we focus on now. As illustrated in Fig. 1, a semantic collapsing function implicitly transforms the LLM into an *LLM-induced semantic classifier*: For a given question, the classifier's output is a distribution over semantic classes, whose probabilities can be empirically estimated by sampling multiple generations from the LLM and applying $B$ to each. From this distribution, we define the semantic confidence as the probability of the most-likely semantic class, and the semantic accuracy as whether the most-likely semantic class matches the ground truth's semantic class. The LLM is *semantically confidence-calibrated* if these confidences and accuracies are calibrated across a dataset—e.g., among questions with 70% semantic confidence, the average semantic accuracy is also 70%. This definition coincides with Lamb et al. (2025)'s definition of "Empirical Semantic Confidence" when applied to the full distribution. For example, Fig. 2 measures calibration of several models using this approach (full experimental details in Sec. 5).

### 2.1 NOTATION AND SETUP

We now establish the notation used throughout the paper. We assume that our semantic collapsing function outputs at most $K \in \mathbb{N}$ classes, which we represent by the set of indices $[K] \equiv \{1, \ldots, K\}$. We allow $K$ to be arbitrarily large. We identify these classes with the set of standard basis vectors $\mathcal{E}_K \subset \mathbb{R}^K$. The set of probability distributions over a finite set $S$ is denoted $\Delta(S)$. For convenience, we use the shorthand $\Delta_K \equiv \Delta([K])$ for the probability simplex over the $K$ classes.

**Language Model and Data.** Let $\mathcal{V}$ be the model's vocabulary. We assume throughout that the evaluation data comes from a ground-truth distribution $\mathcal{D}$ over prompt-completion pairs $(x, y) \in \mathcal{V}^* \times \mathcal{V}^N$, where $N$ is a maximum generation length. An LLM is a function $p_\theta : \mathcal{V}^* \to \Delta(\mathcal{V}^N)$ that maps a prompt $x$ to a distribution over output strings. We use conventional notation: $p_x \equiv p_\theta(\cdot \mid x)$ is the entire distribution over sequences for a given prompt, so we can denote $p_x(z) = p_\theta(z \mid x)$ as the probability of a specific sequence $z$. The conditional probability of the next token is denoted $p_\theta(z_i \mid x, z_{<i})$. To distinguish model outputs from the dataset, we use $z \in \mathcal{V}^N$ for generated strings and $y \in \mathcal{V}^N$ for ground-truth completions from $\mathcal{D}$.

**Collapsing function.** The core of our framework is the collapsing function $B : \mathcal{V}^* \times \mathcal{V}^N \to [K]$ that classifies a given prompt-completion pair into one of $K$ categories. In our theory, $B$ is allowed to be arbitrary, but we often will think of it as a **"semantic collapsing" function**, grouping many different strings into a single semantic class, as visualized in Fig. 1. An example of such a function is described in App. D. For convenience, we write $B_x(z) \equiv B(x, z)$ to emphasize its role as a classifier for outputs $z$ given a fixed prompt $x$.

---

[1]To implement this function, we use a strong auxiliary LLM prompted to extract a canonical short answer from a long-form string. Details in App. D.

## 2.2 Confidence Calibration

We first recall the relevant definitions of calibration in the multi-class setting (for a unified treatment, see Gopalan et al. (2024, Section 2)). In the $K$-class setting, classifiers output values $c \in \Delta_K$ and the true labels take values $y \in \mathcal{E}_K$ (one-hot encodings). Calibration is a property defined for *any* joint distribution of prediction-label pairs $(c, y) \in \Delta_K \times \mathcal{E}_K$, regardless of whether it was generated by a classifier. We will focus primarily on *confidence calibration*, which only considers the probability assigned to the predicted class; however, we provide analogous results for full calibration in App. E.3. The following definition is standard:

**Definition 1** (Confidence-calibration). *A distribution $\mathcal{D}$ over prediction-output pairs $(c, y) \in \Delta_K \times \mathcal{E}_K$ is* perfectly confidence-calibrated *if*

$$\mathbb{E}_{(c,y)\sim\mathcal{D}} [y_{k^\star} - c_{k^\star} \mid c_{k^\star}] \equiv 0 \text{ where } k^\star \leftarrow \underset{k\in[K]}{\operatorname{argmax}} c_k.$$

The definition depends crucially on the distribution $\mathcal{D}$. In this work we take $\mathcal{D}$ to be the evaluation distribution of interest (e.g. TriviaQA, GSM8k, etc), unless otherwise specified.

**From Language Model to Categorical Predictor** For a given prompt $x$, we obtain a distribution over $K$ categories by pushing-forward[2] the LLM's output distribution $p_\theta(\cdot \mid x)$ via the function $B_x$. Specifically, the distribution over categories $\pi_x := B_x \sharp p_x \equiv B_x \sharp p_\theta(\cdot \mid x)$ assigns to each category $k \in [K]$ the sum of probabilities of all strings $z$ that $B_x$ maps to that category:

$$(B_x \sharp p_x)(k) = \underset{z \sim p_\theta(\cdot|x)}{\operatorname{Pr}} [B_x(z) = k] = \sum_{z \,:\, B_x(z)=k} p_\theta(z \mid x). \tag{1}$$

This process transforms the original prompt-answer pair $(x, y)$ from the dataset $\mathcal{D}$ into a pair suitable for calibration analysis: $(B_x \sharp p_x, B_x(y))$, where $B_x \sharp p_x$ is the model's predicted distribution over categories and $B_x(y)$ is the ground-truth category. Now, we say that the model $p_\theta$ is $B$-confidence-calibrated if the induced distribution over $(B_x \sharp p_x, B_x(y))$ is confidence-calibrated. That is, $B$-confidence-calibration means if the generated and ground-truth answers are both post-processed by $B$, then the resulting $K$-way-classifier is confidence-calibrated.

**Definition 2** ($B$-confidence-calibration). *The model $p_\theta$ is $B$-confidence-calibrated* with respect to distribution $\mathcal{D}$ if the induced distribution over pairs $(B_x \sharp p_x, B_x(y)) \in \Delta_K \times [K]$ is perfectly confidence-calibrated (per Definition 1).*

Our entire framework is well-defined for any arbitrary computable function $B$, though we usually choose $B$ to be a semantic-collapsing function. In general, an LLM might be $B$-calibrated for some choices of $B$, but not others—one goal of our theory is to understand why.

## 3 Theoretical Mechanism

Our conjectured mechanism for emergent calibration builds on the work of Błasiok et al. (2023; 2024) which connects the *statistical* property of calibration to the *optimization* property of local loss optimality. The core intuition is that a miscalibrated model implies the existence of a "simple" perturbation to the model that would reduce its test loss. For example, suppose an LLM is semantically miscalibrated in the following way: on questions where it is 70% semantically-confident, it is on average only 60% accurate. Then, an obvious way to improve the LLM's test loss is: whenever the original LLM was 70% semantically confident, it should downweight the probability mass it places on all strings in its majority semantic class, thereby decreasing its confidence. We argue that base LLMs, trained to minimize cross-entropy loss, should not leave such "easy wins" on the table, and thus should be well-calibrated.

This example reveals some of the subtlety in the LLM setting: unlike standard classifiers, the LLM does not explicitly output its [semantic] confidences. Thus to implement such a loss-improving perturbation during pretraining, the LLM must implicitly "know" its semantic confidence for a

---

[2]We use "$\sharp$" as the standard notation for the mathematical pushforward of a measure by a function. E.g. for a function $B$ and distribution $p$, the notation $B\sharp p$ denotes the distribution of $\{B(x)\}_{x\sim p}$

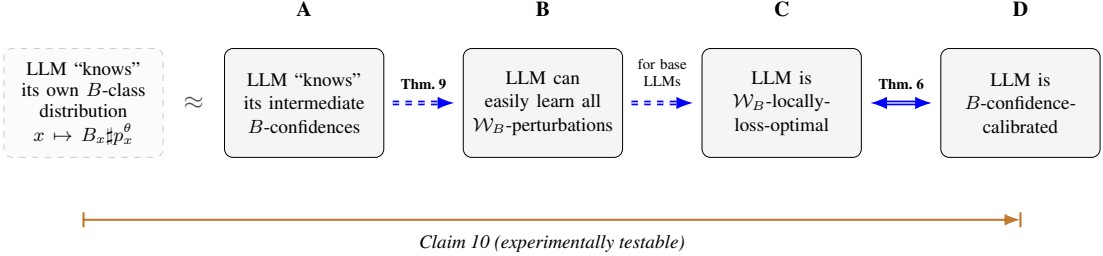

Figure 3: Conjectured Mechanism for Semantic Calibration. Implications have varying levels of support: the solid blue arrow (⟷) has a formal proof; the dashed blue arrows (= =►) have proofs of "morally similar" (but weaker) implications. Claim 10 encompasses the full chain of implications, and has experimental support.

given question even before generating its answer—in order to know what type of upweighting/downweighting of answer strings is required. In settings where the LLM does not "know" its semantic confidences (informally), we may expect poor calibration—we will see this aspect in both our theory and experiments. A technical overview of our results is in Sec. 3.1, followed by formal theorem statements in Sec. 3.2 and Sec. 3.3. All proofs are deferred to App. E.

## 3.1 CONJECTURED MECHANISM: OVERVIEW

Fig. 3 illustrates our conjectured mechanism. There are three main steps in the conjecture, with different degrees of evidence for each step. For the first step, we have a fully rigorous proof. For the other two steps, we have partial theoretical evidence: proofs of weaker claims which are "morally similar" to our conjectured claim. Finally, we have experimental evidence for our overall conjecture (presented later in Sec. 5). We outline each step below, following Fig. 3 from right-to-left.

**(C) ⟷ (D):** The first step of our argument, described in more detail in Sec. 3.2, is a general equivalence between calibration and local loss optimality. We say an LLM is locally-loss-optimal if its test loss cannot be improved by post-processing its output distribution via any function in some given set of perturbation functions (formally, Definition 5). For a particular choice of perturbation functions, this turns out to exactly characterize $B$-calibration. We prove in Thm. 6 that for any choice of collapsing function $B$, $B$-confidence-calibration is *equivalent* to local-loss-optimality with respect to a corresponding family of perturbations, denoted $\mathcal{W}_B$. Roughly speaking, this family $\mathcal{W}_B$ consists of perturbations like our earlier example: "If the $B$-class-confidence was 70%, then downweight the probability of generating all strings in the majority $B$-class." Overall, Thm. 6 tells us that if we want to understand when LLMs are $B$-confidence-calibrated, we can equivalently understand which types of perturbations LLMs are loss-optimal with respect to.

**(B) = =► (C):** At this point, we invoke an informal assumption proposed in Błasiok et al. (2023), and likely folklore much earlier: we assume that base LLMs are nearly locally-loss-optimal on their pretraining distribution, w.r.t. any perturbation that is "easy" for the LLM to learn. We state this assumption more precisely as Claim 16 in the Appendix. Błasiok et al. (2024, Theorem 1.2) offers partial theoretical justification for this claim, by proving that, intuitively, if small models can represent a set of perturbations, then ERM over a family of slightly larger models yields local loss optimality w.r.t. these perturbations; this serves as a approximate representational analog of the desired assumption. The intuition is that pretraining should not leave any easy wins on the table: if a simple (i.e. easily-learnable) perturbation could have improved the test loss, the LLM would have learned it during training. We agree with Błasiok et al. (2023) that this assumption is plausible, because it is fairly weak; it does not require that models are *globally* optimal in any sense.

**(A) = =► (B):** From the above two points, we can conclude that a base LLM will be $B$-confidence-calibrated if the corresponding perturbation family $\mathcal{W}_B$ is simple for the LLM to learn. But when is $\mathcal{W}_B$ simple to learn? This is subtle because the perturbations $\mathcal{W}_B$ are defined over the *sequence-level* probability distribution but LLMs must implement perturbations by modifying *next-token* probabilities. For example, in order to implement a perturbation such as "increase the probability of ultimately generating a Paris-type answer", the model must begin by deciding how to adjust its *first*

*token* probabilities in order to achieve this. We bridge this gap in Thm. 9, by proving a representational analogue of the implication **(A)** ⇢ **(B)** of Fig. 3: we show that if the LLM "knows" its own induced distribution over $B$-classes at each intermediate point during generation (even the very beginning), then it can implement the associated family of perturbations $\mathcal{W}_B$ in a "simple" way. (Notably, this does not require the model to know the *correct answer's* $B$-class, only that of its own generation.) Formally, we prove a circuit-complexity version of this: the next-token probabilities of the perturbed model can be computed with a shallow circuit given oracle access to the intermediate $B$-confidence functions, and the original next-token probabilities. In practice, we will focus primarily on the model's ability to predict its $B$-distribution at the beginning of generation (before outputting the first token). Intuitively, this is more likely to hold for straightforward questions such as "What is the capital of France?" than questions requiring many steps of reasoning— we will say more about this in the following section.

Putting everything together, the overall mechanism predicts that a base LLM will be $B$-confidence-calibrated if the LLM "knows" the distribution of $B$-classes of its own answers (i.e. if it can be LoRA-adapted to immediately output this $B$-class distribution, given only the question). When $B$ is a semantic collapsing function, this theory naturally suggests a number of practical predictions about which models and tasks should be semantically calibrated, which we explore and test experimentally in Sec. 5. The next several sections give the formal theory supporting the mechanism we have just outlined.

## 3.2 $B$-CALIBRATION AND LOCAL LOSS OPTIMALITY

We now setup and establish the equivalence between calibration and local loss optimality (Thm. 6). We use the sequence-level cross-entropy loss, which decomposes into the standard autoregressive next-token log-loss: $\mathbb{E}_{(x,y)\sim\mathcal{D}}[\ell(y, p_x)] = \mathbb{E}_{(x,y)\sim\mathcal{D}}\left[-\sum_{i\in[N]}\log p_\theta(y_i \mid y_{<i}, x)\right]$. We will use the following notion of perturbing a distribution, known as *exponential tilting* (Cover & Thomas, 1999, Chapter 11), which turns out to be the appropriate notion for the cross-entropy loss.

**Definition 3** (Perturbation operator). *Given a distribution $f \in \Delta(\mathcal{V}^N)$ over sequences, and a signed measure $w \in \mathbb{R}^{|\mathcal{V}^N|}$, define the perturbed distribution $(f \star w) \in \Delta(\mathcal{V}^N)$ as:*

$$\forall z \in \mathcal{V}^N : \quad (f \star w)[z] := \mathrm{softmax}\big(w[z] + \log f[z]\big). \tag{2}$$

Next we define a specific class of perturbations which characterize $B$-confidence-calibration. Intuitively, these perturbations modify the probability of the most-likely $B$-class, by modifying the probability of each string $z$ according to (only) its $B$-class $B_x(z)$. The formal definition is somewhat technical, based on the language of weighted calibration developed in Gopalan et al. (2024).

**Definition 4** (Semantic Perturbation Function Classes). *Given an arbitrary collapsing function $B_x(z) \in [K]$, we define the class $\mathcal{W}_B$ of perturbation functions $w(x, p_x) \in \mathbb{R}^{|\mathcal{V}^N|}$ as follows. These functions $w(x, p_x)$ generate a perturbation vector based on the prompt $x$ and the model's predictive distribution $p_x$.*

$$\mathcal{W}_B := \{w \mid \exists \tau : [0,1] \to [-1,1] \,\forall z \in \mathcal{V}^N : w(x, p_x)[z] = \tau\big(\pi_x[k^*]\big) \cdot \mathbb{1}\{B_x(z) = k^*\}\},$$

$$\text{where } \pi_x := B_x\sharp p_x, \quad \text{and } k^* \leftarrow \underset{k\in[K]}{\mathrm{argmax}}\, \pi_x[k].$$

Finally, we define local loss optimality with respect to an arbitrary perturbation class $\mathcal{W}$.

**Definition 5** ($\mathcal{W}$-local loss optimality). *We say that $p_\theta$ is $\mathcal{W}$-locally loss-optimal if*

$$\forall w \in \mathcal{W} : \quad \underset{(x,y)\sim\mathcal{D}}{\mathbb{E}}[\ell(y, p_x)] \leq \underset{(x,y)\sim\mathcal{D}}{\mathbb{E}}[\ell(y, p_x \star w_x)] \quad \text{where } w_x \equiv w(x, p_x)\,, \ p_x \equiv p_\theta(\cdot \mid x).$$

We can now state the main result of this section (see App. E for all proofs).

**Theorem 6** (Equivalence of Calibration and Local Loss Optimality). *Given a model $p_\theta$, a collapsing function $B$, and a distribution $\mathcal{D}$, the model $p_\theta$ is perfectly $B$-confidence-calibrated on $\mathcal{D}$ if and only if $p_\theta$ is $\mathcal{W}_B$-locally loss-optimal on $\mathcal{D}$.*

**Remark 7.** *Thm. 6 states a simplified version of our full theoretical results, for the sake of clarity. Thm. 6 only characterizes perfect confidence-calibration, but it is possible to show a much more*

*robust equivalence: it turns out that a model is "close to" B-calibrated if and only if it is "close to" locally-loss-optimal in the appropriate sense. We state and prove this generalized version as Thm. 36 in App. E, where we also generalize to allow any arbitrary proper-loss $\ell$, and any notion of weighted-calibration (including canonical calibration and confidence calibration).*

### 3.3 SPECIALIZING TO AUTOREGRESSIVE MODELS

It remains to understand when the perturbation class $\mathcal{W}_B$ is easy for an LLM to learn (box **(B)** in Fig. 3). Although we cannot currently fully answer this question, we can gain insight by studying a simpler question of representation: when is a perturbation class $\mathcal{W}_B$ "easy" for the LLM to represent (for example, as a small circuit on top of the original LLM)? The main remaining challenge is that perturbations are defined on probability distributions over *sequences* (Definition 3), whereas autoregressive models must implement perturbations *token-by-token*. Fortunately, for perturbations in $\mathcal{W}_B$, it turns out the perturbed next-token distribution can be expressed as a simple re-weighting of the LLM's original next-token distribution. This re-weighting is governed by a set of scalar-valued functions $\{g_i\}$, defined below. We call these functions "intermediate $B$-confidences", because $g_i(z_{\leq i}; x)$ is the probability mass the model places on its most-likely $B$-class, given both the question $x$ and the response prefix $z_{\leq i}$ generated so far. Thus, the difficulty of representing the sequence-level perturbation reduces to the difficulty of representing these intermediate confidences values during generation.

**Definition 8** (Intermediate $B$-Confidences). *For a given function $B : \mathcal{V}^* \times \mathcal{V}^N \to [K]$ and model $p_\theta$, we define the* intermediate $B$-confidences *as the scalar-valued functions* $\{g_i\}_{i \in \{0,1,\dots,N\}}$:

$$g_i(z_{\leq i}; x) := \Pr_{z \sim p_\theta(\cdot | x, z_{\leq i})} [B_x(z) = k^*] \text{ where } k^* \leftarrow \operatorname*{argmax}_{k \in [K]} (B_x \sharp p_x)[k].$$

We will informally say that the LLM "knows" its intermediate $B$-confidences if the functions $g_i$ have a simple representation (e.g. each $g_i$ is computable by a small circuit on top of the LLM). In that case, we show in Thm. 9 that for any perturbation $w \in \mathcal{W}_B$, the perturbed model $p_\theta \star w$ has an only-slightly-more-complex representation than the original model $p_\theta$. Specifically, the perturbed model can be computed by composing a circuit $C_w$ with the functions $g_i$. Explicit formulas are provided in App. E.6.3.

**Theorem 9.** *For all functions $B : \mathcal{V}^* \times \mathcal{V}^N \to [K]$ and all perturbations $w \in \mathcal{W}_B$, there exists a small circuit[3] $C_w$ such that for all models $p_\theta : \mathcal{V}^* \to \Delta(\mathcal{V}^N)$, all $x \in \mathcal{V}^*, z \in \mathcal{V}^N$, all $i \in [N]$, and with $p_x := p_\theta(\cdot \mid x), w_x := w(x, p_x)$, the perturbed model $x \mapsto p_x \star w_x$ satisfies*

$$(p_x \star w_x)(z_i \mid z_{<i}) \propto C_w(a, g_i(z_{\leq i}; x), g_0(x)) \tag{3}$$

*where the constant of proportionality is independent of $z_i$, $a := p_x(z_i \mid z_{<i})$ is the original next-token probabilities, and $g_0, g_i$ are the intermediate $B$-confidences of Definition 8.*

Putting all the theory together, the message is: if the LLM "knows" its intermediate $B$-confidences, then perturbations $\mathcal{W}_B$ are easy to implement, and we should expect emergent $B$-calibration.

## 4 EXPERIMENTAL PREDICTIONS: WHEN ARE LLMS CALIBRATED?

Our main empirical question is: *Under what conditions and for which functions $B$ should we expect a pretrained LLM to be $B$-confidence-calibrated?*

The theory of the previous section suggests an answer: we should expect emergent $B$-calibration when the autoregressive $B$-confidences (Definition 8) are easy for the LLM to learn. We can simplify this into an experimentally-testable heuristic: for a given question $x$, is it easy for the LLM to predict (i.e. does it "know") the distribution $B_x \sharp p_x$ of its answers post-processed by $B$? Practically, we can operationalize "easy for the LLM to predict" by training a small LoRA on top of the base LLM to predict the $B$-class of the answer.

**Claim 10** (Main, heuristic). *Let $(x, y) \sim \mathcal{D}$ be a distribution on question-answer pairs, let $B : \mathcal{V}^* \times \mathcal{V}^N \to [K]$ be a collapsing function, and let $p_\theta(z \mid x)$ be an autoregressive language model*

---

[3]Specifically, an arithmetic circuit of constant depth and $\Theta(K)$ width.

*trained on $\mathcal{D}$ with cross-entropy loss. Then, $p_\theta$ will be $B$-confidence-calibrated on $\mathcal{D}$ if the function $G : \mathcal{V}^* \to \Delta_K$ defined as*

$$G : x \mapsto B_x \sharp p_x \quad \text{is "easy to learn" for the LLM (e.g. with a LoRA adapter)}$$

*In words: the LLM should be able to accurately estimate the distribution over semantic labels $B_x(z)$, under its own generative process, given the question $x$.*

Finally, we specialize Claim 10 to the practical case of semantic calibration—that is, we let $B$ be a function that collapses long-form answers into semantic equivalence classes, yielding the following:

**Corollary 11** (Main, heuristic). *LLMs trained autoregressively with cross-entropy loss will be semantically calibrated on in-distribution data if: the model can easily predict its own output distribution over semantic answers, given only the question.*

Corollary 11 leads to the following predictions, which we verify experimentally in Sec. 5.

**Prediction 1: Semantic calibration emerges from standard pretraining.** When $B$ is a semantic-collapsing function, we expect it to be easy-to-predict in many settings: Claim 10 only requires that the LLM intuitively "knows" what types of semantic-answers it is likely to output for a given question. Thus, we *should* expect emergent semantic calibration for a large class of pretrained LLMs, a remarkable fact not previously understood.

**Prediction 2: Instruction-tuning can break calibration.** We only theoretically predict calibration in models trained autoregressively with cross-entropy loss, that is, standard pretraining or SFT. (Cross-entropy loss is required to connect calibration with local-loss-optimality in Thm. 6.) We have no reason to expect calibration in models trained in other ways, including Instruct models post-trained with RLHF, DPO, or RLVR – although our theory does not preclude it.

**Prediction 3: Chain-of-thought reasoning (CoT) can break calibration.** To satisfy the conditions of our theory, the distribution over semantic classes must be easy for the model to estimate, even before generating the first token. In hard CoT setting such as math problem-solving, the model usually *does not* know what its answer will be until it has finished "thinking". Therefore, CoT is expected to break our mechanism for calibration. Notably, what makes CoT powerful (allowing the model to leverage more compute to produce a better answer than it could have produced immediately) is exactly what makes our mechanism of calibration fail.

## 5 EXPERIMENTS

In this section, we experimentally test the predictions of our theory on real models and datasets. All of our experiments include 5-shot examples in the prompt, and use temperature $T = 1$ sampling. We compare three different prompts, designed to elicit different styles of responses from the model: "concise" (answer in a single word/phrase), "sentence" (answer in a complete sentence), and "chain-of-thought (cot)". The few-shot examples are formatted in the desired style (e.g. for the "sentence" type, the few-shot examples have complete sentence answers). To measure calibration error, we use the SmoothECE metric introduced by Błasiok & Nakkiran (2024). For lack of space, full experimental details are in App. D.

### 5.1 EXPERIMENTAL RESULTS

We evaluate semantic calibration of Qwen, Gemini, Mistral, and Llama-family models, of varying sizes from 0.5B to 72B, for base and instruct variants, using each of the 3 response styles, on 6 open-ended question-answer datasets: GSM8K (Cobbe et al., 2021), OpenMathInstruct-2 (Toshniwal et al., 2025), TriviaQA (Joshi et al., 2017), SimpleQA (Wei et al., 2024), MATH500 (Lightman et al., 2023), and TruthfulQA (Lin et al., 2022b). This yields over 650 evaluation experiments, which we compile into Fig. 4 by overlaying their reliability diagrams. The box-plots in the bottom row of Fig. 4 show the distribution of calibration errors in aggregate for each dataset and configuration. We will use this condensed figure to discuss our experimental predictions. We expect our theory to apply on all datasets except, notably, TruthfulQA: This dataset contains common human misconceptions, and thus violates our in-distribution assumptions (see Remark 12). The full list of models is in App. D.3 and disaggregated results are reported in App. F.

**Prediction 1: Semantic calibration emerges from standard pretraining.** Our theory predicts that base models, in non-CoT settings, should be semantically calibrated. The top row of Fig. 4

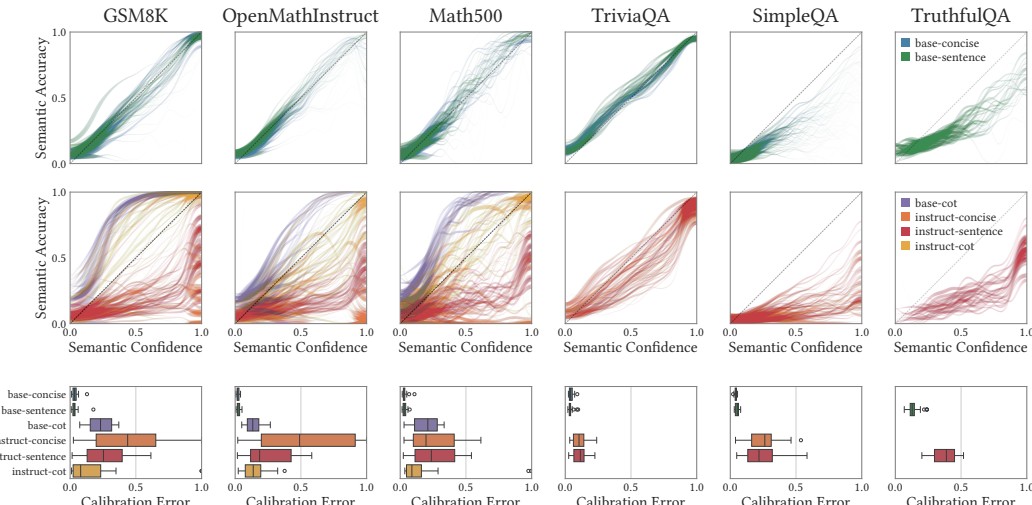

Figure 4: **Semantic Calibration of LLMs.** We evaluate Qwen, Gemini, Mistral, and Llama-family models, with 6 configurations for each model: (model-variant, response-style) ∈ {Base, Instruct} × {Concise, Sentence, CoT}. **First row (predicted calibrated):** Reliability diagrams of all configurations predicted to be confidence-calibrated according to our theory: base models with concise or sentence response types. TruthfulQA, a dataset of common misconceptions, is the exception: it violates the in-distribution assumptions of our theory, and is poorly calibrated. **Second row (not predicted calibrated):** Configurations which need not be calibrated according to our theory: post-trained instruct models with any response type: concise, sentence, chain-of-thought; and base models with chain-of-thought. **Third row:** Box plots summarizing the distribution of calibration errors for each of the 6 configurations. Only the first two configurations (base-concise and base-sentence) are reliably well-calibrated, as predicted by our theory. Note, we only consider chain-of-thought for the math datasets. TruthfulQA reference answers are available only in the sentence form, so we only report results for sentence response-style.

shows reliability diagrams for all such models we evaluated (configurations base-concise and base-sentence), and we observe nearly all of these experiments are well-calibrated. Notably, semantic calibration does not depend significantly on model size for base models: even small models (≤ 1B) are remarkably calibrated; see App. C for a more in-depth look at this aspect. Models are also well-calibrated regardless of the response style ("sentence" vs. "concise"), supporting our theory that semantic calibration depends not on the specific phrasing of the answer, but rather on whether the model "knows" its semantic class distribution before starting to generate.

**Prediction 2: Post-training can break calibration.** The middle row of Fig. 4 includes reliability diagrams for instruct post-trained models, for all three response types. Many of these settings are miscalibrated, typically overconfident (i.e. a curve below the diagonal), as expected from a reward-maximizing RL objective. Fig. 5 takes a closer look at the effect of different types of instruction-tuning on calibration. We compare three models from the same lineage: a base model (Mistral-7B-v0.1), a version of it post-trained via instruction supervised finetuning (SFT, zephyr-7b-sft-full), and a version post-trained via both SFT and Direct Preference Optimization (DPO, zephyr-7b-dpo-full) (Rafailov et al., 2024). The DPO model (not trained with a proper loss) is significantly miscalibrated, while the SFT-only model and the base model (both trained with proper losses) are better calibrated.

**Prediction 3: CoT reasoning can break calibration.** The middle row of Fig. 4 shows CoT with both base and instruct models, which are poorly calibrated in the math settings (GSM8K, OpenMath-Instruct, MATH500). Base-cot responses are underconfident (above the diagonal), while instruct-cot are underconfident for GSM8K, but overconfident for OpenMathInstruct, see Fig. 9. Notably, this miscalibration is not inherent to math: base models are calibrated when asked to provide the answer immediately (base-concise and base-sentence), but become miscalibrated when allowed to reason.

Figure 5: Calibration error for three models based on Mistral-7B-v0.1: pretrained-only, instruction-supervised-finetuned, and DPO-finetuned. Here, "sentence" response style, see Fig. 8 for others.

**Quantitative Learnability Probe.** Claim 10 suggests an explicit experiment to predict when a base model will be $B$-confidence-calibrated for a given choice of $B$: can the model "easily learn" the function $G : x \mapsto B_x \sharp p_x$ mapping a question $x$ to the distribution over the model's own semantic answers for that question? We can test this by training a small LoRA (Hu et al., 2022) on top of the model, to directly generate the semantic class distribution $B_x \sharp p_x$ when prompted with the question $x$. For example, in CoT settings, this would require the LoRA to "short-circuit" the reasoning steps, and immediately generate the final answer that the model would have

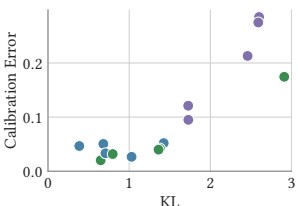

Figure 6: Testing Claim 10 across Qwen2.5 models.

produced with CoT. Notably, this does not require the model to produce the *correct* semantic answer, but just match its own generative distribution. In Fig. 6, we train rank-8 LoRAs on Qwen2.5 models of varying sizes, on GSM8K.

We then compare each LoRA's KL gap to optimality (x-axis) to the underlying model's calibration error (y-axis). The correlation agrees with our theory: models which can easily predict their own semantic class distribution (low KL gap) are also well-calibrated. Full details in App. D.2.

**Discussion: Calibration in LLMs vs other deep networks.** One may wonder why the state of calibration in LLMs seems significantly different from calibration in non-LLM deep networks (e.g. image classifiers). Specifically, deep network classifiers are sometimes severely overconfident, and sometimes well-calibrated, depending on the specific network (e.g. Guo et al. (2017) vs. Minderer et al. (2021)). On the other hand, all base LLMs we tested were well-calibrated (in non-CoT settings) — there was no significant calibration difference between models. This difference between LLMs and classifiers is due to differences in training practices: when training LLMs, practitioners monitor the test/validation loss closely, and stop training before the test loss overfits (increases). On the other hand, when training classifiers, practitioners care about the test *classification error*, and often continue training even as test *loss* increases. Most trained LLMs are therefore locally-loss-optimal w.r.t. test loss (and thus calibrated), but trained classifiers might have high test loss (and thus be miscalibrated). This perspective on the calibration of deep networks was articulated in Section 1.1 of Błasiok et al. (2023).

## 6 CONCLUSION

We find that base LLMs, despite being trained with a token-level syntactic objective, are remarkably calibrated with respect to the *sequence-level semantics* of their generations. Our central contribution is a principled mechanism behind this emergence, building on recent theoretical connections between calibration and loss-optimality (Błasiok et al., 2023; 2024). This theory provides a unified lens through which to understand the nuanced calibration behavior of models in practice, distinguishing settings which are calibrated from those which are not. Among limitations, we only propose one possible mechanism for calibration; it is possible that other types of calibration (e.g. verbalized calibration) emerge for yet-undiscovered reasons; other limitations discussed in Appendix B.1. More generally, our work can be seen as a step towards understanding the formal structure of LLMs' output distribution.

### ACKNOWLEDGMENTS

We are grateful for discussion and feedback from Parikshit Gopalan, Eran Malach, Aravind Gollakota, Omid Saremi, Madhu Advani, Etai Littwin, Josh Susskind, and Russ Webb.

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

APPENDIX CONTENTS

## A    ADDITIONAL RELATED WORKS

**Recalibration Methods.**    A number of prior works study methods to improve the calibration of LLMs, ranging from temperature-scaling at inference-time (e.g., Xie et al., 2024; Shen et al., 2024) to training calibration-specific probes that predict correctness (Mielke et al., 2022) or training with calibration-improving regularization terms (Wang et al., 2025). Other approaches attempt to cluster questions and predict per-cluster accuracy (Lin et al., 2022a; Ulmer et al., 2024), or make use of the fact that ensembling models tends to improve calibration (Jiang et al., 2023b; Hou et al., 2024). Probabilistic approaches (such as Bayesian deep learning, or evidential deep learning) have been found to often yield better calibration (e.g., Li et al., 2025; Yang et al., 2024a).

**Sampling-based Confidences.**    A number of prior works have proposed sampling-based approaches to defining LLM uncertainty. Both Wang et al. (2023) and Wei et al. (2024) sample multiple answers per-question, and define confidence as the frequency of the most-common answer. Wei et al. (2024) additionally groups answers together by string-matching, which allows for some degree of semantic equivalence. This approach was extended and popularized by the notion of *semantic entropy* (Farquhar et al., 2024). Semantic entropy clusters sampled answers together by semantic content, and then measures the empirical entropy of clustered answers. Recently, Lamb et al. (2025) define Empirical Semantic Confidence, which is essentially an empirical version of our notion of semantic confidence. Note that one distinguishing aspect of our formalism is, we parameterize the notion of calibration by the choice of collapsing function $B$. This allows us to develop somewhat more general theoretical insights, which are not tied to a fixed notion of semantics.

**Factors which Harm LLM Calibration.**    Various factors have been observed in prior work to harm LLM calibration. It is well-known that RLHF often harms calibration in multiple-choice QA settings (Kadavath et al., 2022; OpenAI, 2023). Other RL post-training methods such as DPO have also been observed to harm calibration (Leng et al., 2025; Xiao et al., 2025). Some studies have also found chain-of-thought responses to harm calibration, agreeing with our results (Zhang et al., 2024). However, we warn that not all of these works use the same notion of confidence and calibration as we do, and so are not directly comparable.

**Related Technical Tools from Prior Work.**    The connection between local-loss-optimality and calibration was theoretically studied in (Błasiok et al., 2023), which proved a version of Thm. 6 for binary classifiers, and was our inspiration for our theoretical results. Moving from binary classifiers to LLMs posed three main technical challenges. First, in binary classifiers there is essentially only one canonical notion of calibration, and so Błasiok et al. (2023) only required one notion of local-loss-optimality. However in our LLM setting, there are many notions of calibration (parameterized by functions $B$), and so we needed to identify the "right" notion of local-loss-optimality that is also parameterized by $B$. To do this we observed that $B$-calibration can be written as a type of "weighted calibration," a notion introduced in Gopalan et al. (2024). Second, we needed to generalize the 1-dimensional results of Błasiok et al. (2023) to higher dimensions, to handle multi-class settings. This turned out to be a straightforward though somewhat technical generalization, using the Savage representation of proper losses (Savage, 1971). Third, and most significantly: unlike classifiers, LLMs do not output their predicted probabilities explicitly. Rather, they implicitly define a probability distribution via their next-token predictions. This difference between implicit and explicit probability distributions required a number of conceptual adaptations to the theory of Błasiok et al. (2023), which guided our definitions of the perturbation operator and $\mathcal{W}$-local-loss-optimality (Definition 5).

## B    EXTENDED DISCUSSION AND REMARKS

### B.1    LIMITATIONS

**Types of Calibration.**    One limitation of our paper is that we focus on a very specific type of calibration, which is essentially a sampling-based notion ($B$-confidence-calibration). It is possible that other types of calibration (e.g. verbalized calibration) also emerge for certain types of LLM training; we consider this possibility interesting but out-of-scope for the current work.

**Practical Implications.** Our work is primarily scientifically motivated, and so we do not fully explore practical considerations or implications. For example, we do not consider the computational efficiency of our confidence measurements. This is a limitation to using such measures in practice, since computing semantic confidence requires sampling an LLM multiple times for the same question. We consider translating our scientific results into real-world improvements to be an important direction for future work.

**Datasets.** Although we evaluate on a variety of different models, we only evaluate on 6 selected datasets. We chose these datasets to cover a diversity of domains and problem difficulties, from questions about world-knowledge to mathematical reasoning problems. Further, we chose datasets with *open-ended* answers, since calibration of multiple-choice datasets is already extensively studied (Kadavath et al., 2022; Zhu et al., 2023). Although we do not expect our results to depend significantly on the choice of dataset, it is possible that certain other datasets have different calibration behavior; this is a limitation of our experiments.

**Remark 12.** *Notably, there are some datasets which we would expect to behave differently, such as TruthfulQA (Lin et al., 2022b), which is a dataset containing common human misconceptions. This dataset fails to satisfy the "in-distribution" requirement of our results (e.g. Claim 10), and so it is consistent with our theory for models to be miscalibrated.*

## B.2 POTENTIAL EXTENSIONS

The theoretical framework described here is fairly general, and extends beyond the setting of confidence-calibration in LLMs. Briefly, since most of our theory is stated in the language of *weighted calibration* (Gopalan et al., 2024), it applies to any property that can be written as weighted calibration. This includes slightly stronger notions of calibration, such as top-label calibration, and also includes conformal-prediction type of guarantees (more details in App. E.8.1. See Gopalan et al. (2024) for a number of properties which can be expressed as weighted calibration, and App. E.8 for the connection to conformal prediction. Our general theoretical results appear in App. E.

Intuitively, the high-level message of our results is that if a model is trained with a max-likelihood / log-loss objective, then we should expect it to satisfy weighted calibration for a "simple" family of weight functions. The appropriate notion of simplicity depends on the model architecture; simple weight functions should roughly correspond to easy-to-learn perturbations to the model's output distribution. At this level of generality, we expect some version of our results to apply even for real-valued density models, such as continuous normalizing flows (e.g. Zhai et al. (2025)), which are also trained with the log-likelihood objective. That is, we should expect such normalizing flows to also exhibit certain (weak) types of calibration. We believe this is a promising avenue for future work.

## B.3 TECHNICAL REMARKS

We collect several technical remarks regarding the theory of Sec. 3.

**Remark 13** (Heuristic Simplifications). *In translating the theoretical results of Sec. 3 to the practical heuristic of Claim 10, we took several steps which we describe more explicitly here. First, Thm. 9 is about ease of* representation, *but in Claim 10 we chose to use ease of* learning. *This is both more practical (since learning can be directly tested) and, we believe, more natural (since then both the premise and conclusion of Claim 10 involve the learning procedure of the LLM).*

*Now, Thm. 9 suggests that for $B$-confidence-calibration, it is sufficient for the functions $\{g_i\}$ of Definition 8 to be "easy to learn" for the LLM, for all prefix lengths $i \in [N]$. Claim 10 deviates from this in two ways. First, instead of considering all prefix lengths $i$, we only consider the empty prefix ($i = 0$) i.e. the model's distribution given only the question. Intuitively, the prediction from the empty prefix is likely the most challenging, and practically, this simplification means that only one simple-to-implement probe is required. Second, instead of considering learnability of only the semantic* confidence *function ($g_0$), Claim 10 considers learnability of the entire semantic distribution ($B_x \sharp p_x$). Practically, this improves robustness of the empirical estimator, since the KL divergence can be estimated from samples. Empirically, we did not find these simplifications to significantly affect the conclusions.*

**Remark 14** (Multicalibration). *One detail of the theory worth discussing further is the role of the distribution $\mathcal{D}$. For clarity of exposition, we described the theory as if there is only one distribution $\mathcal{D}$*

*of interest, but in reality, we evaluate calibration across multiple distributions (TriviaQA, GSM8K, etc), and we pretrain on yet another distribution. Moreover, we find that a single model can be simultaneously calibrated across many evaluation distributions.*

*Formally, requiring $B$-calibration across multiple distributions simultaneously can be thought of as a* multi-calibration *property (Hébert-Johnson et al., 2018). Suppose for example that the pretraining distribution $\mathcal{D}$ is some mixture of disjoint sub-distributions: $\mathcal{D} = \alpha_1 D_1 + \alpha_2 D_2 + \ldots$. Suppose we are interested in $B$-calibration simultaneously for distributions $D_1$ and $D_2$. Then, it is possible to show a generalization of Thm. 6:*

> *A model is $B$-confidence-calibrated across both $D_1$ and $D_2$ if and only if it is locally-loss-optimal on $\mathcal{D}$ w.r.t. an expanded class of perturbations $\mathcal{W}_B^*$.*

*Informally, the class of perturbations $\mathcal{W}_B^*$ is essentially the usual class $\mathcal{W}_B$ (of Definition 4) augmented by indicator functions $\mathbb{1}\{x \in \mathcal{D}_1\}$, $\mathbb{1}\{x \in \mathcal{D}_2\}$ for membership in each sub-distribution.*

*We will not get into the technical details, but using this version of Thm. 6, it is possible to carry out the remaining steps of the argument from Sec. 3 and Fig. 3. Applying the same heuristics, for example, we would conclude: an LLM will be simultaneously $B$-confidence-calibrated on distributions $\mathcal{D}_1, \mathcal{D}_2$ if it is easy for the LLM to (1) estimate its own distribution on $B$-classes and (2) identify samples as either $x \in \mathcal{D}_1$ or $x \in \mathcal{D}_2$.*

*The second condition is likely to be satisfied in all our experiments, since all our evaluation datasets are distinct and easy to identify. Thus, the predictions of our theory remain unchanged, justifying our choice to avoid discussing multicalibration in the main body.*

**Remark 15** (Full calibration). *At first glance, it may seem that a minor generalization of our mechanism (Fig. 3) would also imply* full *$B$-calibration (i.e., canonical calibration of the $B$-induced classifier), rather than just* confidence*-calibration. After all, Thm. 6 formally generalizes to arbitrary weight families $\mathcal{W}$ (see Thm. 27), including the family corresponding to full $B$-calibration (defined as $\mathcal{W}_B^{(\text{full})}$ in Definition 20). However, full $B$-calibration is too strong a property to hold in general[4]. So, which part of our argument in Fig. 3 breaks for full calibration? The culprit is the heuristic step in Fig. 3. The weight family $\mathcal{W}_B^{(\text{full})}$ relevant for full calibration is, roughly speaking, "too large" for the same heuristic to hold.*

*To better understand why the heuristic fails, here is more general version of the heuristic step in Fig. 3, which we believe is plausible for arbitrary weight families $\mathcal{W}$.*

**Claim 16** (heuristic, informal). *If a perturbation family $\mathcal{W}$ is easy-to-learn for a pretrained LLM, meaning: for all perturbations $w \in \mathcal{W}$, the LLM $p_\theta : \mathcal{V}^* \to \Delta(\mathcal{V}^N)$ can be easily LoRA-fine-tuned to match the distribution of a perturbed-model $G : \mathcal{V}^* \to \Delta(\mathcal{V}^N)$,*

$$G : x \mapsto p_x \star w_x \equiv p_\theta(\cdot \mid x) \star w(x, p_x) \tag{4}$$

*then $p_\theta$ will be $\mathcal{W}$-locally-loss-optimal w.r.t. its pretraining loss.*

*In other words, if all perturbations in the family $\mathcal{W}$ can be "easily learnt," then we should expect the LLM to be loss-optimal w.r.t. $\mathcal{W}$. Claim 10 is essentially a special case of this more general claim, for the specific class $\mathcal{W}_B$ relevant to $B$-confidence-calibration.*

*If we believe Claim 16, we can see why our mechanism would apply to confidence-calibration but not to full-calibration: For confidence-calibration, the perturbation class $\mathcal{W}_B$ (Definition 4) is simple enough to be learnable, while for full calibration, the corresponding perturbation class $\mathcal{W}_B^{(\text{full})}$ (Definition 20) is too large to be efficiently learnable from samples. To gain intuition for this, it helps to directly compare Definition 20 to Definition 4. From this discussion, we can see it is likely possible to extend our results to certain types of calibration which are weaker than full-calibration, but stronger than confidence-calibration. We leave this direction for future work.*

---

[4]For example, when $K$ (the number of $B$-classes) is large, full $B$-calibration would be computationally intractable to even estimate (Gopalan et al., 2024).

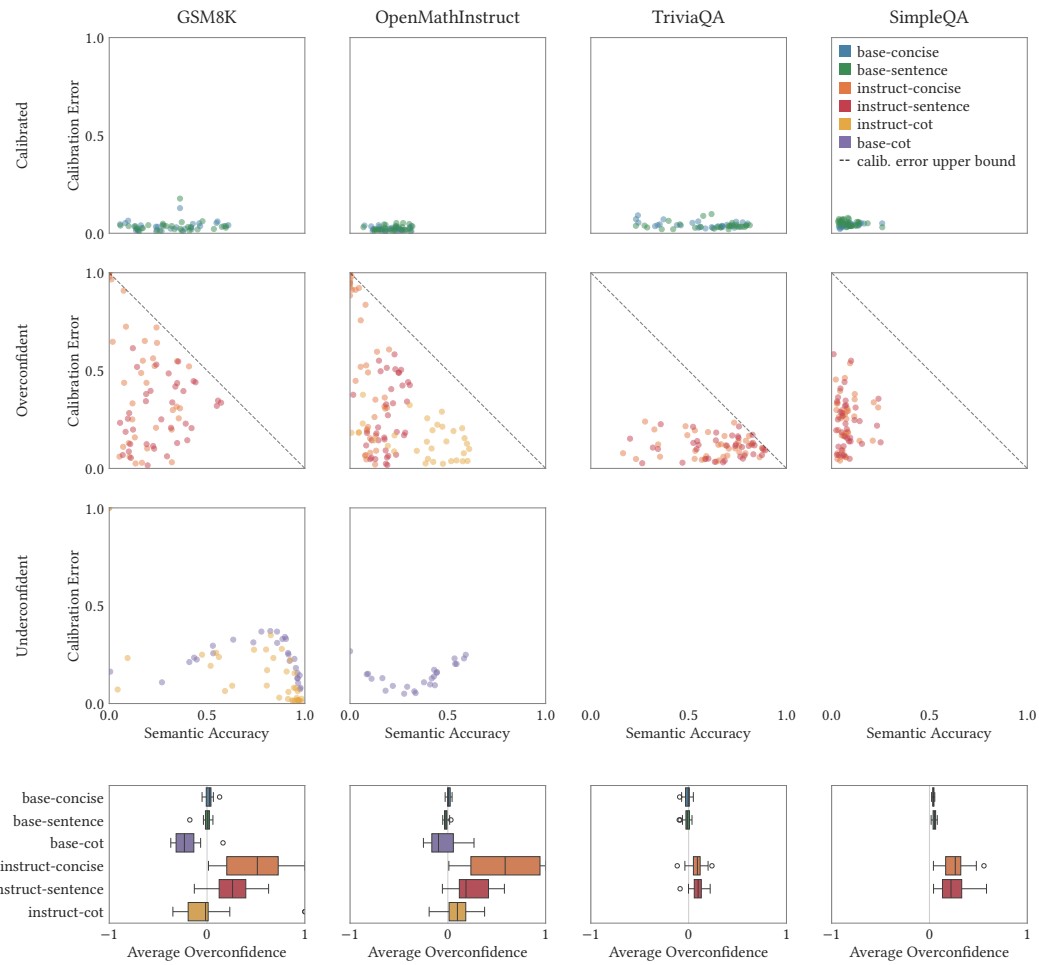

Figure 7: **Effect of Scale:** We plot Calibration Error vs. Semantic Accuracy for all models in Fig. 4; each dot represents a separate model. **First row (predicted calibrated):** In the settings our theory applies, we see no correlation between the model capability (semantic accuracy) and the calibration error. **Second row (overconfident):** Configurations which we empirically observed to be mostly overconfident. The dashed line illustrates the upper bound on the calibration error w.r.t. the accuracy, for a maximally overconfident predictor. We see little correlation between semantic accuracy and the calibration error beyond what is dictated by the upper bound. **Third row (underconfident):** We see little correlation between calibration and accuracy, except near the extreme when models approach perfect accuracy. TriviaQA and SimpleQA plots are empty because there are no underconfident configurations. **Fourth row:** The distribution of *average overconfidence* across models, for each configuration; positive/negative values indicate over–/underconfidence.

## C   ADDITIONAL EXPERIMENTAL RESULTS

Due to their volume, disaggregated reliability diagram results are reported separately in App. F.

**Model Scaling Effects.** Here, we aim to explore the effect of model scaling (parameter count, compute, data) on calibration. Since information about training details are most often not publicly available, we use the model capability (measured with accuracy) as a proxy variable for model scale. In Fig. 7, we plot calibration error (smECE) vs. semantic accuracy. For base models without chain-of-thought (first row), we see no correlation between model capability (semantic accuracy) and calibration error. This is consistent with our theoretical predictions, which have no explicit dependency on model scale or capability. It is worth noting that prior works have observed that calibration of base models can improve with model scale for other notion of calibration: next-token

prediction in multiple-choice question setups (Kadavath et al., 2022; Zhu et al., 2023; Plaut et al., 2025). We do not find such improvements for semantic calibration of *base models*.

For instruct models and base models with chain-of-thought, we empirically observe that some configurations are overconfident, while other underconfident, and we divide those configurations into separate rows in Fig. 7. The dashed line illustrates the upper bound on the calibration error w.r.t. the accuracy for an overconfident configurations, which is dictated by the behavior of a maximally-overconfident predictor that puts its entire probability mass on a single choice. For the overconfident configurations, we see little correlation between calibration error and accuracy beyond what is dictated by the upper bound. For the underconfident configurations, we see also little correlation overall, except for in the high-accuracy regime: calibration error tends to decrease when models approach perfect semantic accuracy[5]. However, it is not clear whether this is a robust phenomenon.

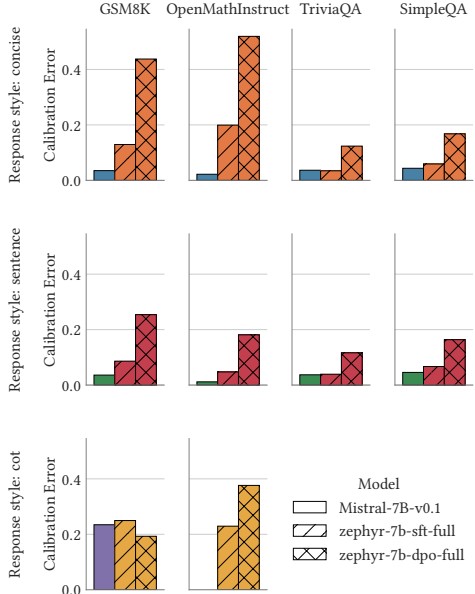

Figure 8: Calibration error for three models based on Mistral-7B- v0.1: pretrained-only, instruction-SFT model (zephyr-7b-sft-full), DPO model (zephyr-7b-dpo-full). We did not evaluate TriviaQA and SimpleQA for the "cot" response style. The "cot" result for Mistral-7B-v0.1 for OpenMathInstruct is missing due to the model not terminating generation within its maximum context length.

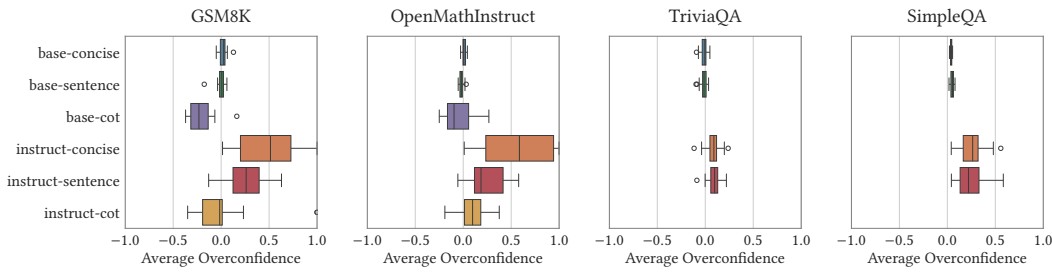

Figure 9: Distribution of average overconfidence (or "mean non-absolute calibration error") for data in Fig. 4. Positive values indicate overconfidence, negative values indicate underconfidence.

---

[5]Notably, this is not mathematically necessary. Since semantic accuracy involves only the *argmax*-probability class, it is possible for a predictor to be perfectly semantically-accurate while having high calibration error.

# D    ADDITIONAL EXPERIMENTAL DETAILS

**Datasets.**    We focus on open-ended question-answer (QA) settings, since calibration for multiple-choice QA is already well-studied (Kadavath et al., 2022; Zhu et al., 2023), and a special case of our results. We evaluate on: GSM8K (Cobbe et al., 2021), OpenMathInstruct-2 (Toshniwal et al., 2025), TriviaQA (Joshi et al., 2017), and SimpleQA (Wei et al., 2024), from Huggingface datasets (Wolf et al., 2019; Lhoest et al., 2021).

**Models.**    We evaluate on models including the Qwen, Gemini, Mistral, and Llama family, of sizes from 0.5B to 72B. The full list of models we evaluate is in App. D.3. We use vLLM (Kwon et al., 2023) for inference.

**Prompt format.**    See App. D.4 for the exact phrasing used in prompts. All of our experiments include 5-shot examples in the prompt. We use three different prompt types, designed to elicit three different styles of responses from the model: "sentence", "concise", and "chain-of-thought (cot)". The few-shot examples are formatted in the desired style (e.g. for the "sentence" type, the few-shot examples have complete-sentence answers). For instruct models, in addition to formatted few-shot examples, the prompt also includes explicit formatting instructions. The "concise" prompt type encourages the model to respond with just the final answer (a single word, phrase, or number). The "sentence" prompt type asks the model to answer each question in a complete sentence (making it likely to phrase the same semantic answer in different ways, so the $B$-collapsing function is essential for a meaningful notion of semantic calibration). The "cot" prompt type elicits chain-of-thought reasoning from the model; this prompt type is only used for math datasets.

These prompts are typically successful in eliciting the desired type of responses from the model. However, in some cases we observed models (especially Qwen models) produce "chain-of-thought" responses even when prompted to reply in a single word. To exclude such cases, we exclude any responses for the "concise" prompt on math datasets which are too long (heuristically, more than 15 characters before the first newline).

**The semantic collapsing function.**    Recall, the function $B$ is intended to collapse semantically-equivalent generations into a single class, an idea proposed by Kuhn et al. (2023). We implement the function $B$ with a two-stage procedure as follows.

The first stage is canonicalization: we extract a short "canonical form" answer from the LLM's response. For "concise" and "cot" prompt types, this is done via simple string parsing (for "cot", extracting only the final answer). For the "sentence" type, we use a strong LLM (Qwen3-14B-Instruct) prompted to extract a short-answer from the generation, given the question as context. The prompts used for canonicalization are in App. D.4: Prompt 4 for non-math settings, and Prompt 5 for math settings. We also normalize strings at this stage, converting to lower-case and stripping spaces, including a math-specific normalization for domains with LaTeX outputs. Specifically, we use the MATH string-normalization from Minerva, given in Listing 1, Appendix D.1 of Lewkowycz et al. (2022).

The second stage, used only for non-math settings, is semantic clustering: we prompt an LLM judge (Qwen3-14B-Instruct) to assess whether two responses to a question are semantically equivalent, and use the output to cluster responses[6]. This is necessary for non-math settings to handle irrelevant differences in canonical forms (e.g. "Seattle, WA" vs "Seattle"). The prompt used for semantic equivalence is Prompt 6 in App. D.4. For math settings, the second stage is unnecessary, since the first stage already outputs a number or symbol that can be directly compared.

**Measuring calibration.**    We first produce an LLM-induced semantic classifier, following the experimental procedure described in Sec. 2 and illustrated in Fig. 1. For each dataset, we take 10K random evaluation samples (or the entire dataset for those with fewer than 10K total samples). For each question, we construct the appropriate 5-shot prompt, sample $M = 50$ responses from the LLM at temperature 1, and then apply the semantic collapsing function (described above) to each response. The semantic confidence is defined as the empirical frequency of the plurality semantic class, and the semantic accuracy is the $0/1$ indicator of whether this plurality class matches the ground-truth's semantic class. This yields, for each question, a pair of (semantic-confidence, semantic-accuracy)

---

[6]This is a slight variation of the two-way entailment method used by Farquhar et al. (2024).

$\in [0, 1] \times \{0, 1\}$. We then evaluate the calibration of the resulting classifier over the entire dataset of questions using SmoothECE (smECE, Błasiok & Nakkiran (2024)), a theoretically-principled version of the Expected Calibration Error (ECE). We use the SmoothECE implementation provided by: `https://github.com/apple/ml-calibration`.

## D.1   VISUALIZING CALIBRATION: RELIABILITY DIAGRAMS

We follow the guidance of Błasiok & Nakkiran (2024), and visualize calibration using kernel-smoothed reliability diagrams.

**Reading the Diagram.** Fig. 2 gives several examples of reliability diagrams. The solid red line is the regression line, an estimate of $\mu(c) := \mathbb{E}[\text{semantic accuracy} \mid \text{semantic confidence} = c]$. The black cross is the point $(\mathbb{E}[\text{semantic confidence}], \mathbb{E}[\text{semantic accuracy}]) \in [0, 1] \times [0, 1]$, that is, the average semantic confidence and accuracy. The gray histograms at the bottom of the plot visualize the density of semantic confidences. We plot two overlaid histograms, one for the confidence distribution of correct predictions (i.e. the confidence of samples where semantic-accuracy=1), and another for the confidence distribution of incorrect predictions. The width of the red regression line varies with the overall density of semantic-confidences.

**Implementation Details.** For reliability diagrams, we use the implementation of `relplot` (`https://github.com/apple/ml-calibration`) with minor modifications: we use a fixed kernel bandwidth $\sigma = 0.05$ for the regression line, and we visualize the density of confidences using histogram binning with 15 constant-width bins.

To compute the scalar SmoothECE (smECE) metric, we use the original implementation of `relplot` without modification (including its automatic choice of bandwidth).

## D.2   LoRA FINE-TUNING

To test Claim 10 more quantitatively, we train a LoRA version of the LLM to explicitly learn the function $G$ defined in Claim 10. We do this as follows. Let $p_\theta$ be the base model. Instantiate a rank=8 LoRA adapter (Hu et al., 2022) on top of the original model $p_\theta$, which we denote $p_\phi$.

We want to train $p_\phi$ to behave as the "semantically-collapsed" version of $p_\theta$. That is, when prompted with a question $x$, the model $p_\phi$ should generate a distribution on answers $b$ which imitates the base model's semantic answers $B_x(z)$:

$$p_\phi(b \mid x) \approx \Pr_{z \sim p_\theta(\cdot \mid x)}[B_x(z) = b] \equiv (B_x \sharp p_x)(b) \qquad (5)$$

Since our implementation of the collapsing function $B$ produces string outputs (canonical answers), we can train $p_\phi$ as a standard autoregressive model. Explicitly:

1. For each question in the dataset $x$, sample the original model 50 times, and apply the collapsing function $B$ to each generation. This produces 50 samples $\{(x, b_i)\}$ of question $x$ and canonical-answer $b_i$ for each original question $x$, effectively expanding the original dataset size by 50 times.

2. Train $p_\phi$ with the standard autoregressive objective, on the prompt-completion pairs $\{(x, b_i)\}$ from above. That is, train $p_\phi$ to complete prompt $x$ with generation $b_i$.

Our training procedure is similar to the procedure used to train "P(IK)" in Kadavath et al. (2022), in that we also train on an "expanded" training set defined by base model samples. Similar to Kadavath et al. (2022), we do this mainly for convenience.

For GSM8K, we hold-out 2000 questions for evaluation, and use the remainder for training as above. We train all models on an 8xA100 node for 1 epoch on the expanded dataset, using the `SFTTrainer` implementation from Huggingface TRL (von Werra et al., 2020) with the following parameters in Table 1. Note, we shuffle the expanded training set manually beforehand, so we do not ask the dataloader to shuffle.

Table 1: Hyperparameters for Supervised Fine-Tuning (SFT).

| Parameter | Value |
|---|---|
| *Training & Hardware* | |
| num_train_epochs | 1 |
| per_device_train_batch_size | 4 |
| gradient_accumulation_steps | 2 |
| (Effective Batch Size) | 64 (4 x 8 GPUs x 2) |
| bf16 | True |
| *Optimizer & Scheduler* | |
| optim | adamw_torch_fused |
| learning_rate | 5e-5 |
| weight_decay | 0.0 |
| warmup_ratio | 0.05 |
| *PEFT (LoRA) Configuration* | |
| use_peft | True |
| lora_r | 8 |
| lora_alpha | 16 |
| lora_dropout | 0.0 |
| lora_target_modules | all-linear |
| task_type | CAUSAL_LM |
| bias | none |
| *Data Handling* | |
| packing | False |
| dataloader_shuffle | False |

After training, we evaluate how closely Eq. (5) holds, by estimating the KL divergence between RHS and LHS of Eq. (5). This KL measures how well our LoRA $p_\phi$ matches its training distribution. Conveniently, the KL can be written as the difference between the *negative-log-loss* of $p_\phi$ and the *semantic entropy* of the original model $p_\theta$:

$$\text{Gap to optimality} := KL(\ (B_x \sharp p_x)\ \|\ p_\phi(\cdot\mid x)\ ) \tag{6}$$

$$= \underbrace{\mathbb{E}_{\substack{x \sim \mathcal{D} \\ z \sim p_\theta(z|x)}} [-\log p_\phi(B(z)\mid x)]}_{\text{Eval NLL loss of } p_\phi} - \underbrace{H(B_x \sharp p_x)}_{\text{Semantic entropy of } p_\theta} \tag{7}$$

This is particularly convenient because the eval log-loss is a standard metric tracked during training. Note that for our purposes, it is important to compute the *unnormalized* log-loss (i.e., not normalized by sequence-length).

In **Fig. 6**, we plot the KL gap of Eq. (7) on the x-axis, and the SmoothECE of the original model $p_\theta$ on the y-axis. We evaluate base models: Qwen2.5-{0.5B, 1.5B, 3B, 7B, 14B}, with all three response styles: concise, sentence, cot. This results in 15 points plotted in Fig. 6, colored according to response style using the color scheme of Fig. 4. We observe that, consistent with Claim 10, configurations where the semantic class distribution is easy-to-learn (low KL gap) also have small calibration error. The points with high KL (and high calibration error) are the chain-of-thought experiments, as well as the small 0.5B model with the "sentence" response type.

### D.3 LLMs EVALUATED

Below, we list all models evaluated in this paper. All were obtained from HuggingFace.

Table 2: Pretrained-only base models evaluated in this paper. Models sharing a prefix and reference are grouped.

| Family Prefix | Model Suffix | Reference |
|---|---|---|
| google/ | `gemma-2-2b`
`gemma-2-9b`
`gemma-2-27b` | (Gemma Team et al., 2024) |
| | `gemma-3-1b-pt`
`gemma-3-4b-pt`
`gemma-3-12b-pt`
`gemma-3-27b-pt` | (Gemma Team et al., 2025) |
| Qwen/ | `Qwen2.5-0.5B`
`Qwen2.5-1.5B`
`Qwen2.5-3B`
`Qwen2.5-7B`
`Qwen2.5-14B`
`Qwen2.5-32B`
`Qwen2.5-72B` | (Yang et al., 2024c) |
| | `Qwen2.5-Math-1.5B`
`Qwen2.5-Math-7B`
`Qwen2.5-Math-72B` | (Yang et al., 2024b) |
| | `Qwen3-0.6B-Base`
`Qwen3-1.7B-Base`
`Qwen3-4B-Base`
`Qwen3-8B-Base`
`Qwen3-14B-Base` | (Yang et al., 2025) |
| mistralai/ | `Mistral-7B-v0.1`
`Mistral-7B-v0.3` | (Jiang et al., 2023a) |
| | `Mistral-Small-24B-Base-2501` | (Mistral AI Team, 2024b) |
| | `Mixtral-8x7B-v0.1` | (Mistral AI Team, 2023) |
| meta-llama/ | `Llama-3.1-8B`
`Llama-3.1-70B` | (Grattafiori et al., 2024) |

Table 3: Instruction-tuned models evaluated in this paper. Models sharing a prefix and reference are grouped.

| Family Prefix | Model Suffix | Reference |
|---|---|---|
| google/ | gemma-2-2b-it
gemma-2-9b-it
gemma-2-27b-it | (Gemma Team et al., 2024) |
| | gemma-3-1b-it
gemma-3-4b-it
gemma-3-12b-it
gemma-3-27b-it | (Gemma Team et al., 2025) |
| Qwen/ | Qwen2.5-0.5B-Instruct
Qwen2.5-1.5B-Instruct
Qwen2.5-3B-Instruct
Qwen2.5-7B-Instruct
Qwen2.5-14B-Instruct
Qwen2.5-32B-Instruct
Qwen2.5-72B-Instruct | (Yang et al., 2024c) |
| | Qwen2.5-Math-1.5B-Instruct
Qwen2.5-Math-7B-Instruct
Qwen2.5-Math-72B-Instruct | (Yang et al., 2024b) |
| | Qwen3-0.6B
Qwen3-1.7B
Qwen3-4B
Qwen3-8B
Qwen3-14B
Qwen3-32B | (Yang et al., 2025) |
| mistralai/ | Mistral-7B-Instruct-v0.1
Mistral-7B-Instruct-v0.3 | (Jiang et al., 2023a) |
| | Ministral-8B-Instruct-2410 | (Mistral AI Team, 2024a) |
| | Mistral-Small-24B-Instruct-2501 | (Mistral AI Team, 2024b) |
| NousResearch/ | Nous-Hermes-2-Mixtral-8x7B-SFT | (Nous Research, 2024b) |
| | Nous-Hermes-2-Mixtral-8x7B-DPO | (Nous Research, 2024a) |
| alignment-handbook/ | zephyr-7b-dpo-full
zephyr-7b-sft-full | (Tunstall et al., 2023) |
| meta-llama/ | Llama-3.1-8B-Instruct
Llama-3.1-70B-Instruct
Llama-3.3-70B-Instruct | (Grattafiori et al., 2024) |
| microsoft/ | phi-4 | (Abdin et al., 2024) |

## D.4 PROMPTS

We use 3 different prompt styles: concise, sentence, and chain-of-thought (cot). All prompts use 5 few-shot examples from the dataset. We describe the prompt formatting here by way of example, using our prompts for the GSM8K dataset. For base models, we use the full prompt text as context, while for instruct models we format the few-shot examples using the model-specific chat template (per Huggingface).

`Prompt 1` shows the *"concise"* prompt for GSM8K. This prompt style uses only the final answers provided by the dataset (excluding any chain-of-thought).

`Prompt 2` shows the *"sentence"* prompt type. This prompt formats the few-shot answers in complete sentences, and also includes instructions to format answers accordingly. Note that we intentionally varied the sentence structure of the few-shot examples, to encourage the model to use a diversity of phrasings. This makes the "sentence" responses more syntactically complex than the "concise" responses, though not more *semantically* complex — thus testing the limits of our theory.

`Prompt 3` shows the *"cot"* prompt type. This includes reasoning and formatting instructions, as well as few-shot examples that include reasoning-traces (provided by the dataset).

The prompt formatting for other datasets follow the same conventions as these GSM8K examples. We exclude the "cot" prompt type for non-math datasets.

---

**Prompt 1: GSM8K-concise**

```
Question: Natalia sold clips to 48 of her friends in April, and then she sold half as many clips in
May. How many clips did Natalia sell altogether in April and May?
Answer: 72

Question: Weng earns $12 an hour for babysitting. Yesterday, she just did 50 minutes of babysitting.
How much did she earn?
Answer: 10

Question: Betty is saving money for a new wallet which costs $100. Betty has only half of the money
she needs. Her parents decided to give her $15 for that purpose, and her grandparents twice as much
as her parents. How much more money does Betty need to buy the wallet?
Answer: 5

Question: Julie is reading a 120-page book. Yesterday, she was able to read 12 pages and today, she
read twice as many pages as yesterday. If she wants to read half of the remaining pages tomorrow, how
 many pages should she read?
Answer: 42

Question: James writes a 3-page letter to 2 different friends twice a week. How many pages does he
write a year?
Answer: 624

Question: {QUESTION}
Answer:
```

---

**Prompt 2: GSM8K-sentence**

```
Answer the following question in a single brief but complete sentence.
Question: Natalia sold clips to 48 of her friends in April, and then she sold half as many clips in
May. How many clips did Natalia sell altogether in April and May?
Answer: Natalia sold 72 clips in April and May combined.

Answer the following question in a single brief but complete sentence.
Question: Weng earns $12 an hour for babysitting. Yesterday, she just did 50 minutes of babysitting.
How much did she earn?
Answer: Weng earned only $10 yesterday.

Answer the following question in a single brief but complete sentence.
Question: Betty is saving money for a new wallet which costs $100. Betty has only half of the money
she needs. Her parents decided to give her $15 for that purpose, and her grandparents twice as much
as her parents. How much more money does Betty need to buy the wallet?
Answer: Betty needs $5 more to buy the wallet.

Answer the following question in a single brief but complete sentence.
Question: Julie is reading a 120-page book. Yesterday, she was able to read 12 pages and today, she
read twice as many pages as yesterday. If she wants to read half of the remaining pages tomorrow, how
 many pages should she read?
Answer: She would need to read 42 pages tomorrow.

Answer the following question in a single brief but complete sentence.
```

```
Question: James writes a 3-page letter to 2 different friends twice a week. How many pages does he
write a year?
Answer: James writes 624 pages per year.

Answer the following question in a single brief but complete sentence.
Question: {QUESTION}
Answer:
```

## Prompt 3: GSM8K-cot

```
Answer the following question. To do that, first reason about it by saying 'Reasoning:' and then
derive the answer. After that, when you are done, write 'My answer is: ' and write a short and
concise answer to the question.Last, write <DONE>.
Question: Natalia sold clips to 48 of her friends in April, and then she sold half as many clips in
May. How many clips did Natalia sell altogether in April and May?
Answer: Reasoning: Natalia sold 48/2 = <<48/2=24>>24 clips in May.
Natalia sold 48+24 = <<48+24=72>>72 clips altogether in April and May.
My answer is: 72<DONE>

Answer the following question. To do that, first reason about it by saying 'Reasoning:' and then
derive the answer. After that, when you are done, write 'My answer is: ' and write a short and
concise answer to the question.Last, write <DONE>.
Question: Weng earns $12 an hour for babysitting. Yesterday, she just did 50 minutes of babysitting.
How much did she earn?
Answer: Reasoning: Weng earns 12/60 = $<<12/60=0.2>>0.2 per minute.
Working 50 minutes, she earned 0.2 x 50 = $<<0.2*50=10>>10.
My answer is: 10<DONE>

Answer the following question. To do that, first reason about it by saying 'Reasoning:' and then
derive the answer. After that, when you are done, write 'My answer is: ' and write a short and
concise answer to the question.Last, write <DONE>.
Question: Betty is saving money for a new wallet which costs $100. Betty has only half of the money
she needs. Her parents decided to give her $15 for that purpose, and her grandparents twice as much
as her parents. How much more money does Betty need to buy the wallet?
Answer: Reasoning: In the beginning, Betty has only 100 / 2 = $<<100/2=50>>50.
Betty's grandparents gave her 15 * 2 = $<<15*2=30>>30.
This means, Betty needs 100 - 50 - 30 - 15 = $<<100-50-30-15=5>>5 more.
My answer is: 5<DONE>

Answer the following question. To do that, first reason about it by saying 'Reasoning:' and then
derive the answer. After that, when you are done, write 'My answer is: ' and write a short and
concise answer to the question.Last, write <DONE>.
Question: Julie is reading a 120-page book. Yesterday, she was able to read 12 pages and today, she
read twice as many pages as yesterday. If she wants to read half of the remaining pages tomorrow, how
 many pages should she read?
Answer: Reasoning: Maila read 12 x 2 = <<12*2=24>>24 pages today.
So she was able to read a total of 12 + 24 = <<12+24=36>>36 pages since yesterday.
There are 120 - 36 = <<120-36=84>>84 pages left to be read.
Since she wants to read half of the remaining pages tomorrow, then she should read 84/2 =
<<84/2=42>>42 pages.
My answer is: 42<DONE>

Answer the following question. To do that, first reason about it by saying 'Reasoning:' and then
derive the answer. After that, when you are done, write 'My answer is: ' and write a short and
concise answer to the question.Last, write <DONE>.
Question: James writes a 3-page letter to 2 different friends twice a week. How many pages does he
write a year?
Answer: Reasoning: He writes each friend 3*2=<<3*2=6>>6 pages a week
So he writes 6*2=<<6*2=12>>12 pages every week
That means he writes 12*52=<<12*52=624>>624 pages a year
My answer is: 624<DONE>

Answer the following question. To do that, first reason about it by saying 'Reasoning:' and then
derive the answer. After that, when you are done, write 'My answer is: ' and write a short and
concise answer to the question.Last, write <DONE>.
Question: {QUESTION}
Answer:
```

## Prompt 4: Canonicalization

```
Question: "{QUESTION}"
Response: "{RESPONSE}"

Your task is to return **only** the core answer from this response.
Follow these rules:
- Keep only the core answer (e.g., a number, a name, or a short phrase).
- Remove all extra words and filler.
- Expand all abbreviations to their full form (e.g., 'USA' -> 'United States of America').
- Write all numbers with digits, not as words (e.g., 'eight' -> '8').
- For locations, output only the highest-precision part (e.g. 'Seattle, Washington' -> 'Seattle')
```

```
- For dates, unless otherwise specified, format as YYYY-MM-DD (e.g. "August 1, 1990" -> "1990-08-01")
. If only a month or year is specified, leave as-is (e.g. "August" or "2003" or "July, 2000"). Do not
 make up unspecified information.
- No explaining or reasoning. Output the core answer only.
- If the response does not address the question, or if you are unsure what to do, return the response
 unchanged.
- Never alter the meaning of the response, even if it is incorrect.
- Do not infer missing information; only rephrase what is given in the response.
```

## Prompt 5: Canonicalization (math)

```
Response: "{RESPONSE}"

Your task is to return **only** the core answer from this response.
Follow these rules:
- Keep only the core answer, as a raw number or LaTeX string (e.g. '0.5' or '\frac{1}{2}').
- If the answer is the value of a variable, only output the value itself (e.g. 'x=10' -> '10').
- Write all numbers with digits, not as words (e.g., 'eight' -> '8').
- Remove all extra words and filler.
- No explaining or reasoning. Output the core answer only.
- If the response does not contain a numeric value, or if you are unsure what to do, return the
response unchanged.
- Never alter the value of the response, even if it is incorrect.
- Do not infer missing information; only extract what is given in the response.
```

## Prompt 6: Semantic Equivalence

```
You will be given a question, and two possible responses. Your task is to determine whether the two
answers are semantically consistent, i.e., whether the two responses agree on what the answer to the
question is.

Question: {QUESTION}
Response 1: {RESPONSE1}
Response 2: {RESPONSE2}

Are these two responses semantically aligned responses to the question? Respond only with either the
string "Yes" or the string "No".
```

# E  THEORY

## E.1  QUICK REFERENCE

For convenience, we give references to proofs of theorems from the main body.

- Thm. 6 is proved in App. E.5.
- Thm. 9 is re-stated and proved as Thm. 31 in App. E.6.3.

Proving these theorems involves some additional theoretical machinery, which we develop in the remaining sections. The following sections restate some of the notation and definitions from the main body for convenience.

## E.2  WEIGHTED CALIBRATION

A key object in our theory is the notion of *weighted calibration*, from Gopalan et al. (2024), which is capable of expressing many different types of calibration. We use a version of this definition suitable for our LLM setting, stated below.

**Definition 17** (Weighted Calibration, Gopalan et al. (2024)). *For a set $\mathcal{W}$ of weight functions $w :$ $\mathcal{V}^* \times \Delta(\mathcal{V}^N) \to \mathbb{R}^N$, and a distribution $\mathcal{D}$ over pairs $(x, y) \in \mathcal{V}^* \times \mathcal{V}^N$, a model $p_\theta$ is perfectly $\mathcal{W}$-weighted-calibrated on $\mathcal{D}$ if:*

$$\mathbb{E}_{(x,y)\sim\mathcal{D}}[\langle \tilde{y} - p_x, w(x, p_x)\rangle] \equiv 0$$

*where $p_x := p_\theta(\cdot \mid x) \in \Delta(\mathcal{V}^N) \subset \mathbb{R}^{|\mathcal{V}^N|}$ is the model's output distribution on input $x$, and $\tilde{y} \in \{0, 1\}^{|\mathcal{V}^N|}$ is the one-hot-encoding of $y$.*

## E.3  EQUIVALENCE BETWEEN $B$-CALIBRATION AND WEIGHTED CALIBRATION

Here we prove that several kinds of $B$-calibration (including $B$-confidence-calibration and full $B$-calibration) can be characterized in terms of weighted calibration (Definition 17).

**Notation and Setup**   There are two relevant output spaces: the space $\mathcal{V}^N$ of long-form answer strings, and the space $[K]$ of semantic answer classes. Let $M := |\mathcal{V}^N|$. It will be convenient to identify strings $z \in \mathcal{V}^N$ with an index in $[M]$, and we will abuse notation by writing $z \in [M]$.

To simplify some of the proofs, we will rely on an explicit one-hot representation. For a string $y \in \mathcal{V}^N$, we denote its one-hot representation as $\tilde{y} \in \{0, 1\}^M$. For a given prompt $x \in \mathcal{V}^*$, the model's distribution over completions is $p_\theta(\cdot \mid x) \in \Delta(\mathcal{V}^N) \subset \mathbb{R}^M$, which we treat as a vector embedded in $\mathbb{R}^M$. We write $p_x := p_\theta(\cdot \mid x)$ for convenience.

A collapsing function $B : \mathcal{V}^* \times \mathcal{V}^N \to [K]$ assigns to each prompt $x \in \mathcal{V}^*$ and long-answer $y \in \mathcal{V}^N$ a $B$-class $B_x(y) \in [K]$. Moreover, the function $B$ along with the model $p_\theta$ induces a distribution on classes $[K]$ as follows. For a given input $x \in \mathcal{V}^*$, we take the model's distribution $p_\theta(\cdot \mid x)$ and push it forward through $B_x$ to obtain a categorical distribution $\pi_x$ defined as

$$\pi_x := B_x \sharp p_\theta(\cdot \mid x) \in \Delta_K. \tag{8}$$

Explicitly, the probability assigned to a category $c \in [K]$ is:

$$\pi_x(c) = (B_x \sharp p_x)(c) = \Pr_{z \sim p_\theta(\cdot|x)}[B_x(z) = c] = \sum_{z : B_x(z) = c} p_\theta(z \mid x). \tag{9}$$

This push-forward operation can be written in matrix form. Define the collapsing matrix $\mathbf{B}_x$ as:

$$\mathbf{B}_x \in \{0, 1\}^{K \times M}, \qquad [\mathbf{B}_x]_{k,z} = \mathbb{1}_{\{B_x(z) = k\}}. \tag{10}$$

Then the pushforward distribution and ground-truth semantic class can be expressed as

$$\pi_x = \mathbf{B}_x p_x \in \Delta_K, \qquad \mathbf{B}_x \tilde{y} = e_{B_x(y)} \in \mathcal{E}_K.$$

Thus, matrix-vector multiplication exactly implements the pushforward operation:

$$(\pi_x)_k = \sum_{z : B_x(z) = k} p_\theta(z \mid x) = [\mathbf{B}_x p_x]_k. \tag{11}$$

### E.3.1 FULL CALIBRATION

**Definition 18** (Full Calibration). *A distribution $\mathcal{D}$ over prediction-output pairs $(c, y) \in \Delta_K \times \mathcal{E}_K$ is perfectly calibrated if the expected error, conditioned on the prediction, is the zero vector:*

$$\mathbb{E}_{(c,y)\sim\mathcal{D}} [y - c \mid c] \equiv 0. \tag{12}$$

*Note that since $y$ and $c$ are both vectors in $\mathbb{R}^K$, this subtraction is well-defined.*

Now, we apply this template to our setting. We say a model is $B$-calibrated if the distribution it induces over the collapsed, semantic categories is itself perfectly calibrated.

**Definition 19** (B-Calibration). *A model $p_\theta$ is $B$-calibrated on a distribution $\mathcal{D}$ if the induced distribution over pairs $(\pi_x, B_x(y))$ is perfectly calibrated according to Definition 18. Here, $\pi_x = B_x \sharp p_x$ takes the role of the prediction $c$, and the ground-truth category $B_x(y) \in [K]$ takes the role of the outcome $y$. Formally:*

$$\mathbb{E}_{(x,y)\sim\mathcal{D}} [B_x(y) - \pi_x \mid \pi_x] \equiv 0. \tag{13}$$

*Following our convention, the scalar $B_x(y) \in [K]$ is identified with its one-hot vector in $\mathcal{E}_K$ to perform the vector subtraction.*

Now, we provide results for $B$-calibration that are analogous to Definition 4 and Thm. 6 for $B$-confidence-calibration.

**Definition 20** (Semantic Perturbation Function Classes). *Given an arbitrary function $B_x(z) \in [K]$, which we think of as a semantic collapsing function, we define the B-induced weighted function class (a class of perturbation functions $w(x, p_x)$ that generate a perturbation vector based on the context $x$ and the model's predictive distribution $p_x$):*

$$\mathcal{W}_B^{(\text{full})} = \left\{ w \mid w(x, p_x)[z] = \tau(\pi_x)[B_x(z)] \text{ for some } \tau : \Delta^K \to [-1, 1]^K \right\}. \tag{14}$$

Intuitively, every sequence $z$ is assigned a weight based on its semantic category $B_x(z) \in [K]$, and the weighting scheme itself can adapt based on the model's overall categorical prediction $\pi_x$.

**Lemma 21.** *Let $w \in \mathcal{W}_B^{(\text{full})}$ be a weight function defined by $w(x, p_x)[z] = \tau(\pi_x)[B_x(z)]$. Its corresponding vector representation is given by $\mathbf{B}_x^\top \tau(\pi_x)$.*

*Proof.* We will prove the equivalence by showing that for any sequence $z \in \mathcal{V}^N$, the $z$-th component of the vector $\mathbf{B}_x^\top \tau(\pi_x)$ is equal to $\tau(\pi_x)[B_x(z)]$. Let $u = \tau(\pi_x)$, which is a vector in $\mathbb{R}^K$.

Now, we want to analyze the components of the vector $v = \mathbf{B}_x^\top u$.

For any $z \in \mathcal{V}^N$, the $z$-th component of $v$ is given by the definition of matrix-vector multiplication:

$$[v]_z = [\mathbf{B}_x^\top u]_z = \sum_{k=1}^K [\mathbf{B}_x^\top]_{z,k} \cdot u_k = \sum_{k=1}^K [\mathbf{B}_x]_{k,z} \cdot u_k = \sum_{k=1}^K \mathbb{1}_{\{B_x(z)=k\}} \cdot u_k$$

where the last equality is by definition of $\mathbf{B}_x$; see Eq. (10). The indicator function $\mathbb{1}_{\{B_x(z)=k\}}$ is non-zero for only one value of $k$ in the sum, namely when $k$ is equal to the category of the sequence $z$, i.e., $k = B_x(z)$. Therefore, the sum collapses to a single term:

$$[v]_z = 1 \cdot u_{B_x(z)} + \sum_{B_x(z) \neq k} 0 \cdot u_k = u_{B_x(z)}.$$

Substituting back the definition of $u = \tau(\pi_x)$, we get: $[v]_z = \tau(\pi_x)[B_x(z)]$. This expression matches the definition of $w(x, p_x)[z]$ exactly.

Since this holds for all sequences $z$, the vector $\mathbf{B}_x^\top \tau(\pi_x)$ is the vector representation of the function $w(x, p_x)$. $\qquad\square$

With the definition of the weighted class and its vector representation, we can state the main equivalence theorem.

**Theorem 22** (B-Calibration as Weighted Calibration). *A model $p_\theta$ is perfectly B-calibrated if and only if it is perfectly $\mathcal{W}_B^{(\text{full})}$-weighted-calibrated.*

*Proof.* We start from the definition of $B$-calibration, which (as established in Definition 18) is formally expressed as a vector condition:

$$\mathbb{E}\left[e_{B_x(y)} - \pi_x \mid \pi_x\right] = 0.$$

By the properties of conditional expectation, this holds if and only if for all functions $\tau : \Delta_K \to [-1, 1]^K$, it holds

$$\mathbb{E}\left[\langle e_{B_x(y)} - \pi_x, \tau(\pi_x)\rangle\right] = 0. \tag{15}$$

Substituting the matrix representation into Eq. (15):

$$\mathbb{E}\left[\langle e_{B_x(y)} - \pi_x, \tau(\pi_x)\rangle\right] = 0 \iff \mathbb{E}\left[\langle \mathbf{B}_x\tilde{y} - \mathbf{B}_x p_x, \tau(\mathbf{B}_x p_x)\rangle\right] = 0$$
$$\iff \mathbb{E}\left[\langle \mathbf{B}_x(\tilde{y} - p_x), \tau(\mathbf{B}_x p_x)\rangle\right] = 0$$
$$\iff \mathbb{E}\left[\langle \tilde{y} - p_x, \mathbf{B}_x^\top \tau(\mathbf{B}_x p_x)\rangle\right] = 0$$

From Lemma 21, the term $\mathbf{B}_x^\top \tau(\mathbf{B}_x p_x)$ is precisely the vector representation of the function $w(x, p_x)$ from Definition 20. Thus, the condition is equivalent to:

$$\mathbb{E}\left[\langle \tilde{y} - p_x, w(x, p_x)\rangle\right] = 0, \quad \text{for all } w \in \mathcal{W}_B^{(\text{full})},$$

which is exactly the definition of $\mathcal{W}_B^{(\text{full})}$-weighted-calibration ; see Definition 17.

$\square$

### E.3.2 CONFIDENCE CALIBRATION

We first define the standard notion of confidence calibration, a weaker form of calibration that focuses only on the model's top prediction.

**Definition 23** (Confidence Calibration). *A distribution $\mathcal{D}$ over prediction-output pairs $(c, y) \in \Delta_K \times \mathcal{E}_K$ is perfectly confidence-calibrated if, conditioned on the model's top predicted probability, that probability matches the expected outcome. Formally,*

$$\mathbb{E}_{(c,y)\sim\mathcal{D}}\left[y_{k^\star} - c_{k^\star} \mid c_{k^\star}\right] \equiv 0 \text{ where } k^\star = \underset{k\in[K]}{\arg\max}\, c_k. \tag{16}$$

Now, we apply this concept to our LLM setting. A model is B-confidence-calibrated if the categorical distribution it induces is confidence-calibrated.

**Definition 24** (B-Confidence-Calibration). *A model $p_\theta$ is B-confidence-calibrated on a distribution $\mathcal{D}$ if the induced distribution over pairs $(\pi_x, B_x(y))$ is perfectly confidence-calibrated according to Definition 23. This requires that, for $k^\star = \arg\max_{k\in[K]} \pi_x(k)$,*

$$\mathbb{E}_{(x,y)\sim\mathcal{D}}\left[\mathbb{1}\{B_x(y) = k^\star\} - \pi_x(k^\star) \mid \pi_x(k^\star)\right] = 0. \tag{17}$$

We re-state Definition 4 here for convenience:

**Definition 25** (Semantic Perturbation Function Classes). *Given an arbitrary collapsing function $B_x(z) \in [K]$, we define the class $\mathcal{W}_B$ of perturbation functions $w(x, p_x) \in \mathbb{R}^{|\mathcal{V}^N|}$ as follows. These functions generate a perturbation vector based on the prompt $x$ and the model's predictive distribution $p_x$:*

$$\mathcal{W}_B := \left\{w \,\middle|\, \exists \tau : [0, 1] \to [-1, 1] \quad \forall z \in \mathcal{V}^N : w(x, p_x)[z] = \tau\big(\pi_x(k^\star)\big) \cdot \mathbb{1}\{B_x(z) = k^\star\}\right\},$$
$$\text{where } \pi_x := B_x \sharp p_x, \quad k^\star := \underset{k\in[K]}{\arg\max}\, \pi_x(k).$$

Using this definition, we have the following equivalence.

**Theorem 26** (B-Confidence-Calibration as Weighted Calibration). *A model $p_\theta$ is perfectly B-confidence-calibrated if and only if it is perfectly $\mathcal{W}_B$-weighted-calibrated.*

*Proof.* The model is $\mathcal{W}_B$-weighted-calibrated if, for all $w \in \mathcal{W}_B$, the following holds:

$$\mathop{\mathbb{E}}_{(x,y)\sim\mathcal{D}}\left[\langle \tilde{y} - p_x, w(x, p_x)\rangle\right] = 0.$$

For a given $w$ defined by a function $\tau : [0, 1] \to [-1, 1]$, since $\tilde{y}$ is a one-hot vector with a $1$ in the coordinate $z = y$, the first term evaluates to

$$\langle \tilde{y}, w(x, p_x)\rangle = \sum_z \tilde{y}[z]\, w(x, p_x)[z] = w(x, p_x)[y], \tag{18}$$

Substituting the definition of $w$:

$$w(x, p_x)[y] = \tau\left(v_x^\star\right) \cdot \mathbb{1}_{\{B_x(y)=k^\star\}} \text{ where } v_x^\star := \pi_x(k^\star).$$

The second term is $\langle p_x, w(x, p_x)\rangle = \sum_z p_x(z)w(x, p_x)[z]$. Substituting the definition of $w$:

$$\sum_z p_x(z)w(x, p_x)[z] = \sum_z p_x(z)\left(\tau\left(v_x^\star\right) \cdot \mathbb{1}_{\{B_x(z)=k^\star\}}\right)$$

$$= \tau\left(v_x^\star\right) \cdot \sum_z p_x(z)\mathbb{1}_{\{B_x(z)=k^\star\}}$$

$$= \tau\left(v_x^\star\right) \cdot \Pr[B_x(z) = k^\star] = \tau\left(v_x^\star\right) \cdot v_x^\star$$

Putting these together, the weighted calibration condition becomes:

$$\mathop{\mathbb{E}}_{(x,y)\sim\mathcal{D}}\left[\tau\left(v_x^\star\right) \cdot \mathbb{1}_{\{B_x(y)=k^\star\}} - \tau\left(v_x^\star\right) \cdot v_x^\star\right] = 0 \iff \mathop{\mathbb{E}}_{(x,y)\sim\mathcal{D}}\left[\tau\left(v_x^\star\right) \cdot \left(\mathbb{1}_{\{B_x(y)=k^\star\}} - v_x^\star\right)\right] = 0.$$

This condition must hold for all functions $\tau : [0, 1] \to [-1, 1]$. By the properties of conditional expectation, this is true if and only if the term being multiplied by the arbitrary function of $v_x^\star$ has a conditional expectation of zero. This gives us:

$$\mathbb{E}\left[\mathbb{1}_{\{B_x(y)=k^\star\}} - v_x^\star \mid v_x^\star\right] = 0,$$

which is precisely the definition of $B$-confidence-calibration. $\qquad\square$

### E.4 Equivalence between Weighted-Calibration and Local Loss Optimality

For the log-loss $\ell(y, f) := -\sum_i y_i \log(f_i)$, we can analyze perturbations more easily through its dual representation. The dual loss, which operates on a logit vector $z$ is defined as

$$\ell^\star(y, z) = \log\left(\sum_{j=1}^K e^{z_j}\right) - y^T z \text{ and } \nabla_z \ell^\star(y, z) = \mathrm{softmax}(z) - y = f - y$$

The primal and dual views are connected by the variable mapping $z = \log f$, which provides the key equality $\ell(y, f) = \ell^\star(y, z)$. This relationship allows us to translate complex perturbations in the probability space into simple ones in the logit space. A multiplicative re-weighting of the probabilities, defined as $f \star w := \mathrm{softmax}(\log f + w) = \mathrm{softmax}(z + w)$, is equivalent to a simple additive perturbation $w$ on the logits. Therefore, the loss of the perturbed model can be expressed in either world:

$$\underbrace{\ell(y, f \star w)}_{\text{Loss on perturbed probabilities}} = \underbrace{\ell^\star(y, z + w)}_{\text{Loss on perturbed logits}} \tag{19}$$

**Theorem 27** (Equivalence of Calibration and Local Loss Optimality). *Given a model $p_\theta$, a distribution $\mathcal{D}$, and a family of weight functions $\mathcal{W}$ (Definition 17), the model $p_\theta$ is perfectly $\mathcal{W}$-weighted-calibrated on $\mathcal{D}$ if and only if it is $\mathcal{W}$-locally loss-optimal on $\mathcal{D}$.*

*Proof.* We apply the first-order optimality condition to the dual loss $\ell^\star(y, z)$ with a simple additive perturbation $w$ on the logits $z$. With the perturbed loss function, for $\varepsilon > 0$,

$$\mathcal{L}(\varepsilon) = \ell^\star(y, z + \varepsilon w) \text{ and } \frac{d\mathcal{L}}{d\varepsilon}(\varepsilon) = \langle \nabla_z \ell^\star(y, z + \varepsilon w), w\rangle$$

By local loss optimality

$$0 \leq \frac{\mathcal{L}(\varepsilon) - \mathcal{L}(0)}{\varepsilon} = \frac{d\mathcal{L}}{d\varepsilon}(0) + \frac{o(\varepsilon)}{\varepsilon} \longrightarrow \langle \nabla_z \ell^\star(y, z), w \rangle$$

The same reasoning replacing $w$ by $-w$, we also have $\langle \nabla_z \ell^\star(y, z), w \rangle \leq 0$. Thus

$$\ell^\star(y, z) \leq \ell^\star(y, z + \varepsilon w) \implies \langle \nabla_z \ell^\star(y, z), w \rangle = 0$$

The opposite implication follow from convexity, we have:

$$\ell^\star(y, z + w) \geq \ell^\star(y, z) + \langle \nabla_z \ell^\star(y, z), w \rangle.$$

Thus, if $\langle \nabla_z \ell^\star(y, z), w \rangle = 0$ holds, the inequality simplifies to: $\ell^\star(y, z + w) \geq \ell^\star(y, z)$.

Taking the expectation on both side

$$\mathop{\mathbb{E}}_{(x,y) \sim \mathcal{D}}[\ell(y, f)] \leq \mathop{\mathbb{E}}_{(x,y) \sim \mathcal{D}}[\ell(y, f \star w)] \iff \mathop{\mathbb{E}}_{(x,y) \sim \mathcal{D}}[\ell^\star(y, z)] \leq \mathop{\mathbb{E}}_{(x,y) \sim \mathcal{D}}[\ell^\star(y, z + w)]$$

$$\iff \mathop{\mathbb{E}}_{(x,y) \sim \mathcal{D}} \langle f - y, w \rangle = \mathop{\mathbb{E}}_{(x,y) \sim \mathcal{D}} \langle \nabla_z \ell^\star(y, z), w \rangle = 0$$

A model is calibrated under the log-loss if and only if its expected prediction error $f - y$ is orthogonal to any systematic perturbation $w$ of its logits. $\qquad \square$

### E.5 PROOF OF THM. 6

We can now combine the above ingredients to directly prove Thm. 6 from the main body.

*Proof.* Recall we have a model $p_\theta$, a collapsing function $B$, and a distribution $\mathcal{D}$.

We have the following equivalences:

$$p_\theta \text{ is } B\text{-confidence-calibrated on } \mathcal{D} \iff p_\theta \text{ is } \mathcal{W}_B\text{-weighted-calibrated on } \mathcal{D} \quad \text{(by Thm. 26)}$$
$$\iff p_\theta \text{ is } \mathcal{W}_B\text{-locally-loss-optimal on } \mathcal{D} \quad \text{(by Thm. 27)}$$

$\qquad \square$

### E.6 AUTOREGRESSIVE SETTINGS

Recall the definition of the perturbation operator, Definition 3,

$$\forall z \in \mathcal{V}^N : \quad (f \star w)[z] := \mathrm{softmax}\big(w[z] + \log f[z]\big) = \frac{f[z] \exp(w[z])}{\sum_{z' \in \mathcal{V}^N} f[z'] \exp(w[z'])} \tag{20}$$

highlighting that this transformation is a multiplicative reweighting of the reference distribution $f$ by $e^{w[z]}$, followed by a renormalization to get a valid distribution. Applying it to the next-token setting, we obtain significant simplifications for both full and confidence calibration.

#### E.6.1 WEIGHTED CALIBRATION

**Lemma 28** (Autoregressive Decomposition of the Perturbation). *For any position $i$, the perturbed conditional probability of the next token is the original conditional probability multiplied by a ratio of "lookahead expectations":*

$$(p_x \star w_x)(z_i \mid z_{<i}) = p_x(z_i \mid z_{<i}) \cdot \frac{\mathbb{E}_{z_{>i} \sim p_x(\cdot \mid z_{\leq i})}\big[\exp(w_x(z_{\leq i}, z_{>i}))\big]}{\mathbb{E}_{z_{\geq i} \sim p_x(\cdot \mid z_{<i})}\big[\exp(w_x(z_{<i}, z_{\geq i}))\big]}. \tag{21}$$

*Proof.* Let $Z := \sum_z p_x(z) e^{w_x(z)}$. By definition of conditional probability,

$$(p_x \star w_x)(z_i \mid z_{<i}) = \frac{(p_x \star w_x)(z_{\leq i})}{(p_x \star w_x)(z_{<i})}. \tag{22}$$

Expanding the perturbation operator and applying $p_x(z_{\leq i}, z_{>i}) = p_x(z_{\leq i}) p_x(z_{>i} \mid z_{\leq i})$,

$$(p_x \star w_x)(z_{\leq i}) = \frac{1}{Z} \sum_{z_{>i}} p_x(z_{\leq i}, z_{>i}) \, e^{w_x(z_{\leq i}, z_{>i})}$$

$$= \frac{p_x(z_{\leq i})}{Z} \underset{z_{>i} \sim p_x(\cdot \mid z_{\leq i})}{\mathbb{E}} [e^{w_x(z_{\leq i}, z_{>i})}].$$

Similarly,

$$(p_x \star w_x)(z_{<i}) = \frac{p_x(z_{<i})}{Z} \underset{z_{\geq i} \sim p_x(\cdot \mid z_{<i})}{\mathbb{E}} [e^{w_x(z_{<i}, z_{\geq i})}]. \tag{23}$$

Taking the ratio and canceling $Z$,

$$(p_x \star w_x)(z_i \mid z_{<i}) = \frac{p_x(z_{\leq i})}{p_x(z_{<i})} \cdot \frac{\mathbb{E}_{z_{>i} \sim p_x(\cdot \mid z_{\leq i})}[e^{w_x(z_{\leq i}, z_{>i})}]}{\mathbb{E}_{z_{\geq i} \sim p_x(\cdot \mid z_{<i})}[e^{w_x(z_{<i}, z_{\geq i})}]}$$

$$= p_x(z_i \mid z_{<i}) \cdot \frac{\mathbb{E}_{z_{>i} \sim p_x(\cdot \mid z_{\leq i})}[e^{w_x(z_{\leq i}, z_{>i})}]}{\mathbb{E}_{z_{\geq i} \sim p_x(\cdot \mid z_{<i})}[e^{w_x(z_{<i}, z_{\geq i})}]}.$$

□

### E.6.2 $B$-CALIBRATION

The general decomposition in Lemma 28 is insightful but computationally intractable, as it requires summing over all possible future sequences. We now show that for our specific class of semantic perturbations $\mathcal{W}_B$, this complex ratio simplifies dramatically into a small, efficient arithmetic circuit. The key is to define two "autoregressive B-confidence" vectors that can be tracked during generation.

**Autoregressive $B$-confidence**  Given a model $p_x$ and a semantic mapping $B_x$, we define:

1. The initial B-confidence $g_0(x) \in \Delta_K$, which is the model's overall predicted distribution on the $K$ categories before generation begins. This corresponds to the $B$-induced pushforward distribution $\pi_x = B_x \sharp p_x$:

$$g_0(x)[b] := \Pr_{z \sim p_x} [B_x(z) = b]. \tag{24}$$

2. The conditional B-confidence $g_i(x, z_{\leq i}) \in \Delta_K$, which is the model's predicted distribution on categories, conditioned on having generated the prefix $z_{\leq i}$:

$$g_i(x, z_{\leq i})[b] := \Pr_{z' \sim p_x(\cdot \mid z_{\leq i})} [B_x(z_{\leq i}, z') = b]. \tag{25}$$

**Theorem 29** (Simple Circuit for B-Perturbations). *For any perturbation $w \in \mathcal{W}_B$ (defined by a scaling function $\tau$), the perturbed next-token probability is proportional to the original conditional probability multiplied by a simple circuit $C_w$:*

$$(p_x \star w_x)(z_i \mid z_{<i}) \propto p_x(z_i \mid z_{<i}) \cdot C_w(g_0(x), g_i(x, z_{\leq i})), \tag{26}$$

*where the constant of proportionality does not depend on $z_i$, and*

$$C_w(g_0, g_i) = \sum_{b=1}^{K} \exp(\tau(g_0)[b]) \cdot g_i[b]. \tag{27}$$

*This circuit has constant depth and width linear in $K$.*

*Proof.* From Lemma 28, we know that

$$(p_x \star w_x)(z_i \mid z_{<i}) = p_x(z_i \mid z_{<i}) \cdot \frac{\mathbb{E}_{z \sim p_x(\cdot \mid z_{\leq i})}[e^{w_x(z_{\leq i}, z)}]}{\mathbb{E}_{z \sim p_x(\cdot \mid z_{<i})}[e^{w_x(z_{<i}, z)}]}. \tag{28}$$

For $w \in \mathcal{W}_B$, by definition, $w_x(z) = \tau(g_0(x))[B_x(z)]$  where  $g_0(x) = B_x \sharp p_x$.

Expanding the expectation,

$$\mathbb{E}_{z \sim p_x(\cdot|z_{\leq i})} \left[ e^{w_x(z_{\leq i}, z)} \right] = \mathbb{E}_{z \sim p_x(\cdot|z_{\leq i})} \left[ e^{\tau(g_0(x))[B_x(z_{\leq i}, z)]} \right]$$

$$= \sum_{b=1}^{K} \Pr[B_x(z_{\leq i}, z) = b] \cdot e^{\tau(g_0(x))[b]}$$

$$= \sum_{b=1}^{K} g_i(x, z_{\leq i})[b] \cdot e^{\tau(g_0(x))[b]}.$$

The denominator is an expectation over $z \sim p_x(\cdot|z_{<i})$, which depends only on the prefix $z_{<i}$ and not on the choice of $z_i$. Hence it is a constant with respect to $z_i$ and can be absorbed into the proportionality. Therefore, $(p_x \star w_x)(z_i \mid z_{<i}) \propto p_x(z_i \mid z_{<i}) \cdot \langle \exp(\tau(g_0(x))), g_i(x, z_{\leq i}) \rangle$. $\qquad\square$

### E.6.3 PROOF OF THM. 9: A SIMPLE CIRCUIT FOR B-CONFIDENCE-PERTURBATIONS

The circuit for general B-perturbations involves a $K$-dimensional inner product. For the more restricted class of B-confidence-perturbations, $\mathcal{W}_B$, the structure simplifies even further to a trivial scalar arithmetic circuit. First, we define the key scalar quantities needed.

**Definition 30** (Autoregressive Top-1 Confidence). *Given a model $p_x$ and mapping $B_x$, let $\pi_x = B_x \sharp p_x$ be the initial categorical distribution, and let $k^\star := \arg\max_{k \in [K]} (\pi_x)_k$ be the single most likely category. We define:*

1. *The top confidence value $v_x^\star \in [0, 1]$, which is the model's confidence in this top category:*

$$v_x^\star := (\pi_x)_{k^\star}. \tag{29}$$

2. *The conditional probability of hitting the top category, $g_i^{(\mathrm{conf})}(x, z_{\leq i}) \in [0, 1]$, which is the probability of eventually generating a sequence in category $k^\star$, given the prefix $z_{\leq i}$:*

$$g_i^{(\mathrm{conf})}(x, z_{\leq i}) := \Pr_{z' \sim p_x(\cdot|z_{\leq i})}[B_x(z_{\leq i}, z') = k^\star]. \tag{30}$$

With these scalars, the autoregressive update becomes a simple linear transformation.

**Theorem 31.** *For any perturbation $w \in \mathcal{W}_B$ (defined by a function $\tau$), the perturbed next-token probability is proportional to the original probability modified by a simple scalar circuit $C_w$:*

$$(p_x \star w_x)(z_i \mid z_{<i}) \propto p_x(z_i \mid z_{<i}) \cdot C_w(v_x^\star, g_i^{(\mathrm{conf})}(x, z_{\leq i})), \tag{31}$$

*where the circuit $C_w$ is a linear function of $g_i^{(\mathrm{conf})}$:*

$$C_w(v, g) := 1 + (\exp(\tau(v)) - 1) \times g. \tag{32}$$

*Proof.* By Lemma 28,

$$(p_x \star w_x)(z_i \mid z_{<i}) \propto p_x(z_i \mid z_{<i}) \cdot \mathbb{E}_{z \sim p_x(\cdot|z_{\leq i})} \left[ \exp(w_x(z)) \right]. \tag{33}$$

For $w \in \mathcal{W}_B$ we have

$$w_x(z) = c_x \cdot \mathbb{1}\{B_x(z) = k^\star\}, \quad \text{with } c_x := \tau(v_x^\star).$$
$$\exp(w_x(z)) = 1 + (\exp(c_x) - 1) \cdot \mathbb{1}\{B_x(z) = k^\star\}.$$

Taking expectation under $z \sim p_x(\cdot \mid z_{\leq i})$ yields

$$1 + (\exp(c_x) - 1) \Pr[B_x(z) = k^\star \mid z_{\leq i}] = 1 + (\exp(\tau(v_x^\star)) - 1) g_i^{(\mathrm{conf})}(x, z_{\leq i}). \tag{34}$$

By Lemma 28, the perturbed conditional probability is the original $p_x(z_i \mid z_{<i})$ scaled by the ratio of this term to an analogous denominator depending only on the prefix $z_{<i}$. Since the denominator is independent of $z_i$, it can be absorbed into the overall proportionality constant. $\qquad\square$

### E.7 QUANTITATIVE BOUNDS ON MULTI-CLASS CALIBRATION AND POST-PROCESSING GAP

Beyond cross-entropy loss, we provide in this section a generalization for the class of proper loss functions and quantitative bounds relating post-processing and calibration gap. The main result in this section, Thm. 36 should be interpreted as a generalization of Theorem E.3 in Błasiok et al. (2023) to the multi-class setting, and a robust version of Thm. 27: it essentially states that a model is "close to" $\mathcal{W}$-weighted-calibrated if it is "close to" $\mathcal{W}$-loss-optimal.

First, we recall a standard result on convex representation of proper losses (Savage, 1971; Schervish, 1989; Gneiting & Raftery, 2007).

**Definition 32** (Savage representation). *A loss function* $\ell : \{e_1, \ldots, e_K\} \times \Delta_K \to \mathbb{R}$ *is* proper *iff there exists a convex function* $\phi : \Delta_K \to \mathbb{R}$ *such that*

$$\ell(y, v) = -\phi(v) + \langle v - y, \nabla\phi(v)\rangle. \tag{35}$$

Next, define the convex conjugate $\psi = \phi^*$, a dual variable, and the dual form of the loss.

**Definition 33** (Dual loss). *For a proper loss* $\ell$ *with potential* $\phi$ *as in Definition 32, define:*

$$\text{Convex conjugate:} \quad \psi(u) := \phi^*(u) := \sup_{v \in \Delta_K} \big(\langle u, v\rangle - \phi(v)\big),$$

$$\text{Dual variable:} \quad dual(v) := \nabla\phi(v),$$

$$\text{Dual loss:} \quad \ell^{(\psi)}(y, z) := \psi(z) - \langle y, z\rangle.$$

**Remark 34.** *The dual parameterization of Definition 33 satisfies:*

1. *Agreement between primal and dual losses:* $\ell^{(\psi)}(y, dual(v)) = \ell(y, v)$.

2. *Probability* $\to$ *dual map:* $dual(v) = \nabla\phi(v)$ *for all* $v \in \Delta_K$.

3. *Dual* $\to$ *probability map:* $v = \nabla\psi(dual(v))$ *for all* $v \in \Delta_K$.

**Definition 35** (Generalized dual calibration and post-processing gap). *Let* $\mathcal{W}$ *be a class of functions* $w : \mathcal{X} \times \mathbb{R}^K \to \mathbb{R}^K$, *and let* $\mathcal{D}$ *be a distribution over* $\mathcal{X} \times \{e_1, \ldots, e_K\}$.

*For a predictor* $f : \mathcal{X} \to \Delta_K$, *let* $g : \mathcal{X} \to \mathbb{R}^K$ *be its dual representation such that*

$$f(x) = \nabla\psi(g(x)) \quad \forall x \in \mathcal{X}. \tag{36}$$

*Define for shorthand*

$$\Delta(w) := \mathbb{E}_{(x,y)\sim\mathcal{D}}\big[\langle y - f(x), w(x, g(x))\rangle\big], \qquad \mathcal{L}(h) := \mathbb{E}_{(x,y)\sim\mathcal{D}}[\ell^{(\psi)}(y, h(x))]. \tag{37}$$

- *The* dual calibration error *of* $g$ *with respect to* $\mathcal{W}$ *is*

$$\text{CE}(g; \mathcal{W}) := \sup_{w \in \mathcal{W}} |\Delta(w)|. \tag{38}$$

- *The* dual post-processing gap *of* $g$ *with respect to a function class* $\mathcal{H}$ *is*

$$\text{Gap}(g; \mathcal{H}) := \mathcal{L}(g) - \inf_{h \in \mathcal{H}} \mathcal{L}(h). \tag{39}$$

**Theorem 36** (General relationship between calibration and post-processing). *Let* $\psi : \mathbb{R}^K \to \mathbb{R}$ *be differentiable and* $\lambda$-*smooth, i.e.* $\nabla\psi$ *is* $\lambda$-*Lipschitz. Let* $\mathcal{W}$ *be a class of bounded functions* $w : \mathcal{X} \times \mathbb{R}^K \to \mathbb{R}^K$ *with* $\|w_x\| \leq 1$. *For* $w \in \mathcal{W}$ *and* $\beta \in [-1/\lambda, 1/\lambda]$, *define the perturbed dual predictor*

$$g_w(x) := g(x) + \beta\, w(x, g(x)). \tag{40}$$

*Let* $\mathcal{G}_\mathcal{W} := \{g_w : w \in \mathcal{W}, \ \beta \in [-1/\lambda, 1/\lambda]\}$. *Then, for every* $g : \mathcal{X} \to \mathbb{R}^K$ *and distribution* $\mathcal{D}$,

$$\frac{1}{2}\Big(\text{CE}(g; \mathcal{W})\Big)^2 \leq \lambda \cdot \text{Gap}(g; \mathcal{G}_\mathcal{W}) \leq \text{CE}(g; \mathcal{W}). \tag{41}$$

*Proof.* By the definition of $\ell^{(\psi)}$,

$$\mathcal{L}(g) - \mathcal{L}(g_w) = \mathbb{E}\big[\psi(g(x)) - \langle y, g(x)\rangle - \psi(g_w(x)) + \langle y, g_w(x)\rangle\big]$$
$$= \mathbb{E}\big[\psi(g(x)) - \psi(g_w(x)) + \beta\langle y, w(x, g(x))\rangle\big].$$

By convexity and $\lambda$-smoothness of $\psi$, for $z = g(x)$, $z' = g_w(x)$ and $w_x = w(x, g(x))$

$$\langle \nabla \psi(z), \beta w_x \rangle \;\leq\; \psi(z') - \psi(z) \;\leq\; \langle \nabla \psi(z), \beta w_x \rangle + \frac{\lambda \beta^2}{2} \|w_x\|^2. \tag{42}$$

Since $f(x) = \nabla \psi(g(x))$ and $\|w_x\| \leq 1$, this yields

$$\beta \, \Delta(w) - \frac{\lambda \beta^2}{2} \;\leq\; \mathcal{L}(g) - \mathcal{L}(g_w) \;\leq\; \beta \, \Delta(w). \tag{43}$$

*Lower bound.* For $w \in \mathcal{W}$, set $\beta = \Delta(w)/\lambda$ (which lies in $[-1/\lambda, 1/\lambda]$). Then

$$\frac{1}{2\lambda} \, \Delta(w)^2 \;\leq\; \mathcal{L}(g) - \mathcal{L}(g_w). \tag{44}$$

Taking $\sup\limits_{w \in \mathcal{W}}$ yields

$$\frac{1}{2} \left( \mathrm{CE}(g; \mathcal{W}) \right)^2 \;\leq\; \lambda \cdot \mathrm{Gap}(g; \mathcal{G}_\mathcal{W}). \tag{45}$$

*Upper bound.* For $g_w \in \mathcal{G}_\mathcal{W}$, since $|\beta| \leq 1/\lambda$

$$\mathcal{L}(g) - \mathcal{L}(g_w) \;\leq\; \beta \, \Delta(w) \leq \frac{1}{\lambda} |\Delta(w)|. \tag{46}$$

Taking $\sup\limits_{w \in \mathcal{W}}$ gives

$$\lambda \cdot \mathrm{Gap}(g; \mathcal{G}_\mathcal{W}) \;\leq\; \mathrm{CE}(g; \mathcal{W}). \tag{47}$$

Combining the upper and lower bounds proves Eq. (41). $\qquad\square$

**Remark 37** (Tighter exponent under strong convexity). *If, in addition, $\psi$ is $\mu$-strongly convex for some $\mu > 0$ i.e.*

$$\psi(z') \geq \psi(z) + \langle \nabla \psi(z), z' - z \rangle + \tfrac{\mu}{2} \|z' - z\|^2,$$

*then one obtains matching upper and lower bounds. In this case, both inequalities in Thm. 36 become quadratic in the calibration error:*

$$\frac{\mu}{2\lambda^2} \left( \mathrm{CE}(g; \mathcal{W}) \right)^2 \;\leq\; \mathrm{Gap}(g; \mathcal{G}_\mathcal{W}) \;\leq\; \frac{1}{2\mu} \left( \mathrm{CE}(g; \mathcal{W}) \right)^2. \tag{48}$$

*That is, the dual post-processing gap and the squared dual calibration error are equivalent up to constants determined by $(\mu, \lambda)$.*

### E.7.1 SPECIALIZATION TO CROSS-ENTROPY LOSS

For completeness, we summarize the standard facts about the dual parametrization of the negative log-loss in Table 4.

Table 4: Duality relationships for the Negative Log-Loss (Cross-Entropy) proper scoring rule.

| | |
|---|---|
| Primal Proper Loss ($\ell_{\mathrm{nll}}$) | $\ell(y, v) = -\sum_{i=1}^{K} y_i \log v_i$ |
| Convex Function ($\phi$) | $\phi(v) = \sum_{i=1}^{K} v_i \log(v_i)$    (Negative Entropy) |
| Convex Conjugate ($\phi^*$) | $\phi^*(z) = \log\left( \sum_{i=1}^{K} \exp(z_i) \right)$    (Log-Sum-Exp) |
| Dual Loss ($\ell_{\mathrm{nll}}^*$) | $\ell^*(y, z) = \phi^*(z) - y^T z$ |
| Dual Mapping ($\nabla \phi^*$) | $\nabla \phi^*(z) = \mathrm{softmax}(z)$ |

The log-sum-exp function $\phi^*(z) = \log\left( \sum_{i=1}^{K} \exp(z_i) \right)$ is 1/4-smooth, as shown in Beck & Teboulle (2003) and Nesterov (2005), so Thm. 36 applies with $\lambda = 1/4$. Moreover, to translate

the result into the notation of our main theorems, recall the relationship between the primal prediction $f(x)$ and its dual representation $g(x)$:

$$f(x) = \nabla\phi^*(g(x)) = \text{softmax}(g(x))$$
$$g(x) = \log(f(x))$$

The perturbed loss can then be expressed in terms of the dual variables. The dual loss on perturbed logits $g + w$ is equivalent to the primal loss on the perturbed probability distribution $f \star w$:

$$\ell^*_{\text{nll}}(y, g + w) = \ell_{\text{nll}}(y, \text{softmax}(g + w)) = \ell_{\text{nll}}(y, f \star w)$$

where $f \star w = \text{softmax}(\log(f) + w)$.

### E.8 CONFORMAL PREDICTION VIA WEIGHTED CALIBRATION

Here we observe that conformal prediction guarantees can be expressed as a type of *weighted calibration* (Gopalan et al., 2024), for a particular weight family.

Recall conformal prediction asks for a model $F(x)$ which outputs a *set* of labels, with the guarantee that this set contains the true label with high probability. Specifically, a conformal predictor has *coverage $\alpha$* if:

$$\Pr_{x,y\sim\mathcal{D}}[y \in F(x)] \geq 1 - \alpha. \tag{49}$$

For an introduction to conformal prediction, see Angelopoulos et al. (2023) or the lecture notes of Tibshirani (2023).

#### E.8.1 CONFORMAL PREDICTION FROM FULL CALIBRATION

Given a standard predictor $f$, which outputs a distribution on labels, one natural way to construct a conformal predictor $F_\alpha$ is: given input $x$, and prediction $f(x)$, output the set of highest-predicted-probability labels which sum to total probability $1 - \alpha$. This means, outputting the $K$ most-likely classes according to $f(x)$, where $K$ is chosen per-sample based on the predicted probabilities.

The first observation (which is folklore) is: if the predictor $f$ is perfectly calibrated, in the sense of full-calibration, then the induced conformal predictor $F_\alpha$ is correct (i.e. has coverage $\alpha$). This statement is not very relevant in practice, since full calibration is often too strong to hold. However, we can achieve the same result with a weaker notion of calibration. This is a straightforward result; we sketch the argument below.

#### E.8.2 CONFORMAL PREDICTION FROM WEIGHTED CALIBRATION

**Lemma 38.** *Suppose $f : \mathcal{X} \to \Delta_N$ is perfectly weighted-calibrated (in the sense of Gopalan et al. (2024)) with respect to the following family of weight functions $w(f) \in \mathbb{R}^N$:*

$$\mathcal{W} := \{w(f) = \sigma \mathbb{1}_{T_\alpha(f)} \mid \alpha \in [0, 1], \sigma \in \{\pm 1\}\} \tag{50}$$

*Where $\mathbb{1}_T \in \{0, 1\}^N$ is the indicator-vector for set of indices $T$, and the set $T$ contains the highest-probability labels, defined as:*

$$t_\alpha^*(f) := \max\{t : \left(\sum_{i\in[N]} f_i \mathbb{1}\{f_i \geq t\}\right) \geq 1 - \alpha\} \qquad \text{(the threshold probability, given } f\text{)}$$

$$T_\alpha(f) := \{i : f_i \geq t_\alpha^*(f)\} \qquad \text{(The set of top-class indices, for given level } \alpha\text{)}$$

*That is, suppose:*

$$\mathbb{E}_{(x,y)\sim\mathcal{D}} [\langle y - f(x), w(f(x))\rangle] \equiv 0$$

*Then, the induced conformal predictor $F_\alpha$ of $f$ is valid at all coverage levels $\alpha$.*

*Proof.* (Sketch) Notice that by construction, $\langle f, \mathbb{1}_{T_\alpha(f)}\rangle \geq 1 - \alpha$. Therefore by calibration we must have: $\langle y, \mathbb{1}_{T_\alpha(f)}\rangle \geq 1 - \alpha$.

Moreover, the set $T_\alpha(f)$ is exactly the output of the induced conformal predictor $F_\alpha$, given base prediction $f$. Therefore

$$\Pr[y \in T_\alpha(f(x))] = \mathbb{E}[\langle y, \mathbb{1}_{T_\alpha(f)}\rangle] \tag{51}$$

$$\geq 1 - \alpha \tag{52}$$

$\square$

By the general connection of Theorem 27, if a model $f$ is $\mathcal{W}$-locally-loss-optimal w.r.t. the weight class of Equation (50), then the induced conformal predictor $F_\alpha$ has coverage $\alpha$ for all $\alpha \in [0, 1]$.

# F  DISAGGREGATED RELIABILITY DIAGRAM RESULTS

In this section, we report disaggregated reliability diagram results for individual configurations we evaluated. The plots are displayed as follow:

- the right three columns present results for instruct models,
- the left three columns present results for the corresponding base models.

In some cases, there are multiple instruct models trained from a single base models, hence for some base models, their results are being presented multiple times.

Some instruct models do not have a public corresponding base model—in those cases, the left three columns of the row are empty.

As discussed in the Sec. 5, TriviaQA and SimpleQA were not evaluated for the CoT response style.

The figures start on the next page. For a quick references:

- `GSM8K` in App. F.1
- `OpenMathInstruct` in App. F.2
- `TriviaQA` in App. F.3
- `SimpleQA` in App. F.4

## F.1 GSM8K

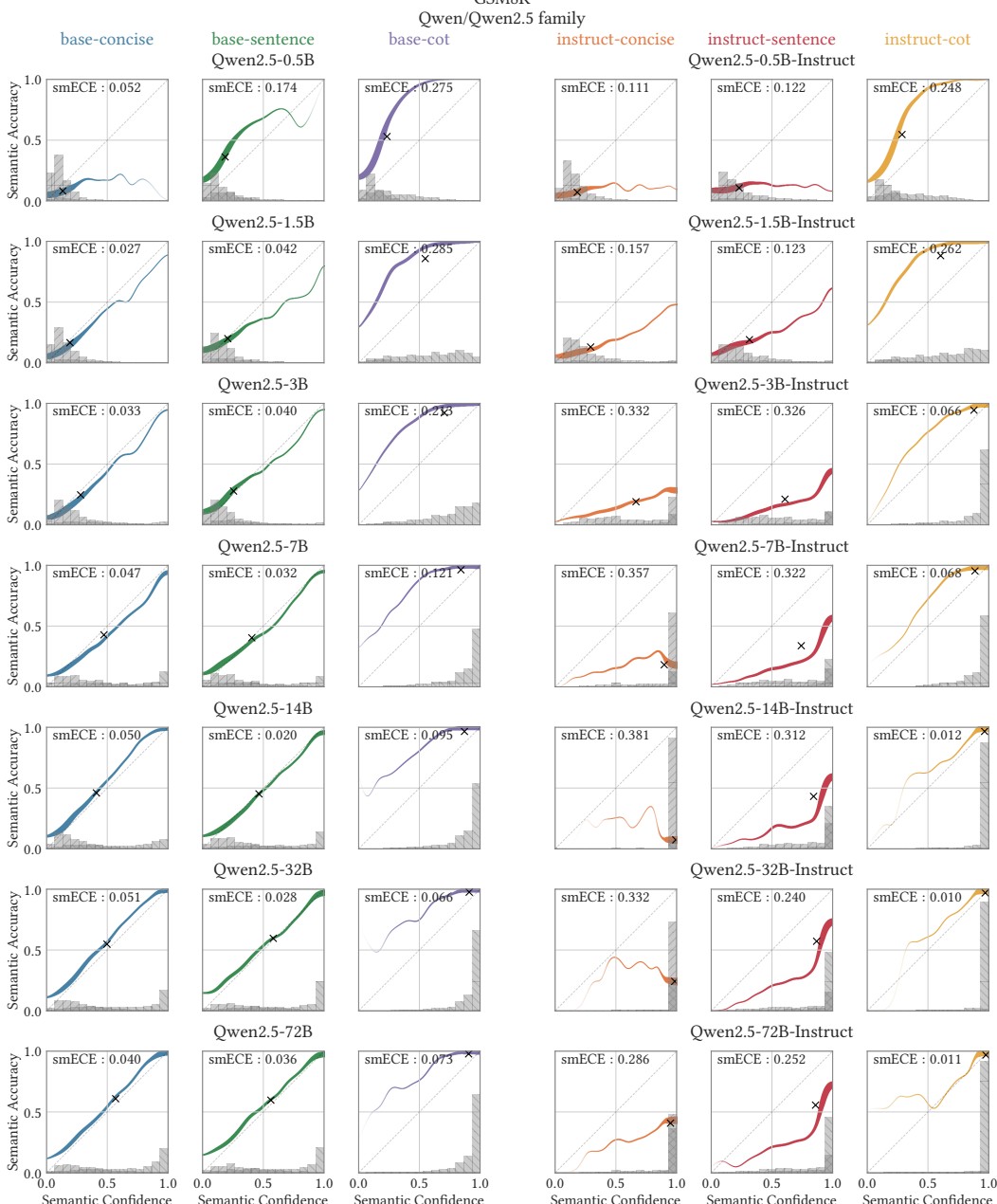

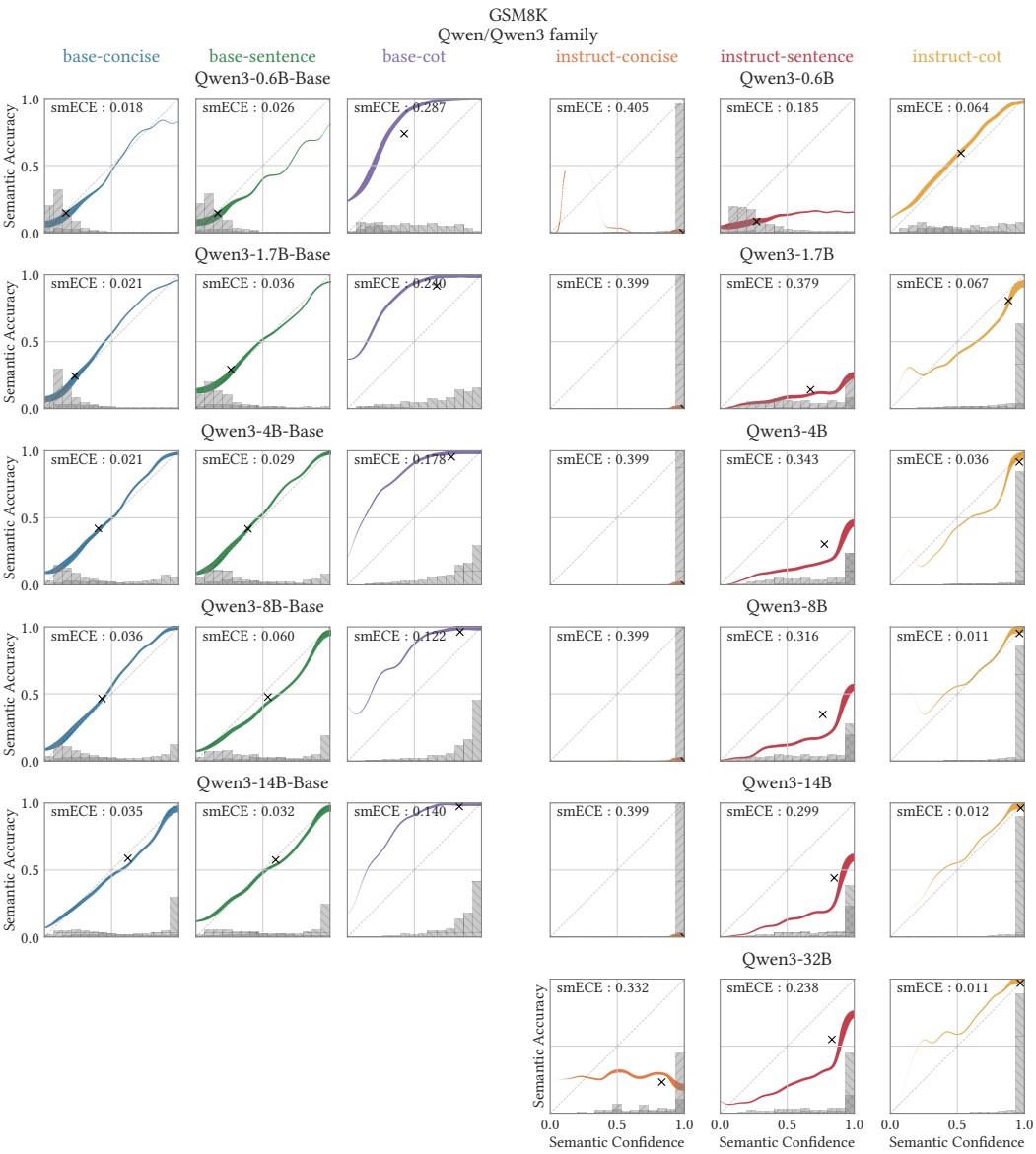

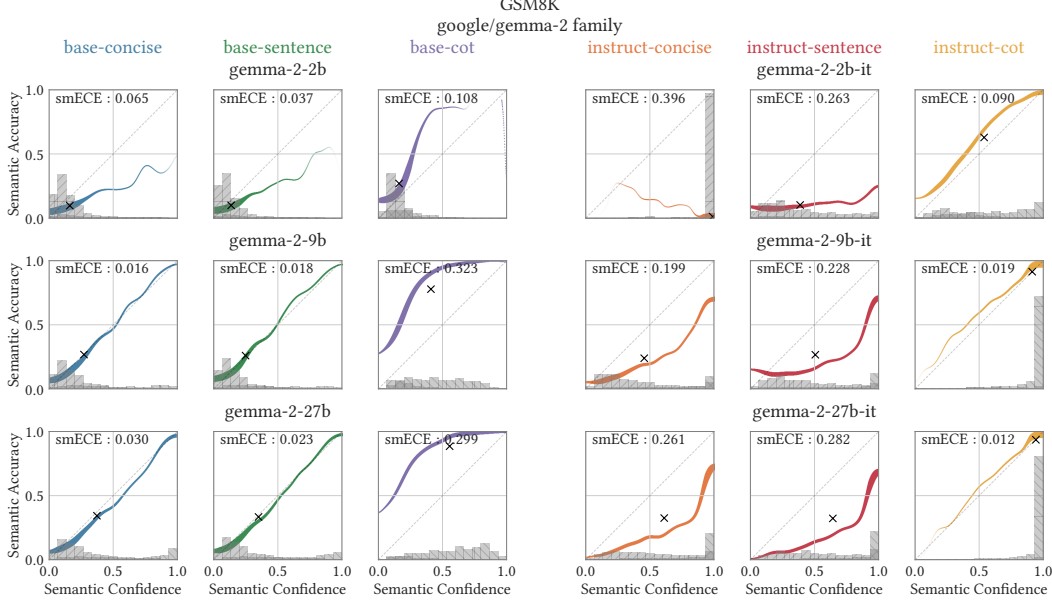

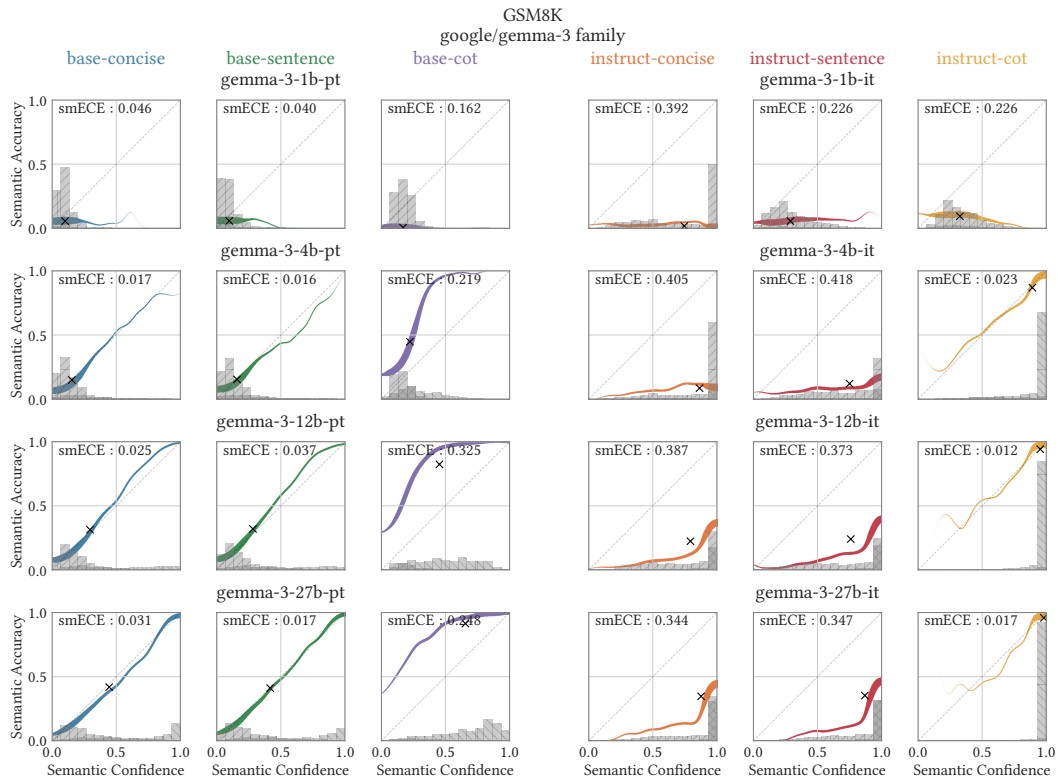

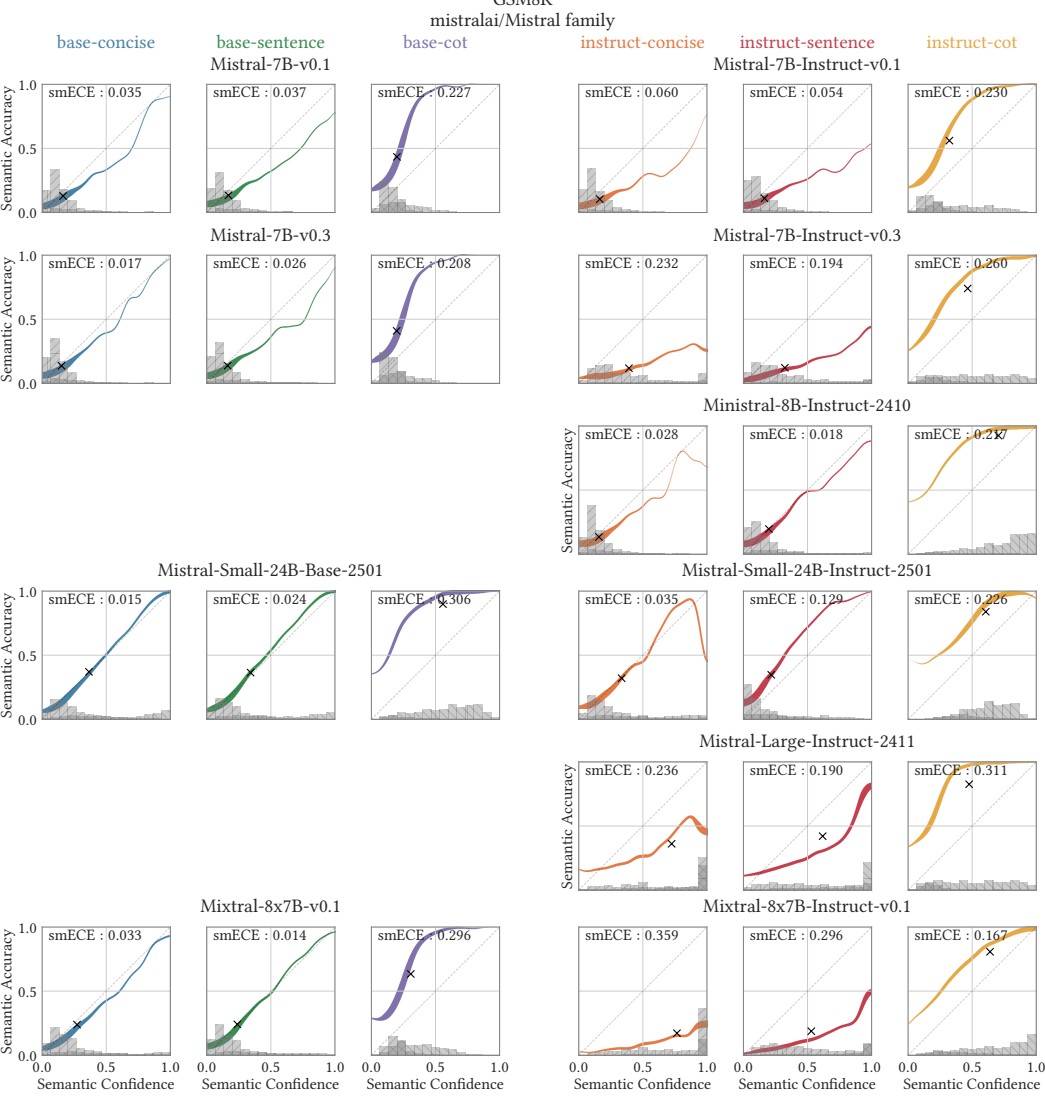

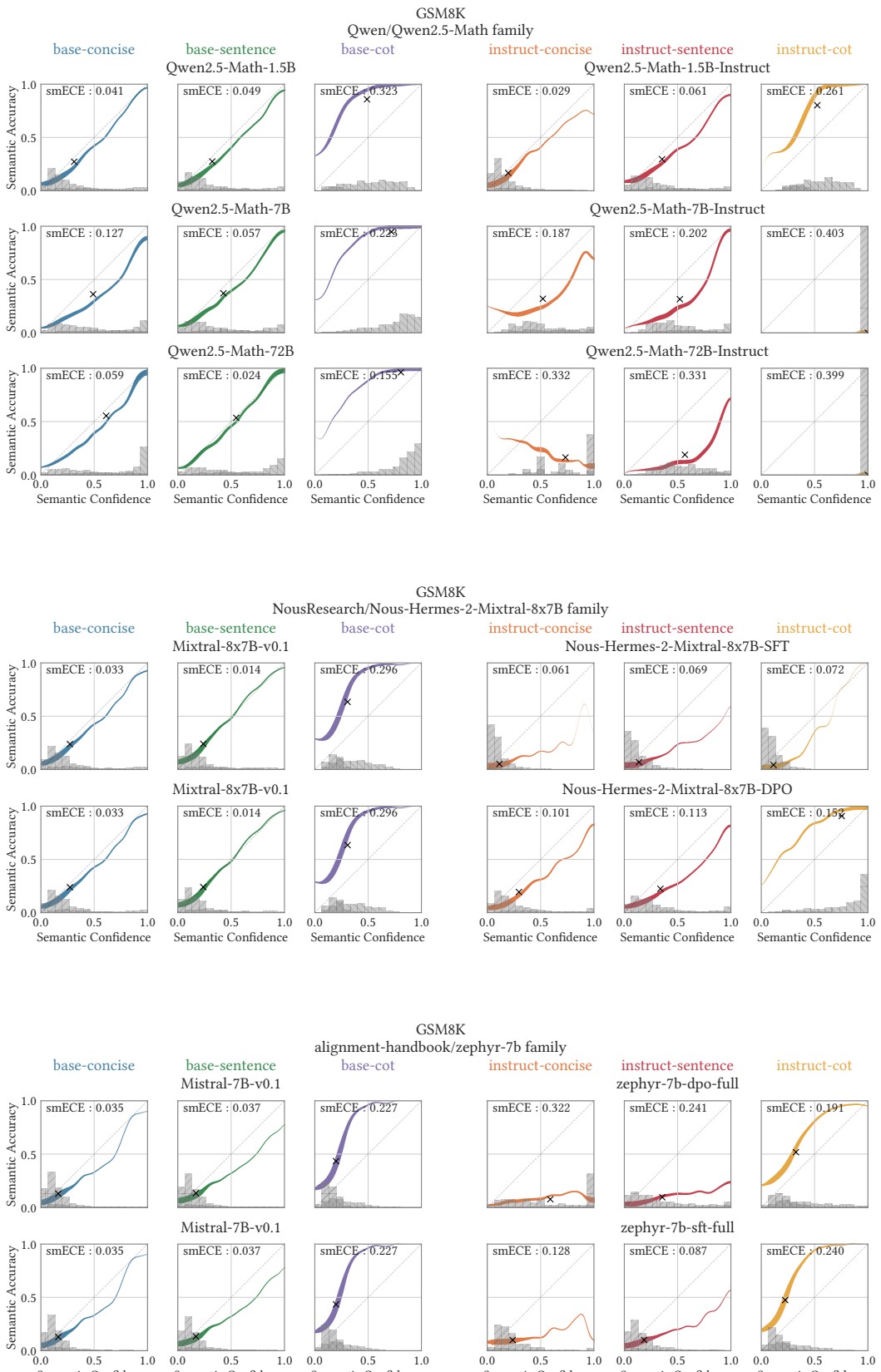

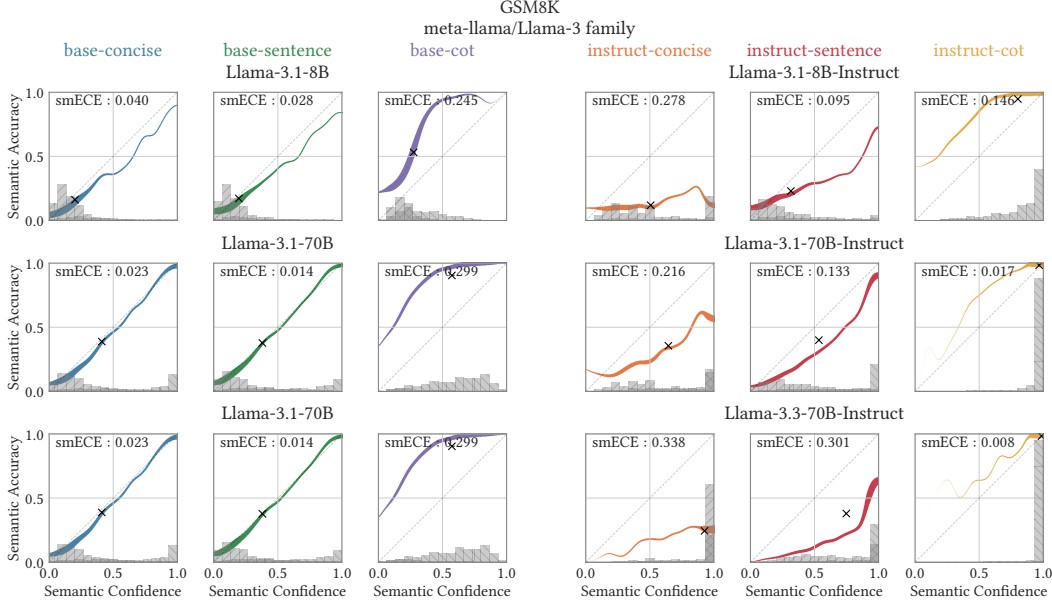

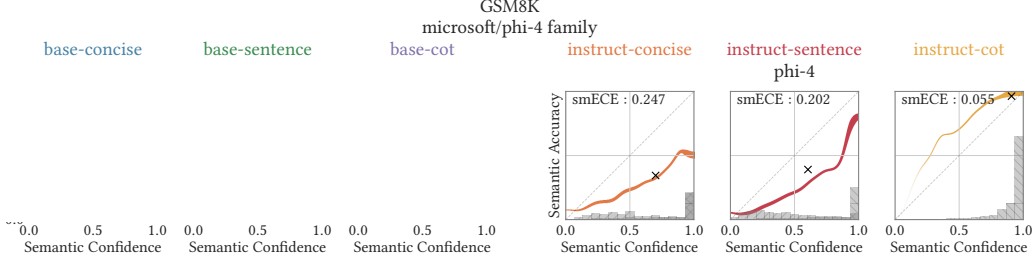

## F.2 OPENMATHINSTRUCT

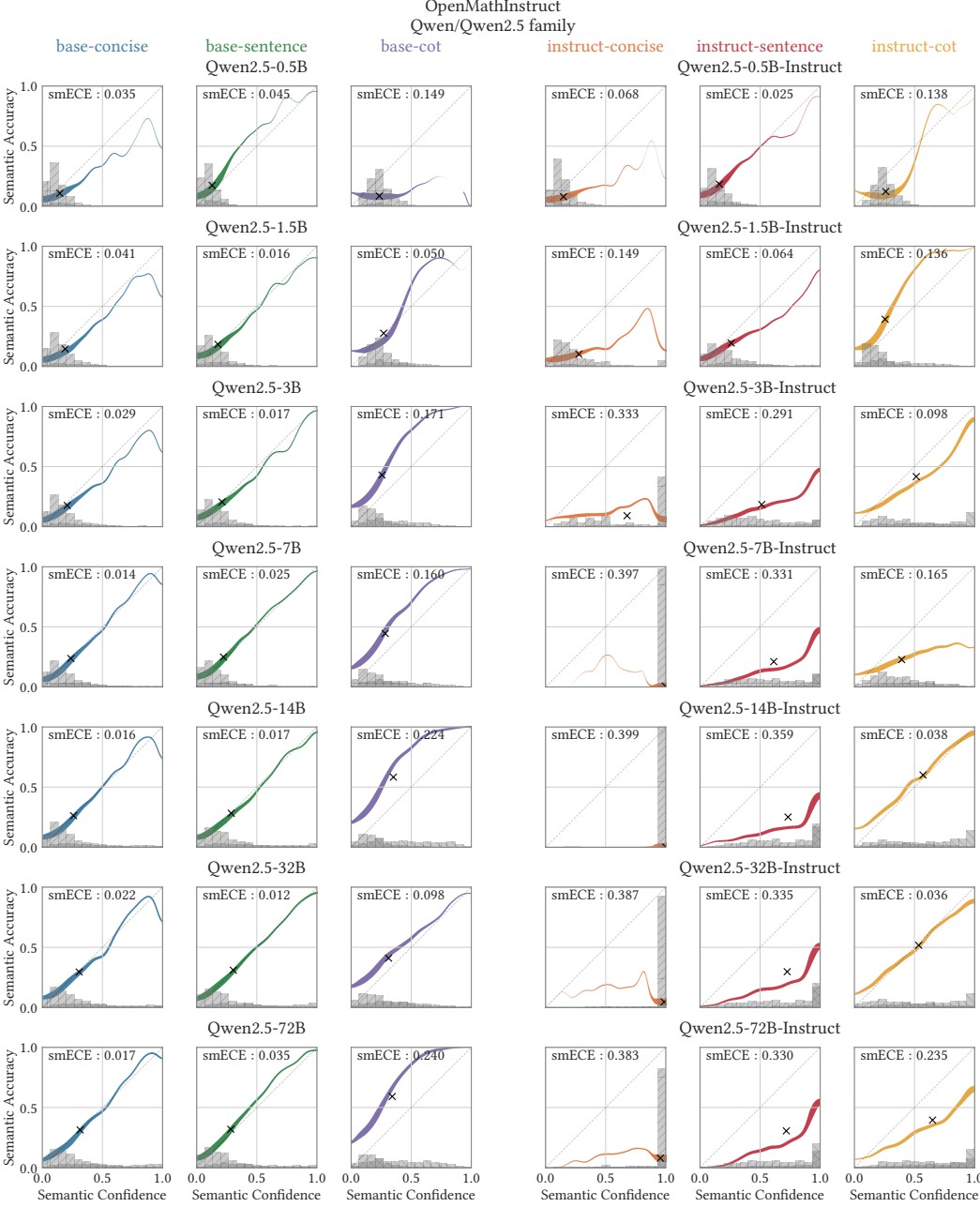

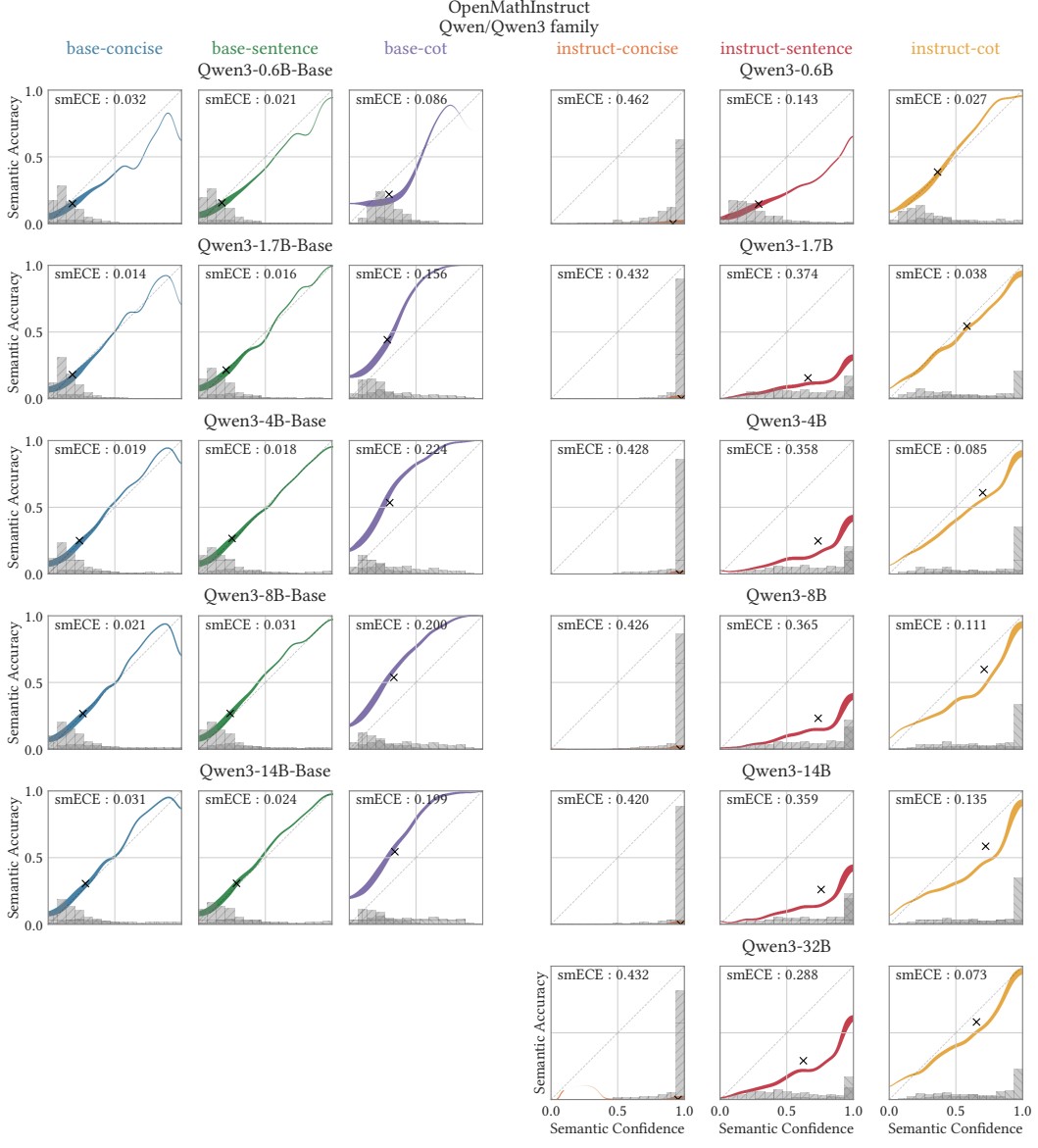

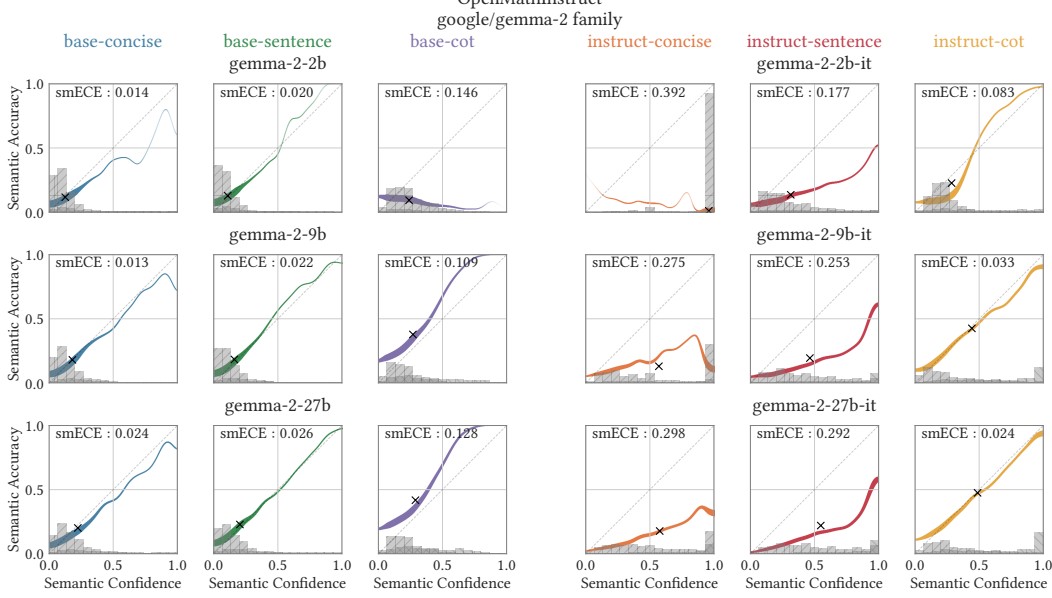

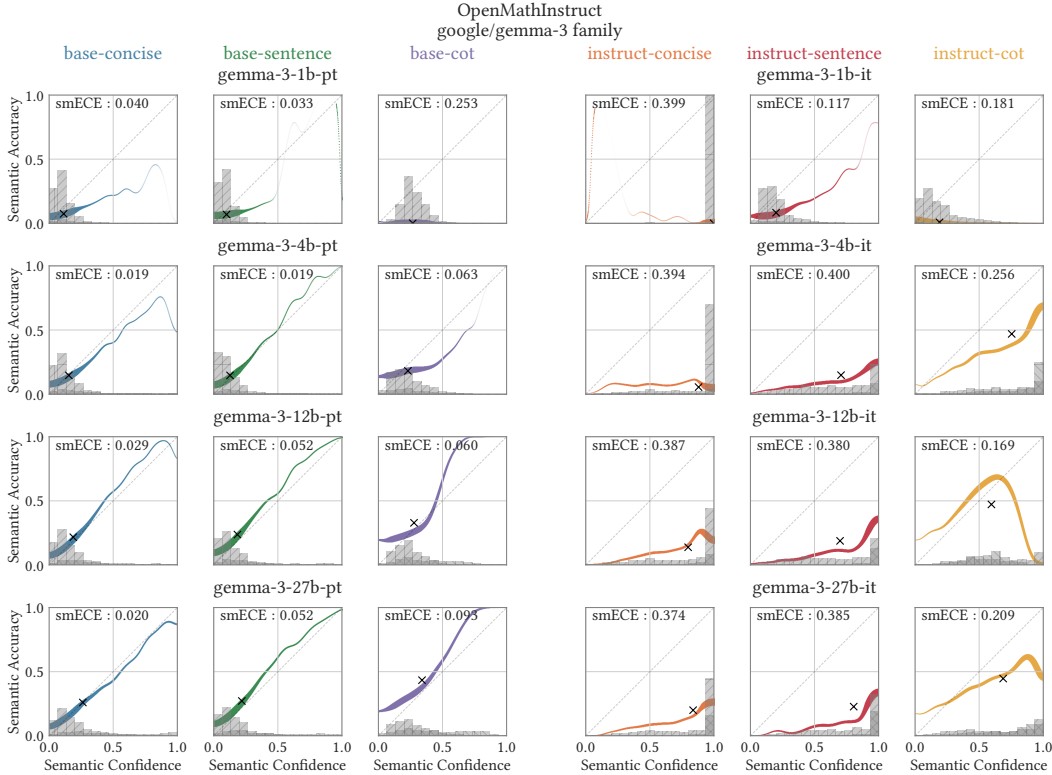

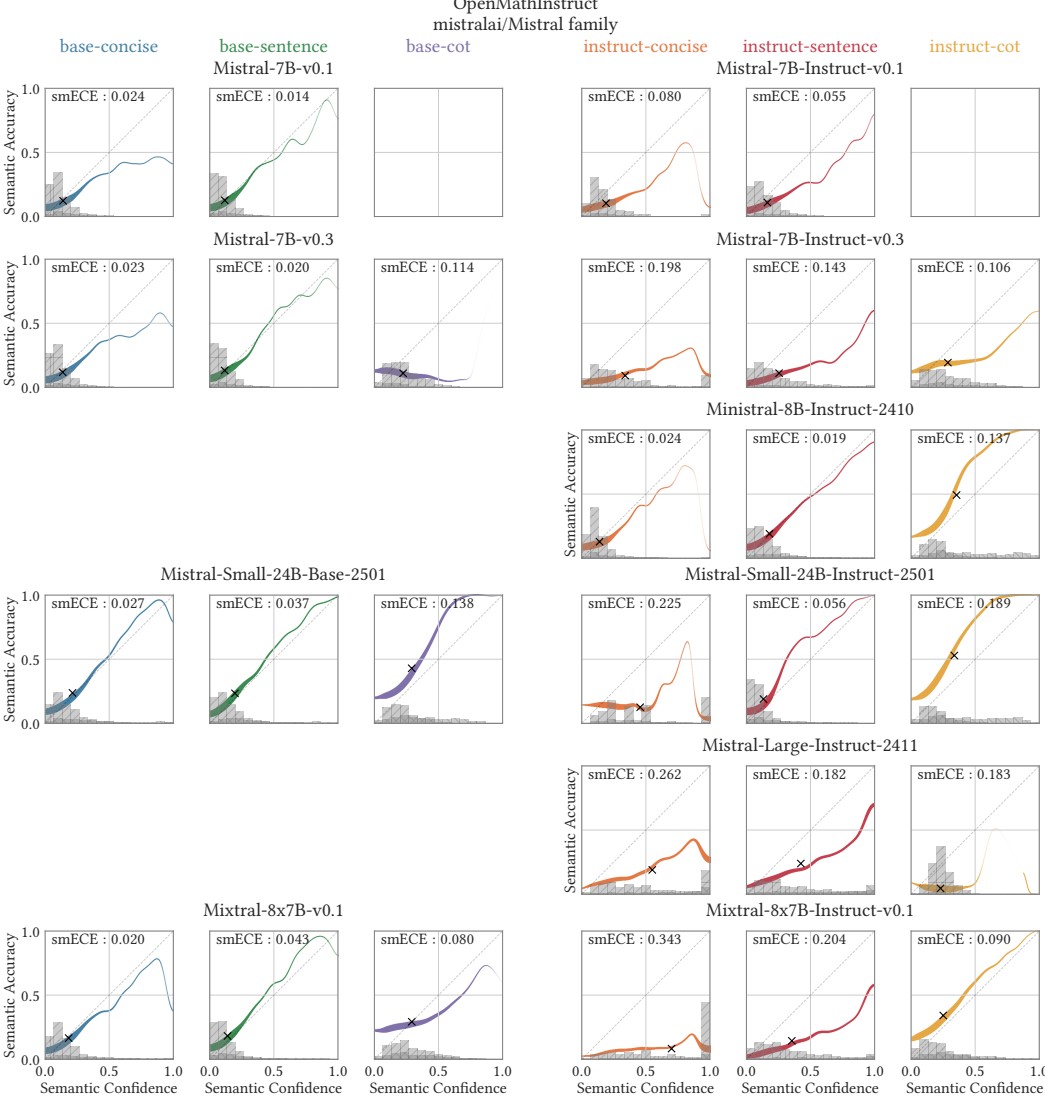

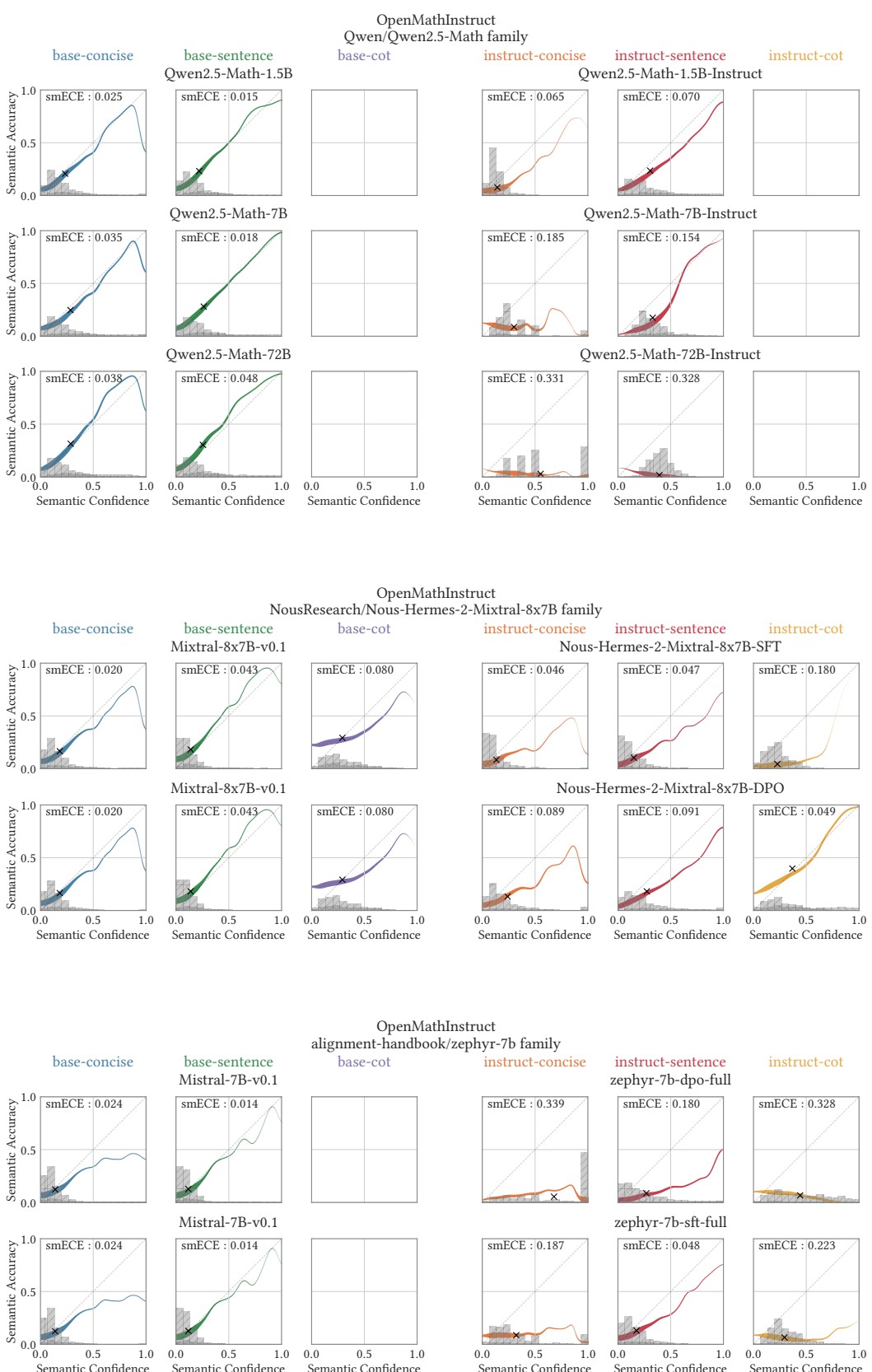

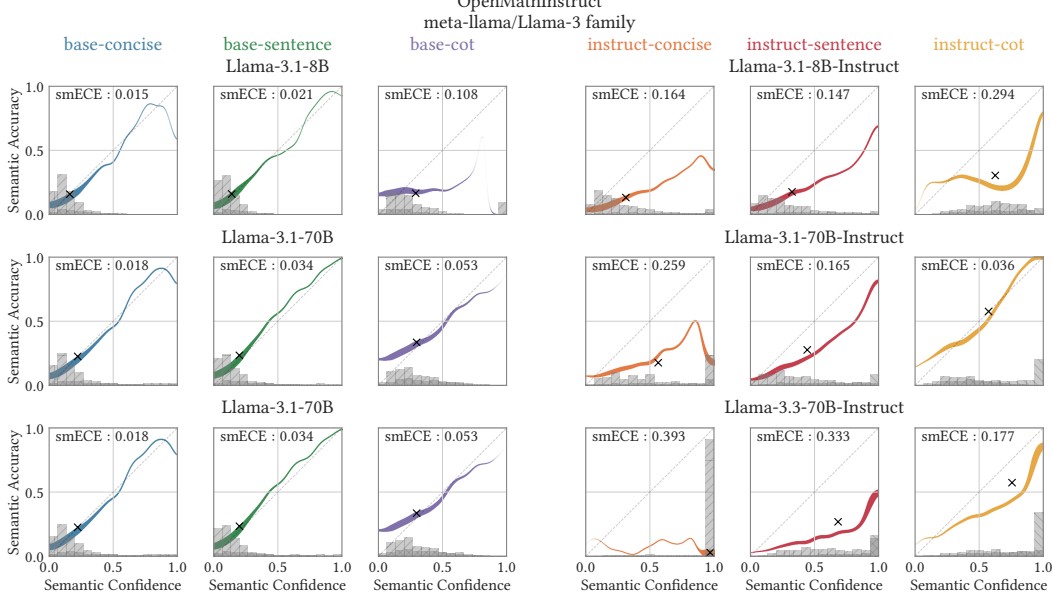

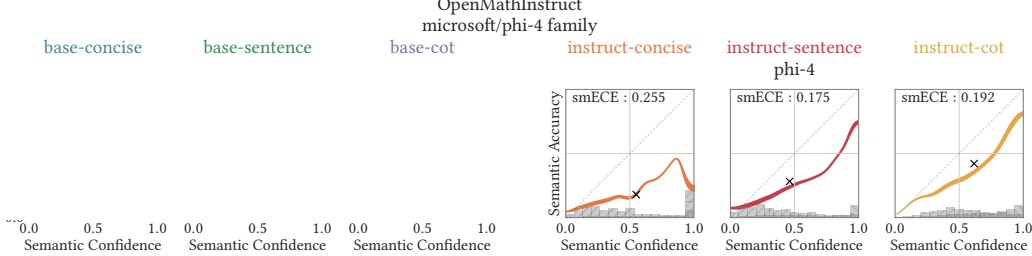

## F.3 TRIVIAQA

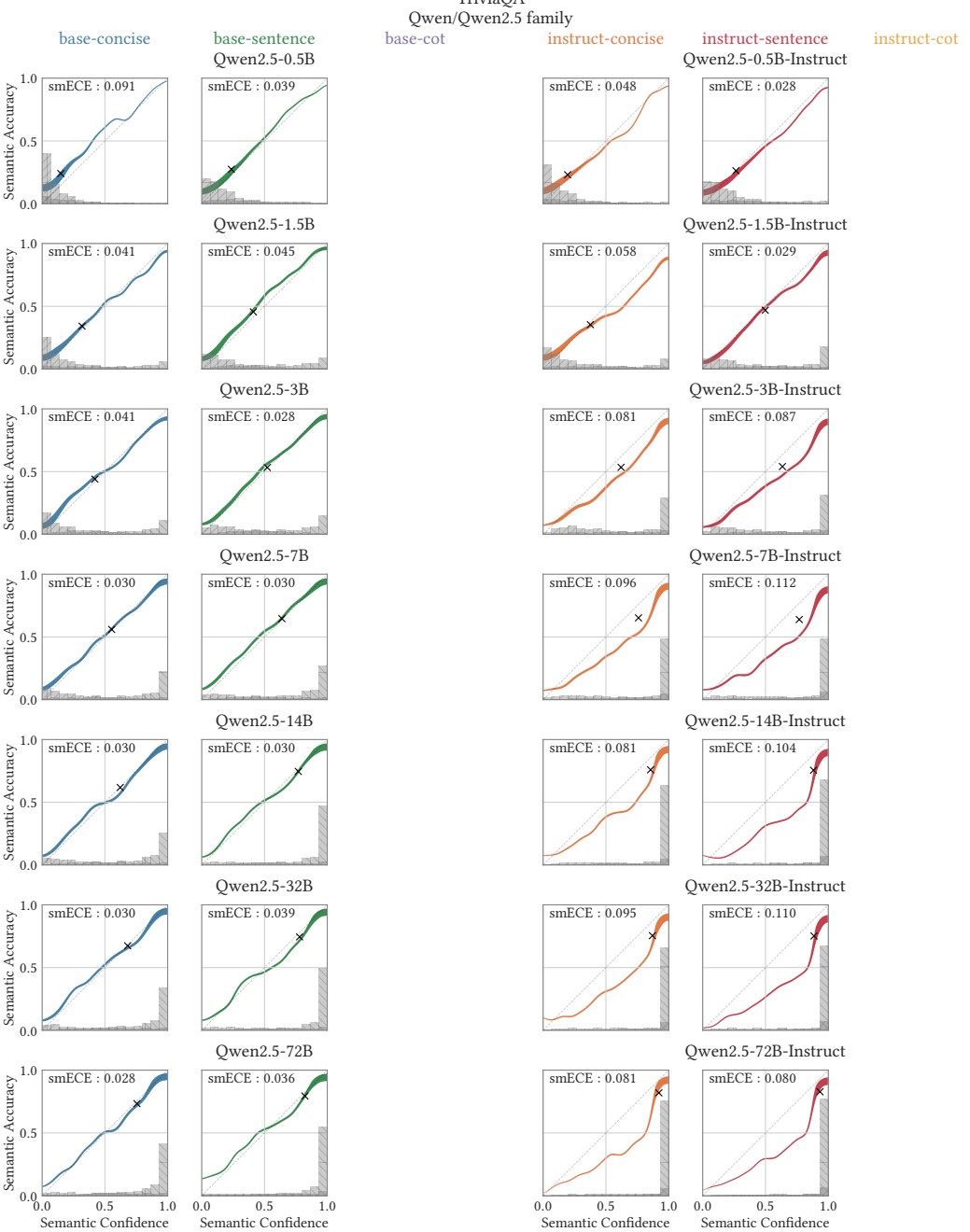

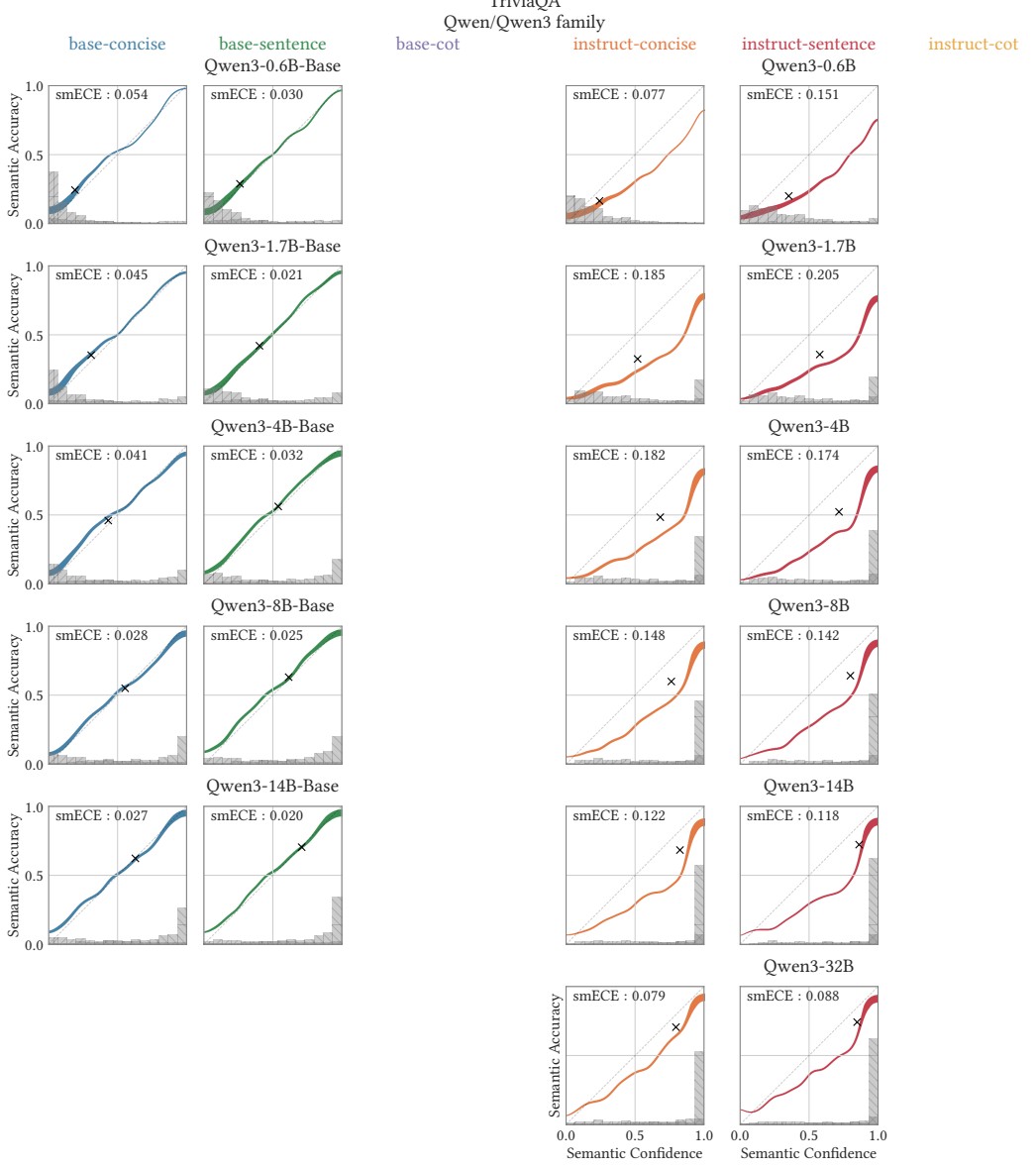

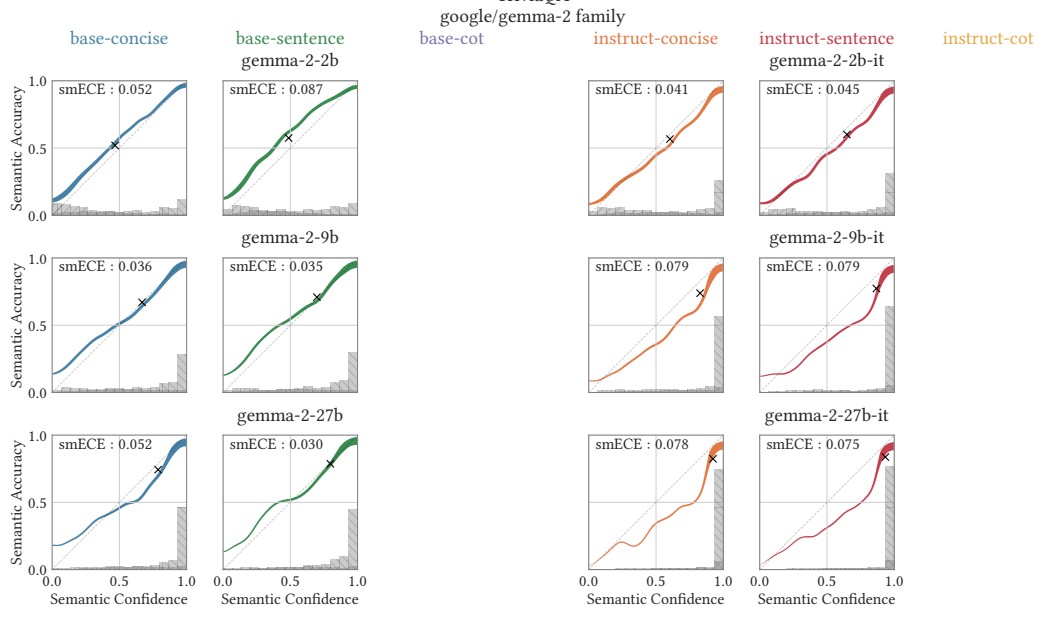

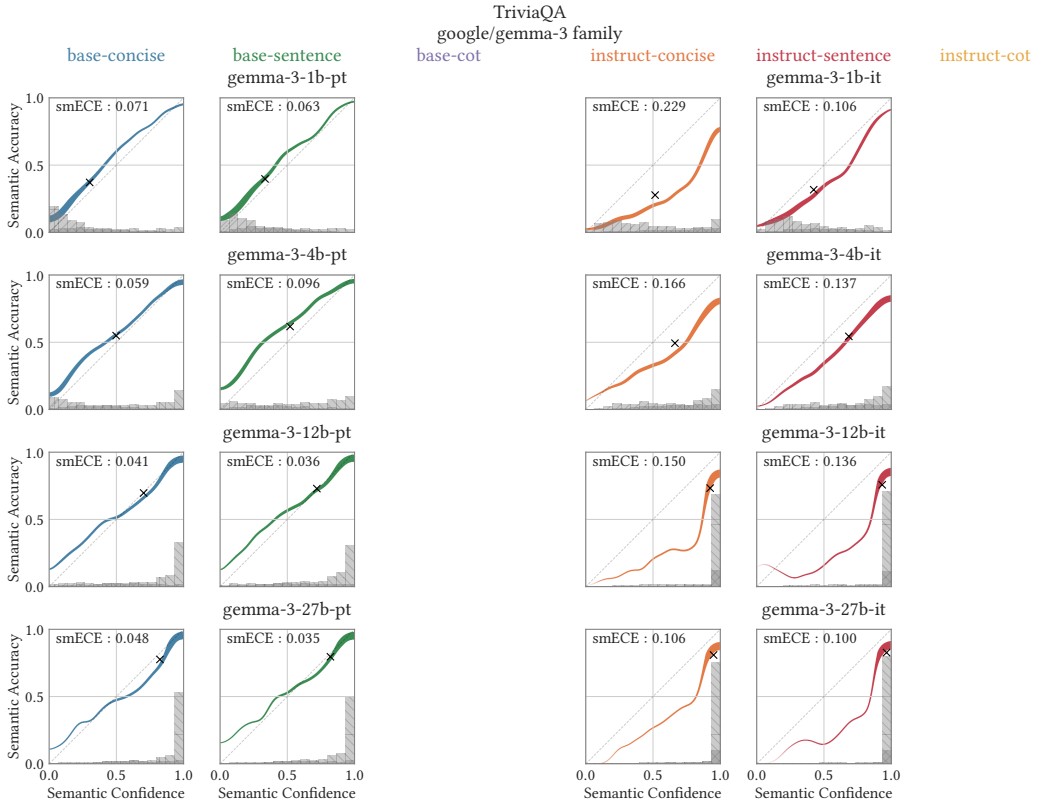

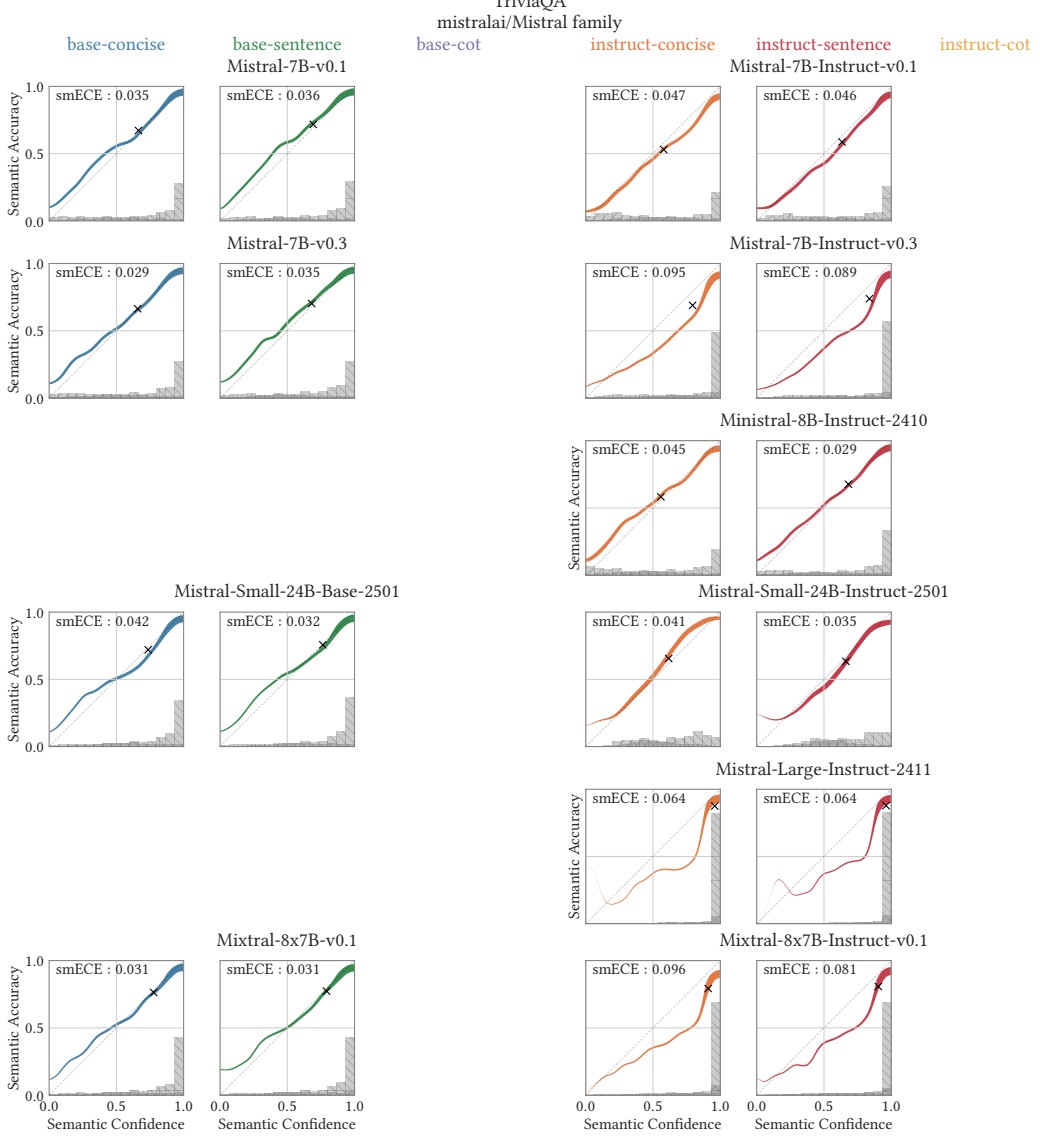

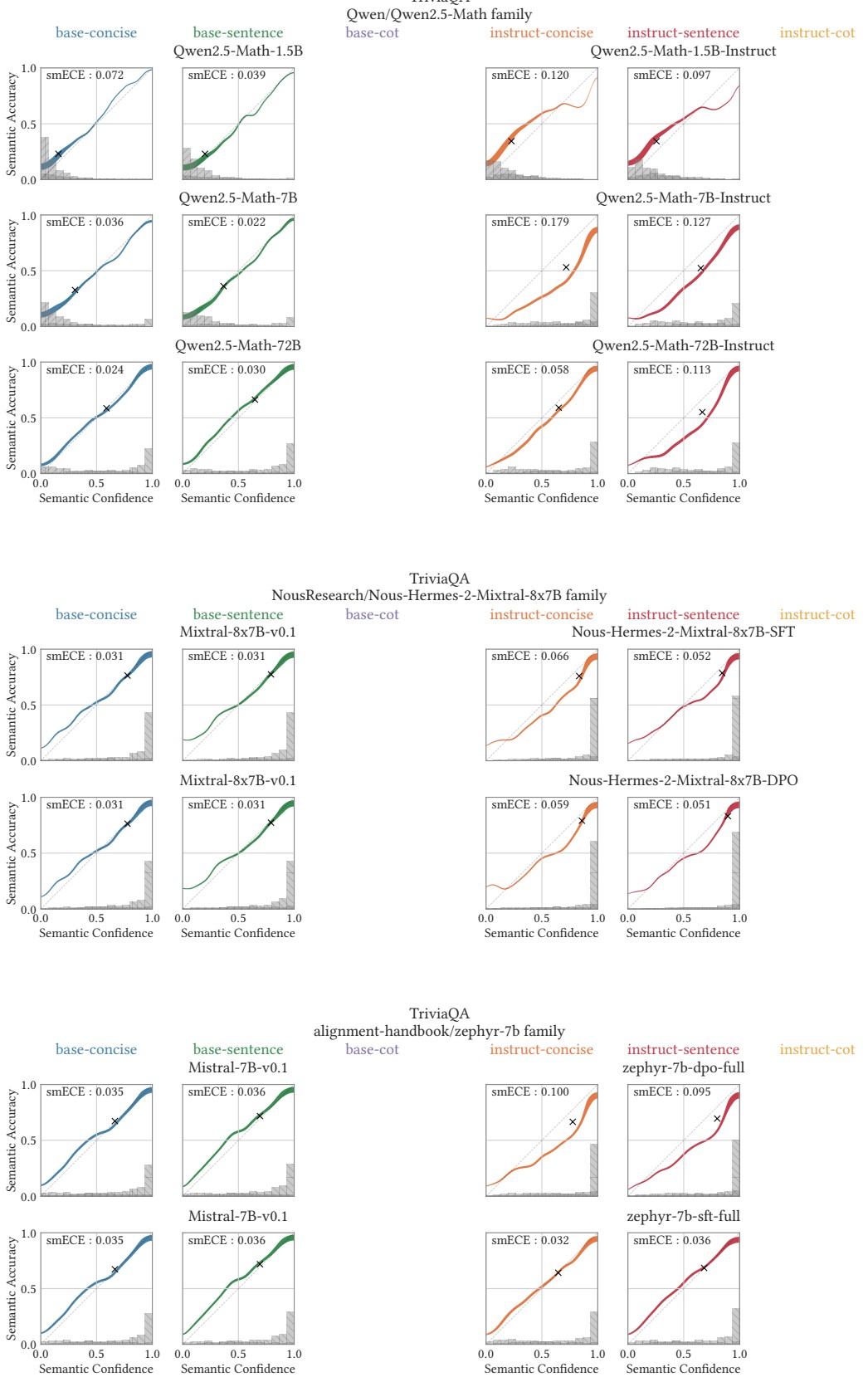

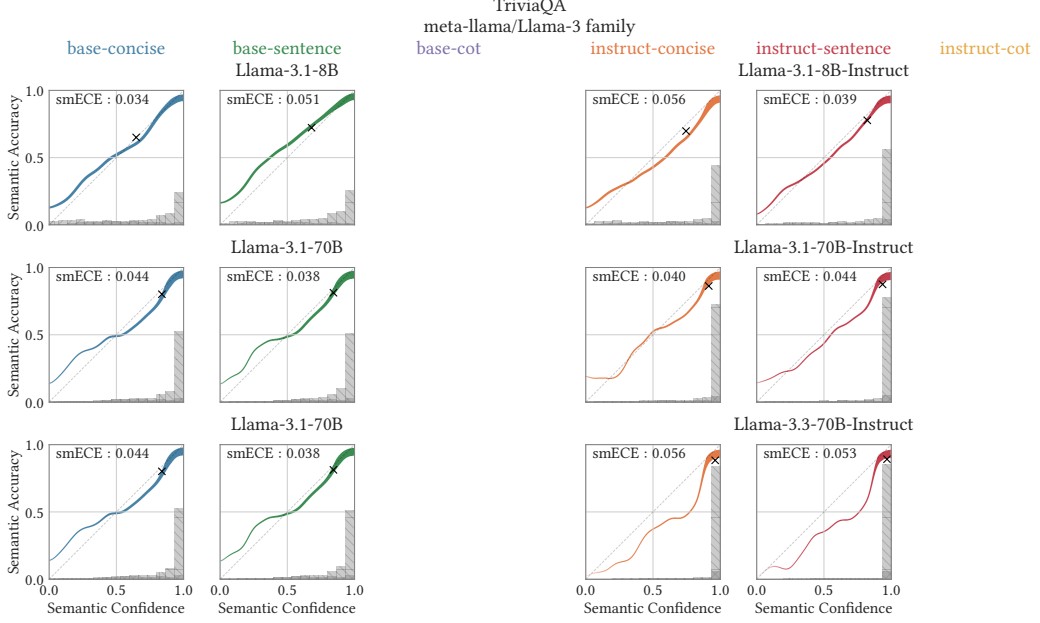

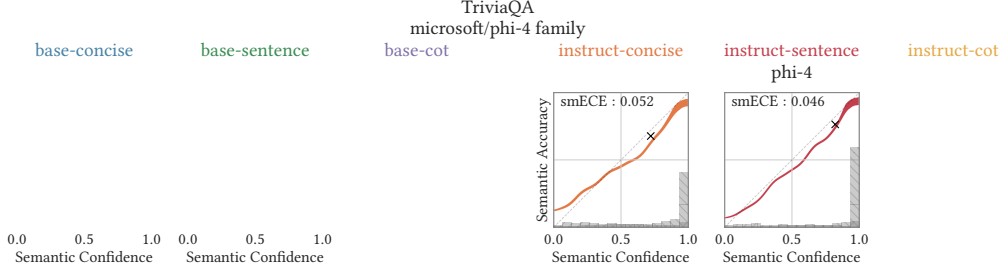

## F.4 SimpleQA

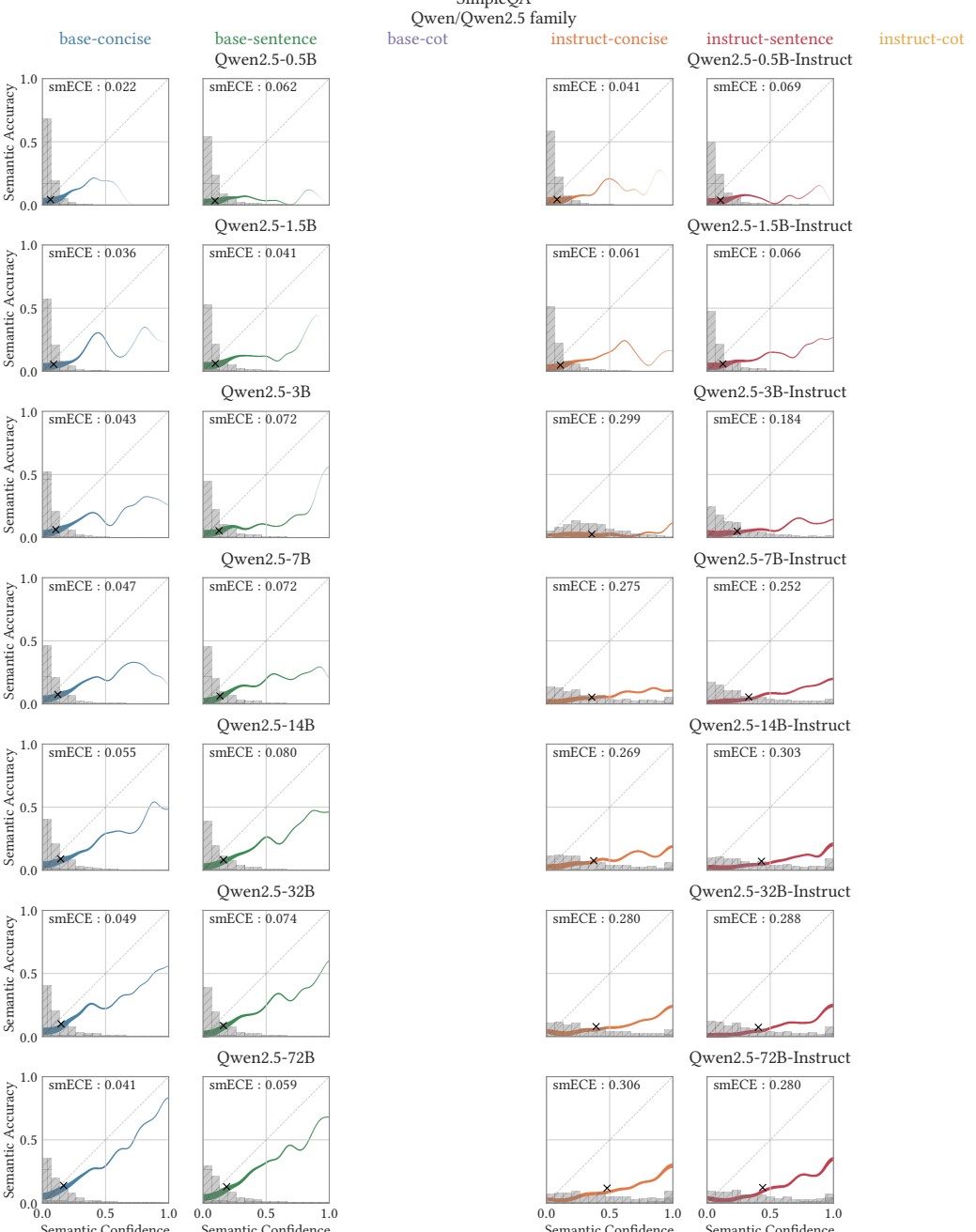

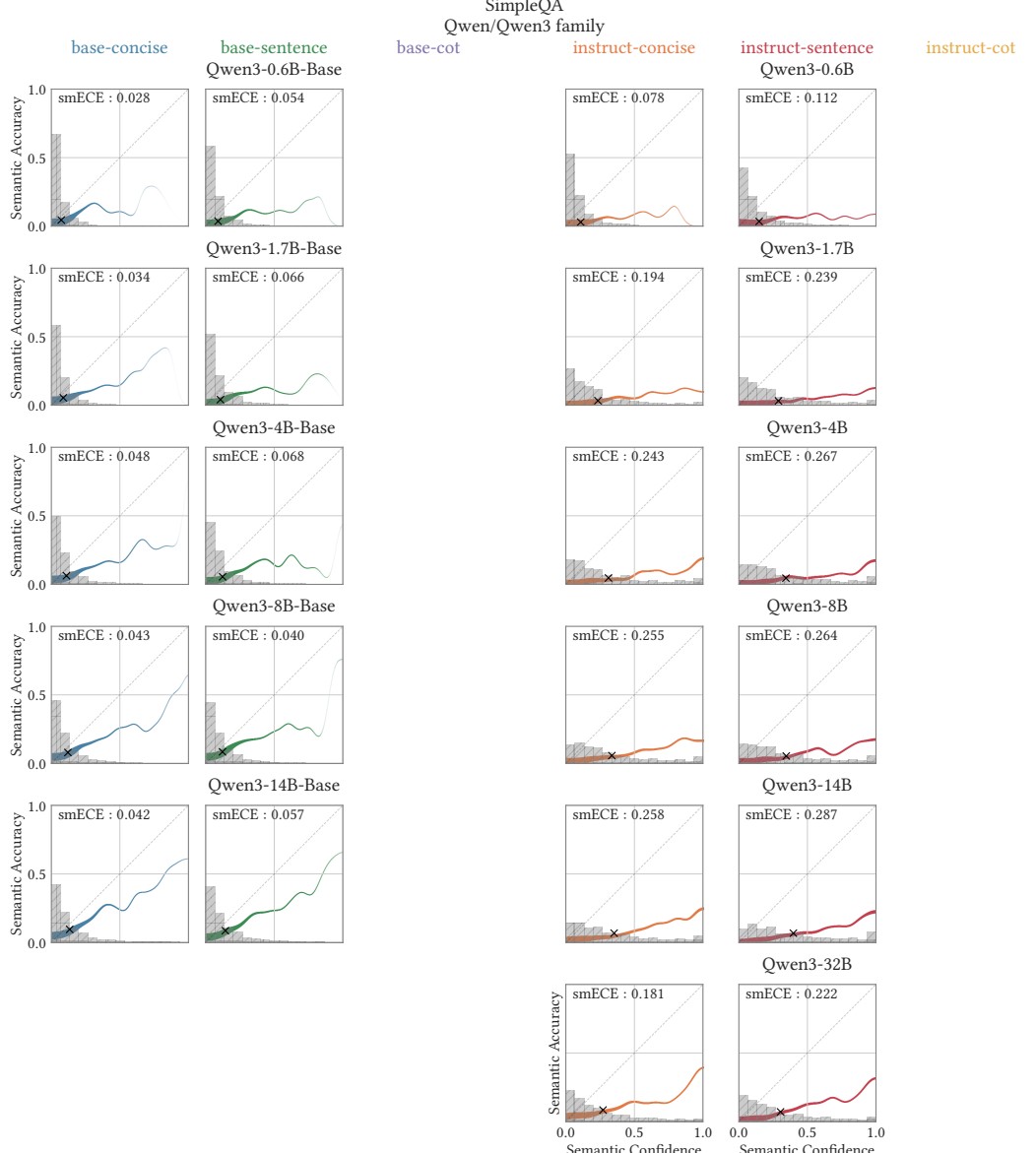

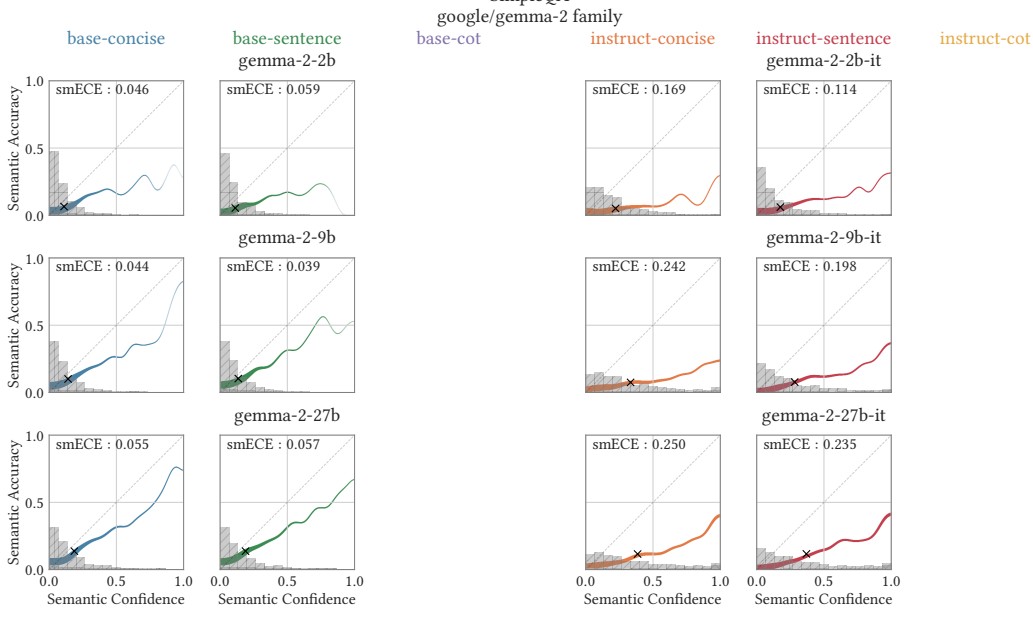

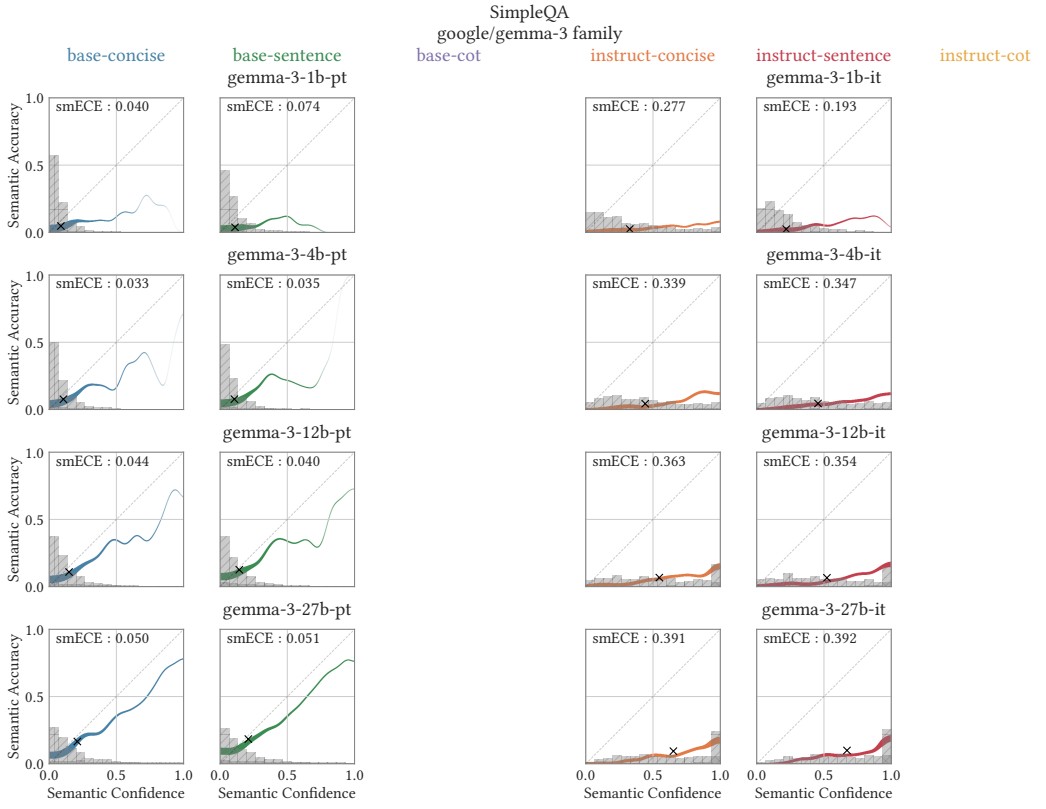

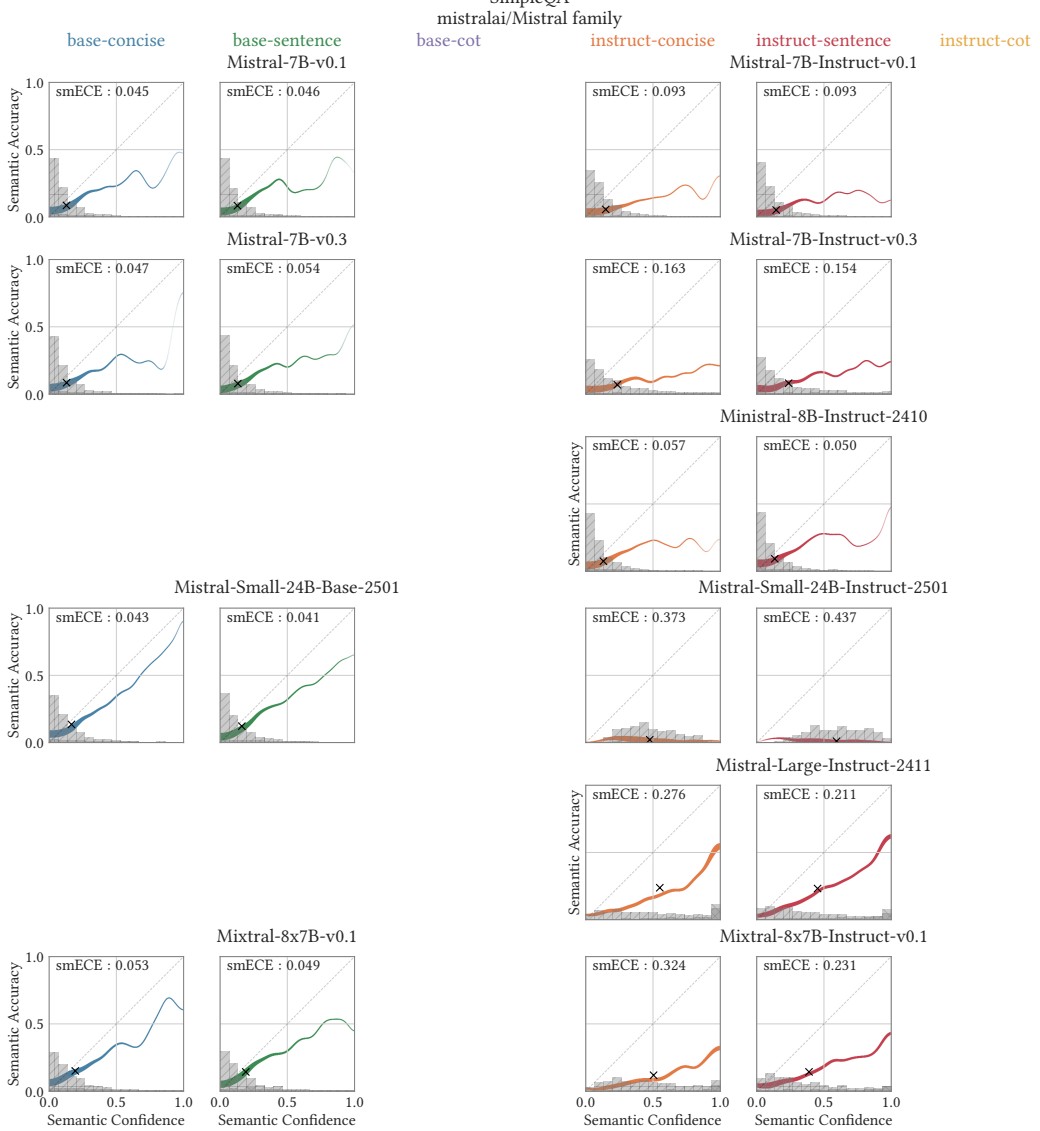

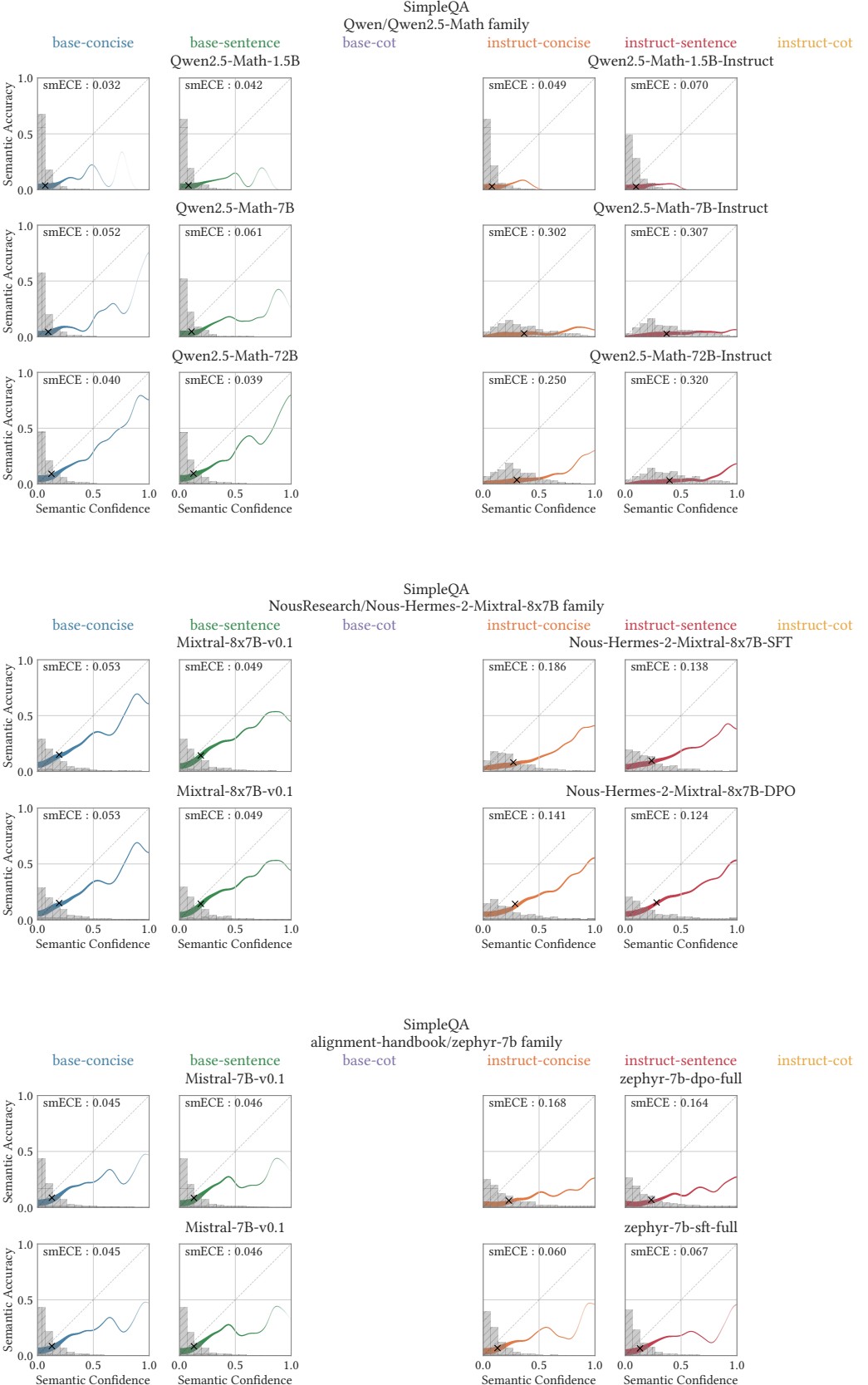

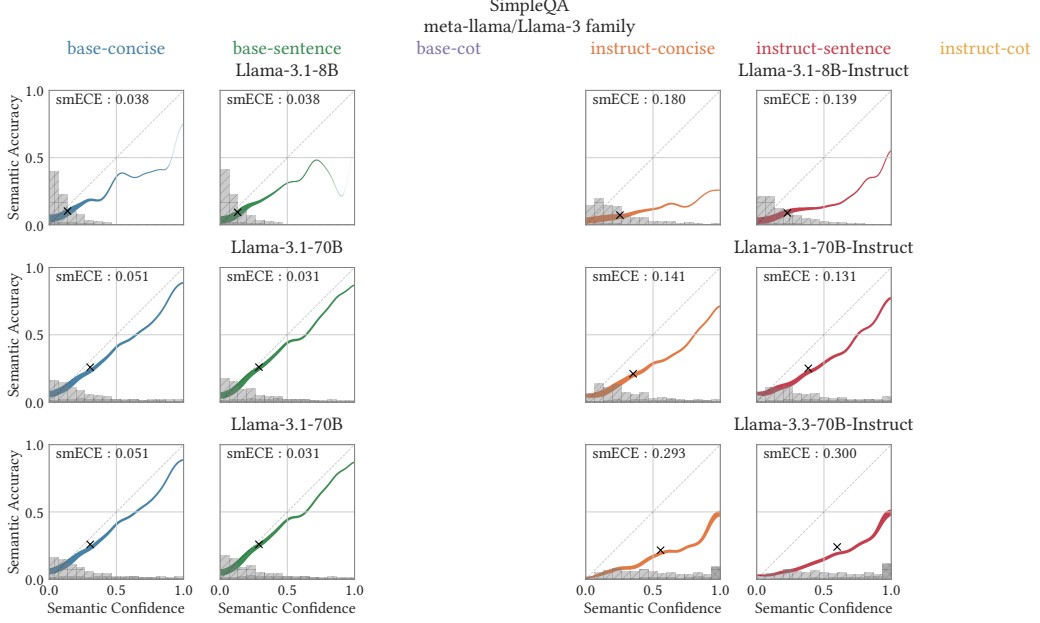

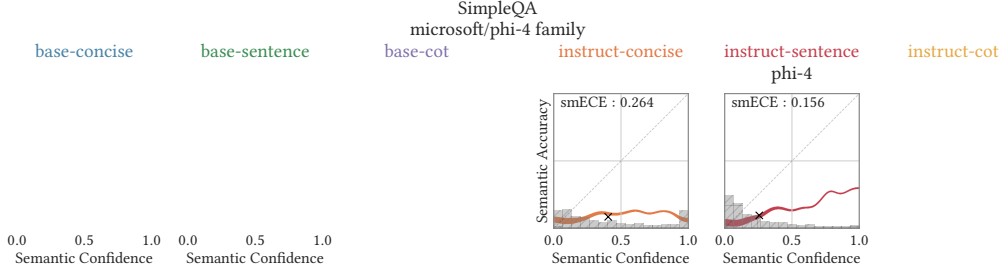

