# OpenReview forum: "Trained on Tokens, Calibrated on Concepts: The Emergence of Semantic Calibration in LLMs"
_ICLR.cc/2026/Conference — ICLR 2026 Poster_

### Official Review · Reviewer_mX9b · 2025-10-18

**Soundness:** 3
**Presentation:** 2
**Contribution:** 2
**Rating:** 4
**Confidence:** 3

**Summary:**

This paper introduces semantic calibration for LLM, where a model’s probability over semantic answer classes matches its empirical accuracy. It proposes a theoretical analysis to link calibration and local loss optimality. Experiments show that instruction tuning and Chain-of-thought (CoT) hurt LLM’s ability of semantic calibration.

**Strengths:**

1. This paper proposed semantic calibration instead of calibration over token-level confidence.

2. The theory can explain the relation between loss optimality and calibration performance.

3. The observations are very interesting. It is quite surprising to me that instruction tuning and CoT hurt the performance of semantic calibration.

**Weaknesses:**

1. The conclusion that CoT breaks calibration is drawn mainly from math datasets.

2. There is a misalignment between theory and experimental results. The theory aims to reveal that a LLM is calibrated if it’s locally optimal under its loss function, but it cannot explain the interesting observations as (1) base LLM is the best for calibration, (2) instruction tuning and CoT hurt the performance of semantic calibration.

3. Any method to alleviate the drop of calibration performance for instruction tuning or CoT?


4. Some Typos:

     4.1 In abstract, we introduce “B-calibration,” a notion... ->  we introduce “B-calibration”, a notation of…

    4.2 In Equation 1, is it formal to use math notation like $#$?

**Questions:**

Please refer to my comments in weakness.

---

> ### Author Response · Authors · 2025-11-18
> **Reply to mX9b**
>
> Thank you for your review and feedback.
> Regrading your questions:
>
> **Q1 (CoT):** We considered CoT only on math datasets because chain-of-though as a technique is used primarily in math settings. For “knowledge-based” datasets such as TriviaQA, for example, it is not clear what chain-of-thought should mean, or how it should be elicited.
>
> **Q2 (Theory vs Experiment)**:
> There might be a misunderstanding about our theoretical results. Our theory predicts the experimental behavior of base, Instruct, and CoT models, as we explain in Lines 379-395 (end of Section 4).
>
> To summarize:
> Our theoretical mechanism involves 3 steps (outlined in Section 3.1). Let’s call these the Blue, Green, and Orange steps following Figure 3. The point is, base LLMs in standard settings satisfy all 3 steps. But, instruction-tuning and CoT break at least one of the steps, in the following way:
>
> * Instruction Tuning: Breaks the Green step (since it uses a non-proper-loss)
> * Chain-of-Thought: Breaks the Blue step (since “the model usually does not know what its answer will be until it has finished thinking”)
>
> When any of these steps breaks, calibration is no longer guaranteed to hold, which is consistent with our experiments.
> Please let us know if the explanation in Section 4 was unclear, and we are happy to elaborate further.
>
>
> **Q3:** Our paper was focused on the scientific aspects of calibration in existing models, so we did not study new methods. However, we believe our results could help inform future methods work on improving calibration in these settings. In particular, our theory implies certain fundamental barriers which any method to improve calibration must overcome. This rules out certain approaches, and thus can help guide the search towards successful methods.
>
> **Q4:**
> * 4.1: “Notion” was intentional in this case.
> * 4.2: The “#” notation is the standard notation for the mathematical notion of “pushforward” of a probability measure. For completeness, we have added a foonote explaining this; thanks for pointing it out. Please see
> https://en.wikipedia.org/wiki/Pushforward_measure or https://link.springer.com/book/10.1007/978-3-319-20828-2
>
>
> Finally, if our clarifications resolve your concerns and this work meets the standard for ICLR, we would be grateful if you could consider adjusting your score accordingly.

---

### Official Review · Reviewer_2hGk · 2025-10-19

**Soundness:** 4
**Presentation:** 3
**Contribution:** 3
**Rating:** 8
**Confidence:** 2

**Summary:**

Authors proposes a theory explaining how large language models (LLMs), despite being trained only on next-token prediction, can still exhibit semantic calibration—the ability to assign meaningful confidence to the content of their answers. The authors introduce B-calibration, a framework that measures calibration with respect to any “collapsing function” B that maps textual outputs to semantic classes. They prove that B-calibration is equivalent to local loss optimality, meaning a model is calibrated if its loss cannot be reduced by simple perturbations of its output distribution. Empirically, they show that base LLMs (e.g., Mistral, Qwen, Llama) are semantically calibrated across open-ended QA tasks, while instruction-tuning and chain-of-thought reasoning often break this calibration—since these processes alter the model’s ability to anticipate its semantic output before generation. The findings suggest that semantic calibration naturally arises from likelihood training, providing a principled mechanism linking uncertainty estimation to optimization dynamics

**Strengths:**

It introduces the notion of B-calibration—a general framework for defining calibration over arbitrary equivalence classes of outputs—which unifies token-level and semantic-level calibration under a single formalism. This perspective reframes an underexplored question (“can base LLMs meaningfully assess confidence in their answers’ meanings?”) into a rigorous, testable problem, providing a clear theoretical bridge between semantic uncertainty and local loss optimality.

The work’s quality is high: the theoretical arguments are well-grounded in recent calibration theory, the proofs (especially the equivalence theorem and its autoregressive extension) are technically sound, and the experiments are extensive, covering multiple datasets, model families, and prompt styles.

**Weaknesses:**

Although the authors present calibration as emerging from the ability to “predict one’s own semantic output distribution,” this remains correlational. The LoRA probe experiment (Claim 10) demonstrates correlation between learnability and calibration but not causation.

The paper claims novelty in unifying calibration and loss-optimality, but related work in multi-calibration and conformal prediction is discussed mainly in the appendix. Bringing these connections into the main text—perhaps as a discussion of how B-calibration extends or simplifies existing frameworks—would situate the work more clearly within the calibration landscape.

**Questions:**

The paper assumes that base LLMs are “locally loss-optimal” with respect to simple perturbations. Could the authors provide empirical evidence or a small-scale ablation demonstrating this property, even approximately?

---

> ### Author Response · Authors · 2025-11-18
> **Reply to 2hGk**
>
> Thank you for your thoughtful review and feedback.
> Regarding your questions:
>
> > The paper claims novelty in unifying calibration and loss-optimality, but related work in multi-calibration and conformal prediction is discussed mainly in the appendix. Bringing these connections into the main text—perhaps as a discussion of how B-calibration extends or simplifies existing frameworks—would situate the work more clearly within the calibration landscape.
>
> Thanks for this suggestion. We added a remark at the end of Section 3.2 (“Technical Tools and Prior Work”, in blue), explaining in more detail the relation to prior work on loss-optimality and calibration, and some of the challenges we had to overcome in the LLM setting.
> For the camera-ready, we are also considering integrating a discussion of multi-calibration into the body (we initially chose to leave multi-calibration to the appendix just to simplify the presentation).
>
> > The paper assumes that base LLMs are “locally loss-optimal” with respect to simple perturbations. Could the authors provide empirical evidence or a small-scale ablation demonstrating this property, even approximately?
>
>
> Note that the experiments in the paper can already be interpreted as demonstrating a type of local-loss-optimality, for very specific perturbation families.
> By the theoretical equivalence of Theorem 6 (*), measuring B-calibration is equivalent to measuring local-loss-optimality for a specific class of perturbations W_B. Meaning, if a model has small B-calibration error (which we experimentally measured), this implies the model is locally-loss-optimal with respect to perturbations W_B.
>
> However, we agree that it would be interesting to test local-loss-optimality for other types of “simple perturbations”, to more thoroughly test this assumption. One experiment could be: try to fine-tune a [few-shot-prompted] base model on a specific distribution (e.g. GSM8K), using a small/weak LoRA. If the test loss doesn’t reduce significantly, this is evidence that the base model was locally-loss-optimal on GSM8K w.r.t. “perturbations that can be represented by a small LoRA.”
> This is an interesting experiment which we will consider for the future.
>
> (*): Technically, Theorem 6 only says that if the B-calibration error is exactly 0, then the model is perfectly locally-optimal w.r.t. W_B. We need a stronger statement, that roughly “the B-calibration-error both upper- and lower-bounds the gap to local-loss-optimality.” This stronger statement is proven as Theorem 36 in the Appendix, which is the “full version” of Theorem 6.

---

### Official Review · Reviewer_SbxL · 2025-10-31

**Soundness:** 3
**Presentation:** 3
**Contribution:** 2
**Rating:** 6
**Confidence:** 3

**Summary:**

The paper answers the question whether large autoregressive LLMs provide meaningful confidence about the semantic content of their long-form outputs (not only next-token probabilities). This is important both scientifically (do models know what they don't know?) and practically (calibrated confidences enable safer decision-making).

**Strengths:**

1. Parameterizing calibration by an arbitrary collapsing function B is a neat, flexible formalism that connects sampling-based semantic confidence to established calibration literature. This lets the authors transparently say which semantic granularity they evaluate.
2. There is a solid theoretical contribution linking diverse prior work.The equivalence (Thm.6) between B-calibration and local-loss-optimality is a meaningful bridge from optimization theory to semantic calibration; Thm.9 provides a plausible route for autoregressive implementation.
3. Multiple model families, base vs instruct variants, three prompting styles, and the LoRA probe present a coherent story that ties theory and practice together.

**Weaknesses:**

1. The chosen datasets primarily focus on an in-distribution assumption, which may limit the generality of the proposed method.
2. The claim  "instruction-tuning breaks calibration" may require deeper analysis.

**Questions:**

1. Experiments use four datasets (GSM8K, OpenMathInstruct-2, TriviaQA, SimpleQA). The theory relies on an "in-distribution" assumption (Claim 10 / Corollary 11) and the paper notes datasets like TruthfulQA may violate that and be miscalibrated. But many real-world culturally/evolving phenomena (memes, internet-slang, adversarial misinformation) are OOD and precisely the settings where calibration matters most.
The practical value of "semantic calibration" is highest in long-tailed, dynamic, and OOD settings. Showing calibration only in relatively conventional in-distribution QA may somehow limit applicability.
2. The empirical semantic confidence measurement requires many samples per prompt (M=50 at T=1) and then applying the B pipeline, which may be costly and may be impractical for real-time use.
It is suggested to provide experiments or analysis on how calibration estimates change with M (e.g., M=5,10,20,50) and temperature — to show whether cheaper approximations are viable.
3. It is not yet clear which aspects (objective mismatch, dataset selection, reward shaping) cause the breakdown. The DPO result is interesting (DPO model miscalibrated vs SFT-only less so) but needs deeper analysis. You may add some controlled ablations within a single model lineage varying (a) SFT size and loss (proper vs improper), (b) small RLHF/DPO steps with controlled reward shapes, and (c) temperature / decoding strategies. Report which factor correlates most tightly with calibration degradation.

---

> ### Author Response · Authors · 2025-11-19
> **Reply to SbxL**
>
> Thank you for your careful review, and for appreciating the strengths of our work. In our revised PDF, we have added experiments for TruthfulQA.
> We also agree with your stated weaknesses, and we will be sure to point out these limitations more prominently in the final version.
>
> Regarding your questions:
>
> **Q1 (“In-distribution assumption”):** We agree it would be impactful to extend our understanding of calibration to these “evolving” or OOD settings. However, understanding such settings theoretically is challenging, as it requires formalizing the type of distribution shift. There is no consensus on how best to theoretically formalize such OOD settings; existing techniques usually involve strong distributional assumptions. Thus, we chose to study the in-distribution setting in this work, which did not require distributional assumptions — though we hope future work can tackle more complex settings.
>
> However, we note that the “in-distribution” assumption still allows for a significant amount of flexibility in practice. For example, SimpleQA is a dataset of questions that were explicitly selected to be hard for GPT 3.5/4o. In some sense this is an “adversarially selected” dataset, which is not heavily represented in the pretraining distribution. Yet, we still observe (semantic) calibration on SimpleQA.
>
>
>
> **Q2 (Practical Efficiency):**
> Thank you for this suggestion. Changing ‘M’ is, as you said, mainly a matter of practical efficiency. Lower ‘M’ would result in higher-variance confidence estimates. However, changing the temperature ‘T’ will likely fundamentally change the nature of our results. Our theory relies crucially on T=1, and we expect calibration can fail at other temperatures (this is clear at the extremes, where e.g. T=0 will be fully overconfident). We will consider adding a discussion about these issues in the camera-ready.
>
>
> **Q3 (Instruction Tuning):**
> We agree understanding the precise cause of calibration failure for Instruct-models is interesting and deserves more investigation. We will likely leave most of this to future work, because a proper investigation will be a significant undertaking.
> Briefly, there are many potentially-confounding factors, since every Instruct model is trained with a “bespoke” procedure that is not always public (e.g. which datasets were used for Instruct-SFT, which datasets for DPO/GRPO/RLHF, for how long and in what order, etc). To properly disentangle these factors we should train our own high-quality Instruct-models (as you noted), but this would significantly expand the scope of the current work. We look forward to future work which investigates these aspects in more details.

---

### Official Review · Reviewer_ohGr · 2025-10-31

**Soundness:** 4
**Presentation:** 4
**Contribution:** 3
**Rating:** 8
**Confidence:** 3

**Summary:**

The paper studies whether base LLMs (trained only by next-token prediction) are calibrated not just at the token level but at the semantic level for open-ended QA. It proposes a formal framework—$B$-calibration—that treats calibration with respect to semantic equivalence classes induced by a collapsing function B. Empirically, the authors estimate “semantic confidence” by sampling multiple generations for a question collapsing each answer to a semantic class (using an extractor) to obtain a distribution over classes, and then checking if confidence matches accuracy (standard reliability notion). The main claims are: (1) base LLMs show strong semantic calibration under their sampling-based procedure; (2) instruction-tuning breaks this calibration; and (3) chain-of-thought (CoT) reasoning also breaks it. A diagnostic LoRA experiment supports the mechanism: when a small LoRA can easily predict the model’s own semantic class distribution (low KL gap), the underlying model is better calibrated.

**Strengths:**

The paper is a pleasure to read and provides a nice interplay of theory guided empirical experiments to show the existence of semantic calibration in base models.  As I am not a theory person and don't have a background in multi-class calibration and its relation with loss functions, I am unable to assess the correctness of the main theorems. However, I was able to follow the intuition which broadly makes sense to me and connects well with existing literature. The main conceptual takeaway from the paper is useful and easy to follow.

### Originality

1. Provides a clear, general formalism—$B$-calibration—that parameterizes semantic calibration by an arbitrary collapsing function, decoupling the theory from any single definition of “semantics.”
2. The theoretical link between $B$-calibration and local loss optimality over $W_B$ perturbations offers a principled mechanism for when semantic calibration should emerge in maximum-likelihood–trained LLMs.
3. The paper focuses on open-ended Q/A which is important and useful going forward as prior work on MCQ is already well-studied.

### Quality

1. The mechanism yields intuitive predictions (base models should be calibrated; instruction-tuning and CoT harm calibration), and the paper reports experiments that align with all three.
2. The LoRA diagnostic is a thoughtful probe: correlating the ease of predicting the model’s own semantic class distribution with empirical calibration quality is a useful sanity check that directly targets the proposed mechanism.
3. The empirical measurement procedure for semantic calibration is clearly articulated: sample multiple completions at fixed temperature, collapse to classes, compute confidence and accuracy, then assess calibration. This is methodologically straightforward and reproducible in principle.

### Clarity.

1. The paper is very-well written and easy to follow. The writing is to the point and backed with relevant prior works.

2. The sections are neatly organized and I thank the authors for providing an opening summary for each section on the content about to follow (giving clarity to the reader) and often providing intuition in simple words.

3. The paper carefully motivates why token-level calibration is insufficient for long-form, open-ended tasks and explains the need for semantic calibration. The role of $B$ and the sampling-based estimator is described concretely. Fig.1 clearly shows the contribution of the paper in a simple and useful manner.

4. The “from calibration to local optimality” storyline and the three-step mechanism (two proved, one heuristic) are presented in an accessible way.

5. The plots for Section 5 are beautiful and experiments are well-executed.

### Significance.

If base LLMs are often semantically calibrated under this operational definition, then knowing when to trust base-model answers via sampling becomes a practical tool, and the observation that instruction-tuning and CoT can degrade calibration is a nontrivial, actionable takeaway for practitioners who care about reliability.
I look forward to seeing follow up work based on the concepts established in this work, especially towards verabalized confidence.

**Weaknesses:**

I don't have any issues with material covered in the paper which is thoroughly presented. The authors have presented a solid work worthy of acceptance. The only drawback I see is the limited applicability of this theory and the exclusive focus on their B-confidence calibration which the authors acknowledge in Section B.1.

While this work greatly extends our understanding of calibration on a semantic level for base LLMs, the semantic level is still limited to at most a sentence-level (as longer reasoning would fall into the CoT category). The authors state in footnote 3 that, "We will however assume that the latter holds (which is plausible if each evaluation distribution is a reasonably-sized sub-distribution of the pretraining distribution on which local-loss-optimality holds)."

This assumption assumes evaluation distribution to be already covered in the pertaining distribution. While this often done targetedly by model practitioners, it is unreasonable to expect test query to be already covered by training data. LLMs may be used for OOD tasks and also benefit significantly from inference time-compute (as it is unreasonable to expect an LLM to produce an answer, especially for challenging tasks, in a single forward pass). Thus, to me, it is not surprising that token-level calibration can lead to sentence-level (in this work, *limited to a single sentence*) calibration as the model is saying the same thing but instead of spiting it out directly, it just says the same thing in a sentence. What is more interesting is whether models are/can be made calibrated on long-form generations [1]. I say this more towards future work as the paper is already dense and contains a sizable contribution!

Minor: some plots are missing in Appendix figures like on page 44 for phi-4 and page 46 for Qwen3.

[1] Band, N., Li, X., Ma, T., & Hashimoto, T. (2024). Linguistic calibration of long-form generations. arXiv preprint arXiv:2404.00474.

**Questions:**

Q1: OpenMath-Instruct2 is a dataset generally used for training (as it has more than >1M samples). Any reason to choose a train set for evaluation? How many samples did your test set of OpenMath consist of? I would rather suggest using MATH [2] for evaluation. I don't expect models to be able to solve questions from MATH in just a single word or sentence so I assume this might not follow the properties required by your theory but I still would be interested to see whether base models end up *not* being calibrated on MATH.

Q2: The authors note that some datasets (e.g., TruthfulQA) likely violate in-distribution assumptions and may behave differently. This is a great point and I actually would be interested to see whether the predicted breakdown on TruthfulQA (or something else) happens empirically.

Q3: For a given model and task (prompt), was the semantic classification performed on just 3 samples (1 for each prompting methodology)?

[2] Hendrycks, D., Burns, C., Kadavath, S., Arora, A., Basart, S., Tang, E., ... & Steinhardt, J. (2021). Measuring mathematical problem solving with the math dataset. arXiv preprint arXiv:2103.03874.

---

> ### Author Response · Authors · 2025-11-18
> **Reply to ohGr**
>
> Thank you for your detailed review, and for appreciating our work.
> We agree with your evaluation, and we are also looking forward to future work exploring more fine-grained/complex types of calibration (e.g, linguistic calibration), and other ways of extracting confidences (such as verbalized-confidences).
>
> Regarding your questions:
>
> **Q1:** We have now added evaluations on MATH500 (a subset of questions from MATH) — please see the updated Figure 4 and 5 for these results. The results are consistent with our theoretical predictions:
>
> * When base models answer without chain-of-thought, they are calibrated (although they are low-accuracy, since the problems are hard).
> * When base models use CoT reasoning, they are much more accurate, but no longer calibrated.
>
> Note that MATH500 consists of only 500 questions, so the estimated reliability diagrams are noisier than for our other datasets.
>
> We would have liked to evaluate on the full MATH dataset, but there were certain internal legal licensing restrictions which prevented us from publishing MATH evals. In the initial version, we used OpenMathInstruct-2 (OMI2) as essentially a permissively-licensed proxy for the MATH dataset. Note that 84% of OMI2 are problems directly derived from MATH (either the original problem or a synthetically-augmented variant). Thus we believe OMI2 is a reasonable proxy for MATH; in particular, it contains a large number of problems of similar difficulty. We evaluated on 10K random samples from OMI2.
>
> **Q2:**
> Thank you for your interest and the suggestion. We took the opportunity to run TriviaQA experiments, and updated Figure 4 and Figure 5 to include these results. Looking at Figure 4, we can see TruthfulQA has a noticeably higher calibration error than TriviaQA and SimpleQA.
>
> **Q3:**
> For each model, dataset, and prompt-style, we take a random sample of 10K questions, generate 50 LLM-responses per question, and perform the semantic collapsing on these 50 samples to get a confidence.
> (If we misunderstood your question, please let us know and we can elaborate).

---

### Author Response · Authors · 2025-11-19
**Summary of changes**

We thank all reviewers for their time and constructive feedback. We have responded to each reviewer individually, and we have directly updated the PDF incorporating several suggestions.
The main changes in the PDF are (highlighted in blue):

* Added experiments on TruthfulQA and MATH500, and updated Figures 4 and 5 to reflect these results.
* Added “Remark: Technical Tools and Prior Work” at end of Section 3.2

---

### Meta-Review · Area_Chair_r9Yv · 2026-01-06

**Summary:**

Across reviewers, the common theme is that the paper makes a clear conceptual and theoretical contribution. Specifically, the paper formalizes semantic calibration and connects it to local loss optimality, supported by broad experiments that largely match the paper’s predictions. All major concerns raised by the reviewers have been addressed in the rebuttal, making it a clear accept.

**Reviewer Concerns:**

Specifically, concerns about in-distribution assumptions and OOD behavior raised by Reviewer SbxL, dataset choice and evaluation coverage raised by Reviewer ohGr, theory–experiment alignment and the effect of instruction tuning / CoT raised by Reviewer mX9b, and positioning with respect to prior work and local loss optimality raised by Reviewer 2hGk were explicitly addressed through new experiments, added clarifications, and revisions to the theoretical discussion.

**Reviewer Scores:**

Reviewer mX9b likely moves from 4  to 6.  The major concern that the “theory vs experiment mismatch” was clearly clarified.

---

### Decision · Program_Chairs · 2026-01-26

Accept (Poster)